# Single-cell transcriptome analysis indicates fatty acid metabolism-mediated metastasis and immunosuppression in male breast cancer

Handong Sun[1,23], Lishen Zhang[2,3,4,23], Zhonglin Wang[5,23], Danling Gu[6,23], Mengyan Zhu[2,3,4], Yun Cai[2,3,4], Lu Li[2,3,4], Jiaqi Tang[2,3,4], Bin Huang[2,3,4], Bakwatanisa Bosco [2,3,4], Ning Li[2,3,4], Lingxiang Wu[2,3,4], Wei Wu[2,3,4], Liangyu Li[2,3,4], Yuan Liang [2,3,4], Lin Luo[2,3,4], Quanzhong Liu[2,3,4], Yanhui Zhu[1], Jie Sun[7], Liang Shi[1], Tiansong Xia[1], Chuang Yang[1], Qitong Xu[1], Xue Han[8], Weiming Zhang[8], Jianxia Liu[7], Dong Meng[9], Hua Shao[5], Xiangxin Zheng[10], Shuqin Li[11], Hua Pan[12], Jing Ke[13], Wenying Jiang[14], Xiaolan Zhang[15], Xuedong Han[16], Jian Chu[17], Hongyin An[17], Juyan Ge[18], Chi Pan[19], Xiuxing Wang [6,20,21], Kening Li [2,3,4] ✉, Qianghu Wang [2,3,4,22] ✉ & Qiang Ding [1] ✉

Male breast cancer (MBC) is a rare but aggressive malignancy with cellular and immunological characteristics that remain unclear. Here, we perform transcriptomic analysis for 111,038 single cells from tumor tissues of six MBC and thirteen female breast cancer (FBC) patients. We find that that MBC has significantly lower infiltration of T cells relative to FBC. Metastasis-related programs are more active in cancer cells from MBC. The activated fatty acid metabolism involved with *FASN* is related to cancer cell metastasis and low immune infiltration of MBC. T cells in MBC show activation of p38 MAPK and lipid oxidation pathways, indicating a dysfunctional state. In contrast, T cells in FBC exhibit higher expression of cytotoxic markers and immune activation pathways mediated by immune-modulatory cytokines. Moreover, we identify the inhibitory interactions between cancer cells and T cells in MBC. Our study provides important information for understanding the tumor immunology and metabolism of MBC.

Male breast cancer (MBC), a malignant tumor accounting for 1% of all breast cancers[1], is generally diagnosed at a late stage, with a higher degree of malignancy, poorer prognosis, and higher mortality than female breast cancer (FBC)[2]. The overall mortality rate of MBC patients is 19% higher than that of FBC patients because of the clinical characteristics and lack of treatment[3]. Most MBC patients are hormone receptor-positive, similar to late-onset, postmenopausal estrogen receptor (ER) and progesterone receptor (PR) positive (ER⁺PR⁺) FBC[4]. Thus, the clinical management of male patients refers to FBC due to the currently limited understanding of MBC.

The clinical and pathological characteristics of MBC do not entirely overlap FBC[5]. Studies have shown that the ER loci associated with patient prognosis are sex-selective[6]. Hormonal status has raised concerns regarding the use of aromatase inhibitors in male patients,

---

and the best choice of endocrine therapy for MBC is still controversial[6]. Furthermore, the energy metabolism and immune response to malignancy are different between males and females[7,8], such as bladder and lung cancers[9,10]. However, cellular and molecular differences between MBC and FBC remain unclear. The undiscovered pathological characteristics of MBC may contribute to the poor outcome in male patients. Therefore, it is urgent to clarify the tumor microenvironment and metabolism features of MBC to better understand the underlying mechanisms of MBC development.

The complex composition of the tumor microenvironment presents a considerable challenge to study the molecular mechanism of MBC. The tumor microenvironment comprises various cell types, including immune cells, fibroblasts, endothelial cells, and extracellular components surrounding cancerous cells[11]. The immune cells such as T cells and macrophages are reported to play important roles in the tumor immunology and progression of FBC[12,13]. A comprehensive understanding of the tumor microenvironment could provide essential information for developing novel therapeutic strategies for breast cancer. With the emergence of single-cell RNA sequencing (scRNA-seq) technologies, we can now dissect the tumors containing multiple cell types and describe the complex interplays among cancer cells and the microenvironment[14]. ScRNA-seq has been utilized to explore the intratumoral heterogeneity and microenvironment of FBC, providing potential therapeutic targets for female patients[15,16]. However, the cellular states and immunological characteristics of MBC still need further analysis at the single-cell level.

In this work, we utilize scRNA-seq and scTCR-seq technology to explore the tumor microenvironment of breast cancer and compare the immunological and metabolic features between MBC and FBC samples. Our study indicates that the elevated level of *FASN*-mediated fatty acid metabolism is related to the cancer cell metastasis and low immune infiltration of MBC. Moreover, our data reveal the dysfunctional and specific metabolism patterns of T cells in the MBC microenvironment. Our study provides further information for understanding the tumor immunology and metabolism of MBC and sheds light on the development of therapeutic strategies to improve the prognosis of MBC patients.

## Results
### MBC exhibited a lower immune infiltration than FBC
To explore the cellular diversity in breast cancer (BRCA), we integrated the scRNA-seq data of 6 MBC and 13 FBC samples (Fig. 1a). ScTCR-seq was also performed on three MBC and two FBC samples to characterize the T cell receptor clonality. All the collected samples were ER-positive. The clinical characteristics of these samples were shown in Supplementary Data 1. Considering some clinicopathological characteristics such as tumor stage may be associated with the immune microenvironment and metabolism of patients, we compared the clinical characteristics of the collected MBC and FBC samples. Results showed that there were no significant differences in age, HER2 status, KI67 level, and extent of the tumor (T1–T4) between the FBC and MBC groups (Supplementary Data 2), avoiding the influence of these factors on the comparison. An overview of our single-cell analysis was shown in Fig. 1a. By analyzing the expression of marker genes, we annotated the various cell types in the BRCA ecosystem, including epithelial cells, T cells, B cells, plasma cells, macrophages, mast cells, myofibroblasts, cancer-associated fibroblasts (CAFs), arterial endothelial cells, venous endothelial cells, and capillary endothelial cells (Fig. 1b, c and Supplementary Data 3). Based on the chromosomal landscape inferred by scRNA-seq data, we distinguished malignant epithelial cells from non-malignant microenvironment cells (Supplementary Figs. 1 and 2). The genes specifically expressed in each cell type were identified (Fig. 1d). A significant differential enrichment was observed between different sexes, demonstrating the different microenvironment components between MBC and FBC patients (Fig. 1c).

Results showed that compared with FBC, MBC showed a significantly higher proportion of cancer cells and a lower proportion of immune cells, such as T cells and B cells, indicating a lower level of immune infiltration (Fig. 2a–d). These immune cell proportions had no obvious differences between premenopausal and postmenopausal FBC patients (Fig. 2e). To further validate this result, we calculated the scores of various cell types for 722 ER⁺ TCGA-BRCA samples based on the gene signatures derived from our single-cell data (see "Methods"; Fig. 2f). These scores between premenopausal and postmenopausal FBC patients were also compared (Fig. 2g). Results verified that MBC had a relatively higher tumor purity and lower proportions of T cells and B cells, consistent with the observation at the single-cell level. The immunological components of TCGA samples were also verified using three immune-deconvolution tools, including MCP-counter[17], EPIC[18], and xCell[19]. We evaluated the correlation of putative cell type levels derived from single-cell signatures and immune-deconvolution tools and found a significantly positive correlation between these methods (Supplementary Fig. 3a). Consistently, results from immune-deconvolution tools indicated that the levels of T cells and B cells were significantly lower in MBC samples than in FBC samples of the TCGA dataset (Supplementary Fig. 3b). To further verify this result with larger MBC sample size, we also collected two gene expression profiles of MBC samples from previous studies, including RNA-seq data of 46 MBC samples (GSE104730)[6] and microarray data of 74 MBC samples (GSE31259)[20]. We calculated and compared the scores of immune or stromal cells for MBC samples from three datasets, and for FBC samples from the TCGA dataset. Results showed that the scores of T cells and B cells were significantly lower in MBC samples from three independent datasets than in FBC samples (Supplementary Fig. 4a), further confirming the results of low immune infiltration in MBC samples. In order to figure out whether the HER2 status has an influence on the comparison of cellular components between MBC and FBC, we further compared the immune infiltration among groups of ER⁺HER2⁺ MBC, ER⁺HER2⁻ MBC, ER⁺HER2⁺ FBC, and ER⁺HER2⁻ FBC samples. Both the scRNA-seq data and TCGA-BRCA data consistently showed that the ER⁺HER2⁻ MBC samples had the highest level of cancer cell enrichment and significantly lower level of T cell and B cell percentages (Supplementary Fig. 4b, c). Besides, it seemed that the T and B cell percentages were higher in ER⁺HER2⁺ MBC than in ER⁺HER2⁻ MBC samples, although further evaluation was needed in a larger cohort. Furthermore, we performed immunohistochemistry (IHC) analysis for T cell markers CD4 and CD8 in 30 ER⁺ MBC and 30 ER⁺ FBC samples. Results suggested that these T cell markers had a lower expression proportion in MBC than in FBC samples (Fig. 2h, i). Therefore, the analysis of scRNA-seq, bulk transcriptome and IHC consistently demonstrated that MBC had a significantly lower degree of immune infiltration than FBC, especially the lower infiltration of T cells and B cells.

### The metastasis-related programs, and regulons controlled by AR and SREBF1 were significantly activated in MBC cancer cells
In order to further compare the transcriptional pattern of cancer cells between MBC and FBC, we re-clustered the 53,343 cancer cells from 19 BRCA patients (Fig. 3a, b). The cancer cell clusters showed sex-based differences (Fig. 3c). According to the proportion of cancer cells from MBC patients in each cluster, we classified these clusters into three subgroups, including MBC, FBC, and mixed clusters (see "Methods"; Fig. 3d). The differentially expressed genes between MBC and FBC cancer cell clusters were identified and shown in Supplementary Fig. 5. Genes involved in fatty acid metabolism such as *FASN* and *AZGP1* had a higher expression in MBC clusters than in FBC clusters. Previous studies have shown that *FASN* can enhance invasion in breast cancer[21]. We further explored the metastasis-related signature scores in MBC and FBC cancer cells (Supplementary Data 4). Results showed that MBC had higher

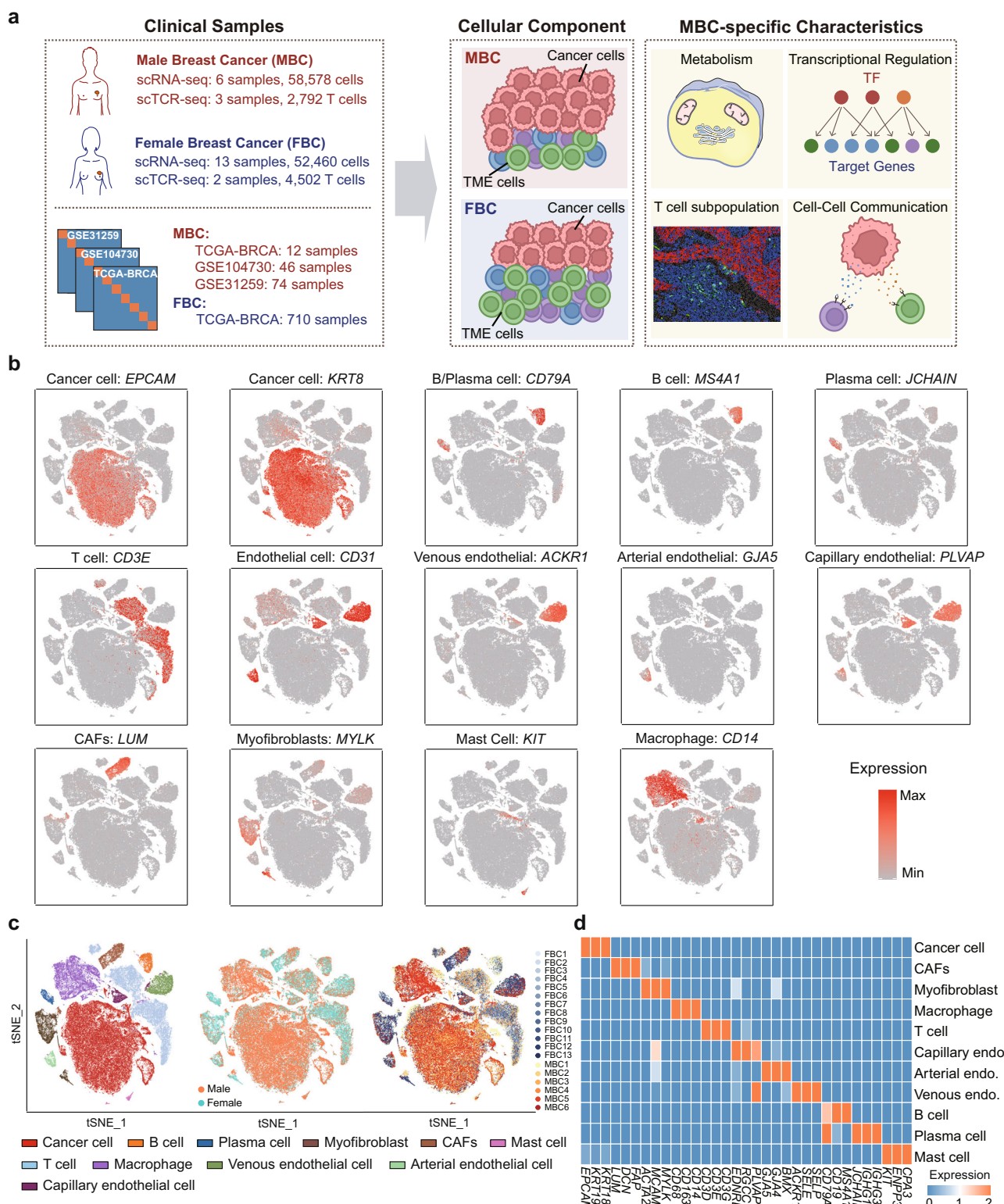

**Fig. 1 | Single cell transcriptome landscape of breast cancer patients.**
**a** Schematic workflow for data collection and single-cell analysis process in this study. **b** Log normalized expression (gray to red) of marker genes of each cell type. **c** The *t*-distributed stochastic neighbor embedding (t-SNE) plots of cells types and resources profiled in this study. Colored by cell types, sex and samples respectively. **d** Heatmap showing genes (columns) that were differentially expressed across various cell types (rows). Scale bar depicting low expression in blue and high expression in orange. Source data are provided as a Source data file.

signature scores of cell migration, epithelial–mesenchymal transition (EMT), and angiogenesis than cancer cells in FBC (Fig. 3e). Besides, cancer cells from both ER⁺HER2⁺ and ER⁺HER2⁻ MBC showed higher scores of metastasis-related signatures than FBC, especially angiogenesis and cell migration (Supplementary Fig. 6). These results suggested the higher metastasis potential of MBC, which is consistent with the clinical observations.

To reveal the specifically activated transcriptional regulons of MBC clusters, we identified the transcriptional factors (TFs) with differential activity between MBC clusters and other clusters

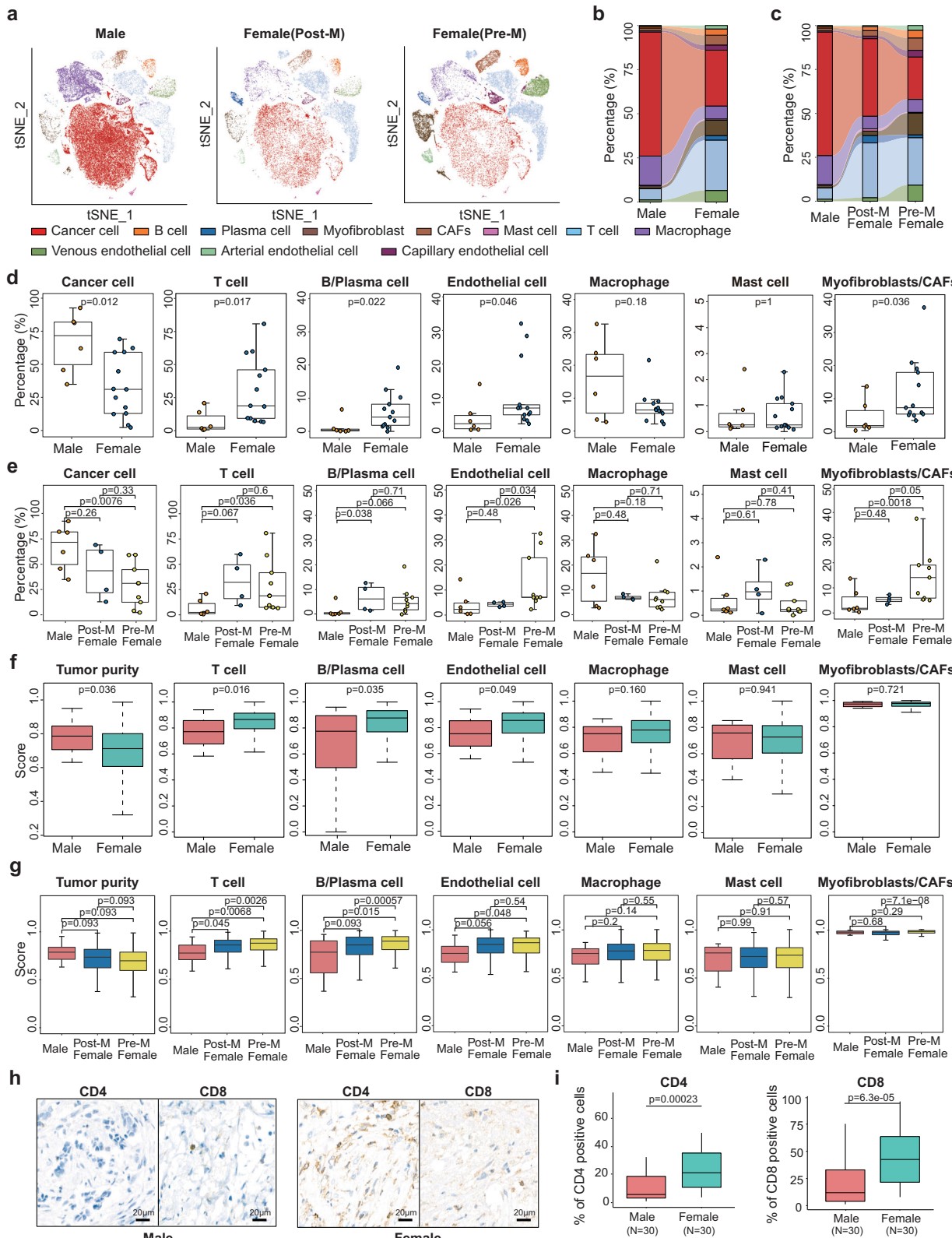

(Fig. 3f). Both the TF activity and expression of androgen receptor (*AR*) and sterol regulatory element binding transcription factor 1 (*SREBF1*) showed significant upregulation in cancer cells from MBC, compared with FBC (Fig. 3g, h). Previous studies have shown that as an important regulator of lipid metabolism, *SREBF1* could promote tumor growth and metastasis of breast cancer, and was highly associated with EMT process[22,23]. To further evaluate the observation

of AR, we retrospectively investigated the AR levels evaluated by IHC in a large sample cohort, including 113 ER⁺ MBC and 86 ER⁺ FBC samples (Fig. 3i, j). Results showed that the percentage of AR-negative patients was significantly lower in MBC than in FBC samples (5.3% vs. 17.4% in MBC and FBC samples, respectively), whereas the percentage of AR+++ patients was higher in MBC than in FBC samples (69.9% vs. 50.0% in MBC and FBC samples, respectively). This

**Fig. 2 | Comparison of cellular components between MBC and FBC samples.**
**a** The t-SNE plot of MBC, postmenopausal and premenopausal FBC samples. Colors represent cell types. **b** Sankey diagram showing the fraction of each cell type between male and female samples. **c** Sankey diagram showing the fraction of each cell type between MBC, postmenopausal and premenopausal FBC samples. **d** Boxplot showing the percentage of cancer cells, T cells, B/Plasma cells, endothelial cells, macrophages, mast cells and myofibroblasts/CAFs in MBC ($n = 6$) and FBC ($n = 13$) samples. P value was calculated by two-sided Wilcoxon rank-sum test. **e** Boxplot showing the percentage of cancer cells, T cells, B/Plasma cells, endothelial cells, macrophages, mast cells and myofibroblasts/CAFs in MBC ($n = 6$), postmenopausal ($n = 4$) and premenopausal ($n = 9$) FBC samples. P value was calculated by two-sided Wilcoxon rank-sum test. **f** Boxplot showing the tumor purity

and signature scores of various cell types between MBC ($n = 12$) and FBC ($n = 710$) in TCGA ER$^+$ BRCA cohort. P value was calculated by two-sided Wilcoxon rank-sum test. **g** Boxplot showing the tumor purity and signature scores of various cell types between MBC ($n = 12$), postmenopausal ($n = 337$) and premenopausal ($n = 372$) FBC samples in TCGA ER$^+$ BRCA cohort. P value was calculated by two-sided Wilcoxon rank-sum test. **h** Representative images of IHC staining detecting T cell markers CD4 and CD8 expression in MBC ($n = 30$) and FBC ($n = 30$) samples, Scale bar, 20 μm. **i** Boxplot indicating the IHC scores of CD4 and CD8 in 30 ER$^+$ male (light-coral) and 30 ER$^+$ female (turquoise) patients (identified by the percentage of positive cells). P value was calculated by two-sided Wilcoxon rank-sum test. In **d**–**g** and **i**, box plots show median (center line), the upper and lower quantiles (box), and the range of the data (whiskers). Source data are provided as a Source data file.

result further validated the activated AR regulon in MBC patients observed at the single-cell level.

## The activated fatty acid metabolism was related to the metastasis and low immune infiltration of MBC

To identify the potential differences in cancer cell metabolism between MBC and FBC samples, we evaluated the activity of metabolic pathways in each cancer cell cluster (Supplementary Data 5) and identified the specifically activated pathways in male clusters (Supplementary Fig. 7). Results showed that the fatty acid metabolism-related pathways were significantly more active in the MBC cancer cell clusters, including fatty acid biosynthesis, fatty acid elongation, fatty acid degradation, and biosynthesis of unsaturated fatty acids (Fig. 4a). As an essential enzyme for de novo lipogenesis[21], *FASN* was remarkably up-regulated in cancer cells from MBC than in FBC samples (Fig. 4b, c and Supplementary Fig. 8a). Cancer cells from both ER$^+$HER2$^+$ and ER$^+$HER2$^-$ MBC samples showed higher expression of *FASN* than FBC samples, independent of HER2 status (Supplementary Fig. 8b). To further validate this result, we compared the gene expression of ER$^+$ MBC and FBC in the TCGA cohort. Consistently, it showed that the expression of *FASN* was significantly higher in MBC patients (Fig. 4d and Supplementary Fig. 8c, d). Moreover, the IHC staining for FASN in 30 ER$^+$ MBC samples and 30 ER$^+$ FBC samples were compared. Results showed that the protein levels of FASN were remarkably higher in MBC than in FBC samples (Wilcoxon rank-sum test, *p*-value: 0.0052; Fig. 4e, f). This observation indicated that fatty acids played an important role in tumor cell energy metabolism in MBC patients. To figure out whether this sex-based difference was breast-cancer-specific, we further compared the activity of fatty acid metabolism between male and female patients of other cancer types. Results showed that the fatty acid metabolic pathway was significantly enriched in the up-regulated genes of male patients with lung adenocarcinoma (LUAD), kidney renal papillary cell carcinoma (KIRP), esophageal carcinoma (ESCA), and diffuse large B cell lymphoma (DLBCL; Supplementary Fig. 8e).

Notably, based on the analysis of ChIP-sequencing data for SREBF1 in ER+ breast cancer cell line (MCF-7), LUAD cell line (A549), and ESCA cell line (KYSE150 and TE-5), we found that the promoter of *FASN* was targeted by SREBF1 in these cells (Fig. 4g), further demonstrating that the *FASN*-mediated lipid metabolism was regulated by SREBF1. Besides, the expression levels of *FASN* and *SREBF1* had a significantly positive correlation in both MBC and FBC samples of three independent datasets (Fig. 4h). In addition, inspired by the previous studies that demonstrated the fatty acid metabolism driven by AR in PRAD[24,25], we investigated the association between *AR* and *FASN* expression in MBC and FBC samples (Fig. 4i). Results showed that the expressions of *AR* and *FASN* were positively correlated in MBC samples from GSE104730[6] and GSE31259[20], but had no obvious correlations in the FBC samples of the TCGA dataset.

As our above results showed that cancer cells from MBC patients had higher metastasis-related signature scores, we further explored the correlations between fatty acid metabolism and metastasis in ER$^+$

breast cancers of the TCGA dataset by calculating the Pearson correlation coefficient (PCC). Results showed that the signature score of fatty acid metabolism was positively associated with the metastasis-related programs, including cell migration, EMT, and angiogenesis (Fig. 4j), suggesting the possible involvement of fatty acid metabolism in promoting the metastasis of breast cancer. Previous studies reported that *FASN* played a vital role in breast cancer metastasis and progression[26,27]. The brain metastasis of breast cancer was significantly reliant on the *FASN*-mediated lipid biosynthesis[27], demonstrating that *FASN* could serve as a target for genetic or pharmacological inhibition of breast cancer metastasis. We also analyzed the data of other cancer types of TCGA datasets to evaluate the correlation between fatty acid metabolism and metastasis. Results showed that the positive correlation between fatty acid metabolism and metastasis-related programs was not generally observed in the majority of cancer types but only observed in testicular germ cell tumors (TGCT), BRCA, and uveal melanoma (UVM; Supplementary Fig. 8f).

Moreover, we found that *FASN* expression was negatively correlated with T cell and B cell signature scores in MBC patients of the TCGA dataset (Fig. 4k and Supplementary Fig. 8g), suggesting the potential mediation of low immune infiltration by activated fatty acid metabolism. Thus, we performed a pan-cancer analysis to evaluate the association between *FASN* expression and immune infiltration in TCGA datasets. Results showed that *FASN* expression and tumor purity were positively correlated in most cancers, while the infiltration scores of T cells and B cells were negatively associated with *FASN* expression (Supplementary Fig. 8h). These results implied that the elevated expression of *FASN* may be associated with the immune exclusion.

We performed analyses for overall survival (OS), progression-free interval (PFI), and disease-specific survival (DSS) of TCGA pan-cancer datasets[28] by categorizing the patients into *FASN*-high and *FASN*-low groups for each dataset according to the median of *FASN* expression. Results showed that *FASN* expression was prognostic for the OS, DSS, and PFI of many types of cancers, especially for male cancer patients (Supplementary Figs. 9–11). Male BRCA patients with higher expression of *FASN* had a relatively poor prognosis but were not statistically significant possibly due to that only 12 MBC samples were present in the TCGA. Besides, high expression of *FASN* could predict poor OS and PFI in male patients with bladder urothelial carcinoma (BLCA) and kidney renal clear cell carcinoma (KIRC). The PFI of *FASN*-high male patients with kidney renal papillary cell carcinoma (KIRP) and uveal melanoma (UVM) was also significantly poor. The DSS of lung squamous cell carcinoma (LUSC) male patients with high *FASN* expression was significantly poorer than those with low *FASN* expression. However, the prognosis of female patients with these cancers was not associated with the *FASN* expression. Notably, higher *FASN* expression was prognostic for the poor DSS of PRAD patients, consistent with a previous study that demonstrated that targeting *FASN* could inhibit the aggressive and resistant PRAD[24]. This result suggested that *FASN* may be a potential therapeutic target for male patients with these cancers.

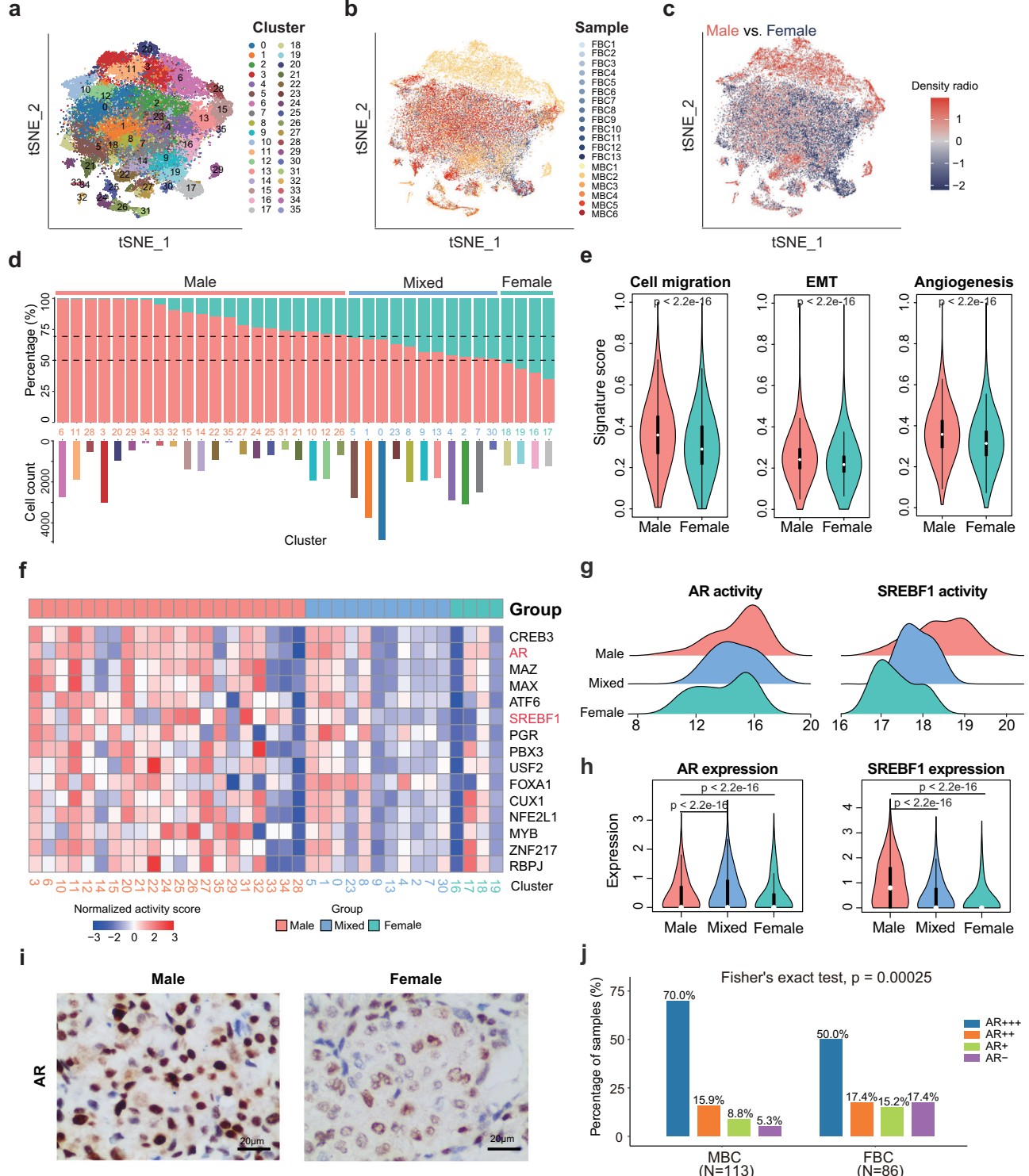

## Different functional characteristics of T cell subpopulations between MBC and FBC

To reveal functional subtypes of the T cell populations in breast cancer, we performed an unsupervised clustering analysis of T cells from MBC and FBC samples. A total of 13 clusters were identified, including seven *CD8*+, four *CD4*+, and two NKT cell clusters (Fig. 5a). Each cluster was defined by the specifically expressed marker genes (Supplementary Fig. 12a and Supplementary Data 6). Accordingly, *CD8*+ T cells were categorized into *GZMK*+T-1, *GZMK*+T-2, *CAPG*+T, *IFIT*+T, *KLRC2*+T, *KRT8*+T, and *TRDV2*+T cells. The *CD4*+T cells were classified as *FOXP3*+T regulatory (Treg) cells, *CXCL13*+T helper (Th) cells,

*CCR7*+naïve, and *FOS*+ naïve T cells. Notably, we found that the *CD8 + KRT8 +* cluster was significantly enriched in MBC samples, whereas the *CD4 + CXCL13 +* Th cells were significantly depleted in MBC samples (Fig. 5b and Supplementary Fig. 12b). Furthermore, the *FGFBP2 +* NKT was enriched in MBC, and *NCAM1 +* NKT was enriched in FBC (Fig. 5b and Supplementary Fig. 12b). These observations suggested the potential differences in T cell functions between MBC and FBC patients.

We further integrated scRNA-seq and scTCR-seq data and compared the enrichment and clone size between MBC and FBC samples (Fig. 5c). We found that the clone sizes of some *CD8 +* T cells, such as

**Fig. 3 | The transcriptional differences of cancer cells between MBC and FBC samples. a** Unsupervised clustering of 53, 343 cancer cells. **b** T-SNE plot of cancer cells colored by samples. **c** The density ratio of the t-SNE projections of cancer cells from male and female samples. The t-SNE visualization is split into 200 × 200 bins. Red represents higher percentage of male cancer cell; Blue represents higher percentage of female cancer cells. **d** Upper: barplot showing the fraction of cancer cells from male and female samples in each cluster. Bottom: barplot showing the number of cells in each cluster. The clusters were ordered by the proportion of cancer cells from male samples. Lightcoral represents male clusters, turquoise represents female clusters and lightblue represents mixed clusters (see "Methods"). **e** Violin plots showing the ssGSEA scores of cell migration, EMT, and angiogenesis of cancer cells from male ($n = 37270$ cells) and female ($n = 16073$ cells) samples. $P$ value was calculated by two-sided Wilcoxon rank-sum test. **f** Heatmap showing the

activity scores of transcription factors (TFs) in cancer cells from male (lightcoral), female (turquoise), and mixed (lightblue) clusters. **g** Ridgeline plot showing the activity levels of MBC-specific TFs in cancer cells from male, female, and mixed clusters. **h** Violin plots showing the expression levels of MBC-specific TFs in cancer cells from male ($n = 21794$ cells), female ($n = 4827$ cells), and mixed ($n = 26722$ cells) clusters. $P$ value was calculated by two-sided Wilcoxon rank-sum test. **i** Representative images of IHC staining detecting AR expression in MBC ($n = 113$) and FBC ($n = 86$) samples Scale bar, 20 μm. **j** Barplot showing the percentage of AR-negative, AR+, AR++, and AR+++ samples from MBC ($n = 113$) and FBC ($n = 86$) ER+ patients. $P$ value was calculated by Fisher's exact test. In **e** and **h**, box plots show median (center line), the upper and lower quantiles (box), and the range of the data (whiskers). Source data are provided as a Source data file.

the *CD8 + CAPG+* and *CD8 + IFIT1+* clusters were remarkably larger in MBC samples than in FBC samples (Fig. 5d). Moreover, the p38 MAPK signature score was significantly higher in *CD8 + T* cells from MBC than in FBC (Fig. 5e), indicating the senescence of *CD8 + T* cells in MBC.

The *CXCL13 + Th* cells that highly expressed *PDCD1* and *CTLA4* had a significant depletion and smaller clone size in MBC samples (Fig. 5b–d and Supplementary Fig. 12a, b). The enrichment of *CXCL13 + Th* cells was previously reported to be associated with the high sensitivity of immunotherapy targeting PD1 or CTLA4 in patients with colorectal cancer[29]. Therefore, our data implied that male patients with breast cancer might be insensitive to the PD1/PDL1 or CTLA4 inhibitors.

Furthermore, we compared the transcriptional patterns and function of *CD4 +*, *CD8 +*, and NKT cells between MBC and FBC samples (Supplementary Data 7–9). Specifically expressed genes and remarkably distinct functions were observed between different sexes (Fig. 5f, g). The mitochondrial pathway of apoptosis was significantly enriched by the up-regulated genes in MBC T cells. Consistent with the above result (Fig. 5e), the p38 alpha/beta MAPK downstream pathway, which was reported to be related to T cell dysfunction and senescence[30], was enriched in MBC T cells. Notably, all three subtypes of T cells in the MBC microenvironment had an activated BDNF signaling pathway, which could enhance lipid oxidation[31]. Lines of evidence have shown that lipid oxidation was one of the most important characteristics of dysfunctional or exhausted T cells[32,33]. We also observed higher expression of *AZGP1* in T cells from MBC. *AZGP1* has been reported to be a key promoter of cancer metastasis and lipid metabolism[34,35]. In contrast, FBC samples exhibited a high expression level of cytotoxic T-cell markers such as *GZMK, KLRB1, KLRD1, XCL1,* and *KLRC1* (Fig. 5f). Also, the specifically expressed genes in FBC T cells were enriched in multiple pathways mediated by immune-modulatory cytokines, such as IL-2, IL-5, IL-4, and TNF-alpha signaling (Fig. 5g). Furthermore, we identified that the fatty acid metabolic pathways were highly activated in MBC T cells, compared with T cells in FBC samples (Supplementary Fig. 12c). Collectively, our data suggested that T cells in the MBC microenvironment were dysfunctional, possibly mediated by the lipid metabolism, whereas T cells in FBC were actively cytotoxic.

### *KRT8*+T cells with high level of fatty acid metabolism were enriched in the MBC microenvironment

According to our comparison analysis, *KRT8 + T* cells were specifically enriched in the MBC samples (Fig. 5b). Moreover, epithelial cell markers, such as *KRT8, KRT18,* and *KRT19,* had significantly higher expressions in MBC T cells (Fig. 5f). We further illustrated the co-expression of *CD3E* and *KRT8* at the single-cell resolution (Fig. 6a) and found that the *CD3E + KRT8 + T* cells tended to enrich in MBC samples (Fig. 6b). To further validate the existence of these cells, we calculated the percentage of *CD3E + KRT8+* T cells of in-house MBC, in-house FBC, and Wu et al.'s FBC samples (Supplementary Fig. 13a, b). Results showed that the percentages of *CD3E + KRT8 + T* cells were similar in in-house and Wu et al.'s FBC samples (Supplementary

Fig. 13c). MBC samples had a significantly higher percentage of *CD3E + KRT8 + T* cells than the FBC samples from the two datasets (Supplementary Fig. 13c). About 50% of T cells from MBC were *KRT8+*, while only 2.1% of T cells from FBC were *KRT8+* (Supplementary Fig. 13d, e). We showed the *KRT8* expression intensity on the t-SNE plot based on sex and whether T cells were *KRT8+*, and found that *KRT8* was expressed on some T cells, especially the T cells from MBC samples (Supplementary Fig. 13f, g).

In order to figure out whether the observed *CD3E + KRT8 + T* cells were patient-specific or generally existed, we evaluated the percentage of *CD3E + KRT8 + T* cells across 19 samples, including 6 in-house MBC samples, 2 in-house FBC samples, and 11 FBC samples from Wu et al. It turned out that 17/19 breast cancer samples had *CD3E + KRT8 + T* cells with different degrees, ranging from 0.2% to 83.1% (Supplementary Fig. 14a). Especially, MBC samples showed higher percentage of *CD3E + KRT8 + T* cell component (6.7–83.1%), and FBC samples had relatively lower percentage (0.2–17.9%). We re-clustered the cells from each sample and then visualized all cell types and marker expression levels at the single-cell level. MBC and FBC samples with the highest percentage of *CD3E + KRT8+* cells were shown in Supplementary Fig. 14b, c. Because only a part of T cells were *KRT8+* (Supplementary Fig. 14d, e), we split the feature-plot into two separate parts to illustrate the expression of *KRT8* more clearly. We found that some T cells did express *KRT8* but others had no expression (Supplementary Fig. 14f, g). Violin plots were used to further statistically compare the *KRT8* expression among epithelial cells, *KRT8 + T* cells, and *KRT8- T* cells, suggesting that *KRT8 + T* cells had a similar or lower level of *KRT8* expression compared with epithelial cells (Supplementary Fig. 14h, i). We also showed the aggregated expression of these markers of epithelial and T cells in each sample using the dot-plot (Supplementary Fig. 14j). The T cells from MBC2, MBC3, MBC4, MBC5, MBC6, and FBC13 had *KRT8/18/19* expression, but were lower than these levels in epithelial cells. Finally, the Wilcoxon rank-sum test showed a significant difference of *CD3E + KRT8 + T* cell enrichment between MBC and FBC groups (Supplementary Fig. 14k; $P$ value: 0.0014).

By evaluating the *CD3E + KRT8 + T* cell percentage under different cell-filtering criteria, we excluded the influence of low-quality cells that would be possibly included during the tissue dissociation, including the doublets or multiplets and broken/dying cells. Considering there may be more expressed genes that could be detected in doublets or multiplets, we limited the number of expressed genes within each single cell using different cutoffs, ranging from 1500 to 5000. Also, dying or broken cells often exhibit extensive mitochondrial contamination. Thus, we calculated the percentage of reads that mapped to the mitochondrial genome in each single cell. Gradient cell-filtering criteria were performed to limit the number of expressed genes and mitochondrial reads percentage. Results showed that the percentage of *CD3E + KRT8 + T* cells did not decline with the screening criteria becoming strict and remained at a robust level in all tests (Supplementary Fig. 15a, b), partially avoiding the technical artifacts caused by low-quality cells. Moreover, we used CellBender[36] to decontaminate

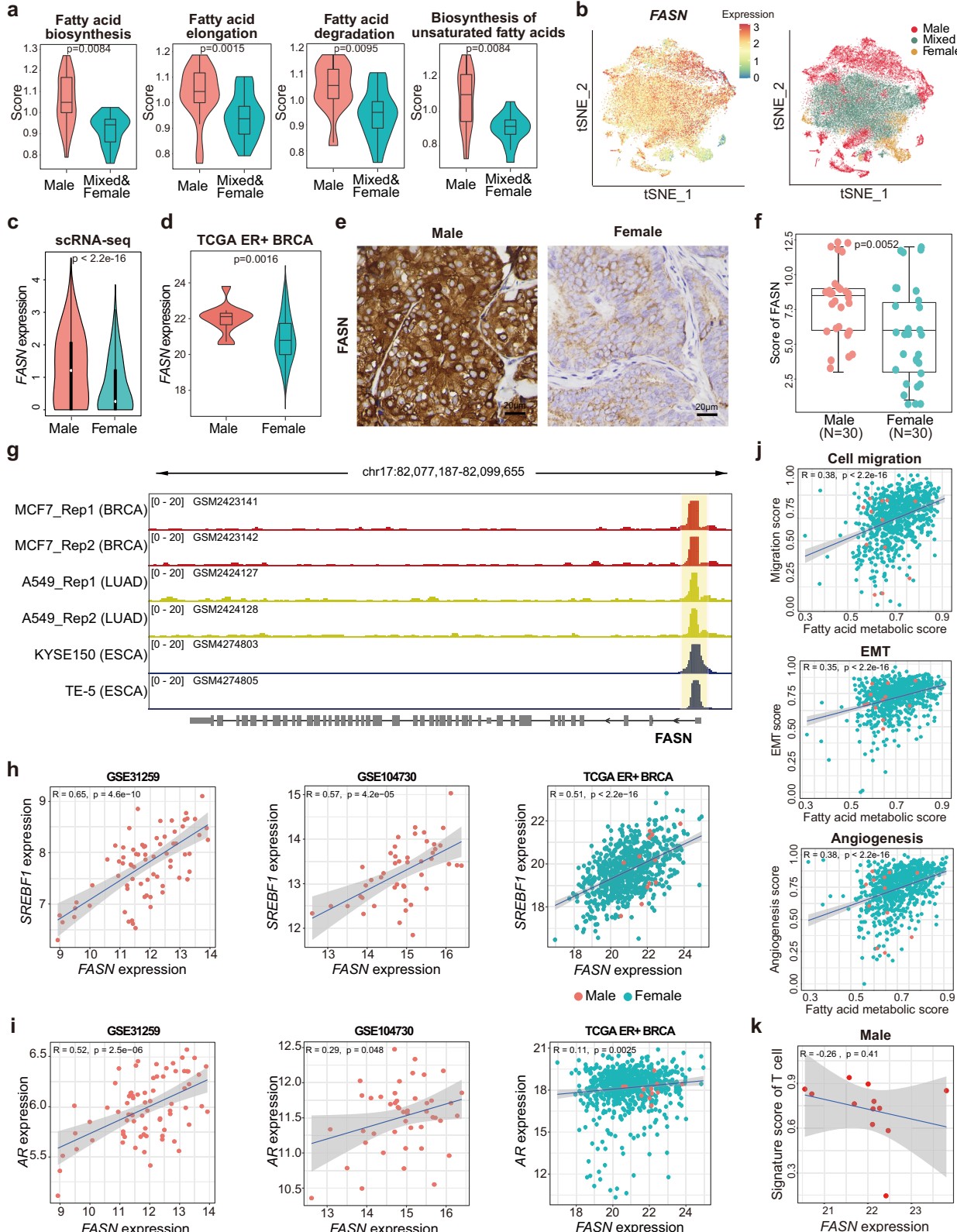

and remove the empty droplets of the in-house scRNA-seq data, of which the raw UMI matrices were available. Also, Scrublet[37] and DoubletFinder[38] were used to identify and remove the doublets in scRNA-seq data. Results showed that *CD3E + KRT8+* cells still existed in all samples after removing the empty droplets and doublets (Supplementary Fig. 15c, d), keeping consistent with the results based on Cell Ranger (Supplementary Fig. 14a). This result double-confirmed the

existence of *CD3E + KRT8+* cells and avoided the potential influence of technical contamination. To further address the concern of cellular stress and dying cell contamination, we performed GSEA analyses using the signature of mitochondria, ribosome, and heat-shock protein for the gene expression profile of T cells. Results showed that the up-regulated genes of *CD3E + KRT8 +* T cells were not enriched in these signatures (Supplementary Fig. 15e).

**Fig. 4 | Identification of the specifically activated metabolic pathways in cancer cells of MBC sample. a** Violin boxplots showing the signature scores of fatty acid metabolic pathways in cancer cells of male (lightcoral) and mixed/female (turquoise) clusters. *P* value was calculated by two-sided Wilcoxon rank-sum test and adjusted for multiple testing using the Benjamini–Hochberg method. **b** Left: t-SNE plot showing the *FASN* expression in cancer cells, color coding for the expression level of *FASN* (blue to red). Right: t-SNE plot of cancer cells colored by Male, Mixed, and Female clusters. **c** Violin plot of *FASN* expression in cancer cells from male (n = 37270 cells) and female (n = 16073 cells) samples. *P* value was calculated by two-sided Wilcoxon rank-sum test. **d** Violin-boxplots showing the *FASN* expression among male (*n* = 12) and female (*n* = 710) samples in TCGA ER⁺ BRCA cohort. *P* value was calculated by two-sided Wilcoxon rank-sum test. **e** Representative images of IHC staining detecting FASN expression in MBC (*n* = 30) and FBC (*n* = 30) samples. Scale bar, 20 μm. **f** Boxplot indicating the IHC score of FASN in MBC (*n* = 30) and FBC (*n* = 30) samples. *P* value was calculated by two-sided Wilcoxon rank-sum test.

**g** IGV plots showing the genomic binding site of SREBF1 on gene *FASN* in various cancer cell lines. The region covered by yellow box represents promoter. **h** The Pearson correlation analysis between the expression of *FASN* and *SREBF1* in independent breast cancer datasets. The color of point represents male (lightcoral) and female (turquoise). **i** The Pearson correlation analysis between the expression of *FASN* and *AR* in independent breast cancer datasets. The color of point represents male (lightcoral) and female (turquoise). **j** The Pearson correlation analysis between the scores of metastasis-related signatures and fatty acid metabolic pathway in TCGA ER⁺ BRCA cohort. The color of point represents male (lightcoral) and female (turquoise). **k** The Pearson correlation analysis between *FASN* expression and the signature score of T cells for MBC samples in TCGA ER⁺ BRCA cohort. In **a**, **c**, **d**, **f**, box plots show median (center line), the upper and lower quantiles (box), and the range of the data (whiskers). In **h**–**k**, 95% confidence interval (CI) is indicated with gray color. Source data are provided as a Source data file.

Further validation using immunofluorescence experiments for the MBC samples confirmed the above observation and showed the existence of CD3 + KRT8+ cells (Fig. 6c and Supplementary Fig. 16a). In order to avoid the artifacts from multiple layers of cells, we further obtained a series of *Z*-stack confocal images of one single CD3 + KRT8+ cell with a confocal microscope (Supplementary Fig. 16b). Besides, we performed flow cytometry experiments for fresh tumor tissues from an MBC patient to validate and quantify CD3 + KRT8+ double-positive T cells (Fig. 6d). Single antibody-labeled compensation samples and fluorescence minus one (FMO) controls were used to determine where the gates should be set (Supplementary Figs. 17 and 18). Firstly, debris was excluded by forward and side scatters gating, and single cells were gated using the FSC-A/FSC-H profile. Dead cells were further excluded using live/dead staining by Zombie. Secondly, KRT8 and CD45 were used to distinguish the epithelial cells (KRT8 + CD45−, 24.0%), immune cells (KRT8-CD45 + , 5.1%), and KRT8 + CD45+ double-positive cells (5.3%). Among the KRT8 + CD45+ double-positive cells, 86.2% were KRT8 + CD45 + CD3 + T cells. Similarly, 87.8% of KRT8-CD45+ immune cells were CD3 + T cells. To better determine the T cell subpopulations, the KRT8 + CD45 + CD3+ and KRT8-CD45 + CD3 + T cells were back-gated and overlaid onto the FSC-A/SSC-A plots. Results showed that both KRT8 + CD45 + CD3+ and KRT8-CD45 + CD3 + T cells were located in the lymphocyte gate. Among all T cells (CD45 + CD3 + ), KRT8+ and KRT8− cells accounted for 50.5% and 49.5% in this MBC sample, respectively. Therefore, these results indicated the biological existence of KRT8 + CD45 + CD3 + T cells.

Furthermore, we attempted to explore the functional implication of these *CD3E + KRT8 +* T cells. Firstly, we asked whether these cells had an enrichment preference in different T cell types. Results showed that these T cells tended to be *CD8*+ (Supplementary Fig. 19a). By comparing the gene expression between *CD3E + KRT8*+ and *CD3E + KRT8*− T cells, we found that *CD3E + KRT8*+ cells down-regulated the cytotoxicity-related genes, such as *GZMA*, *GZMK*, *IFNG*, and *KLRD1* (Fig. 6e and Supplementary Data 10). The important gene for TCR signaling pathways initiation *FYN* was also down-regulated in *CD3E + KRT8 +* T cells[39]. In contrast, genes related to T cell senescence such as *GATA3*, and genes related to histone such as *HIST1H1E* had higher expression levels in *CD3E + KRT8*+ cells. Notably, some genes involved in fatty acid metabolism were significantly up-regulated in *CD3E + KRT8*+ cells, including *FASN*, *HADHA*, *ELOVL5*, and *HACD3* (Fig. 6f). The previous study has shown that the *HADHA* encoded a subunit of the multienzyme complex that catalyzed mitochondrial beta-oxidation of long-chain fatty acids[40]. Furthermore, we also explored the cytotoxic activity of *CD3E + KRT8*+ and *CD3E + KRT8*− T cells and observed less cytotoxic activity in *CD3E + KRT8*+ cells (Supplementary Fig. 19b and Supplementary Data 11).

We found that the *CD3E + KRT8 +* T cells had significantly higher expression levels of genes related to apoptosis induced by the immune response, such as granzyme-A and T cell receptor mediated apoptosis

pathway, but not enriched in the apoptosis related to cellular stress (Supplementary Fig. 17c). Besides, oxidative phosphorylation and the BDNF signaling pathway were significantly activated in the *CD3E + KRT8 +* T cells (Fig. 6g), further confirming the association between lipid metabolism and T cell dysfunction in MBC patients. It is worth noting that AR signaling, proteolysis, and transcription regulation showed the highest enrichment score in the up-regulated genes of *CD3E + KRT8 +* T cells (Fig. 6g). A previous study demonstrated that the transcriptional programs regulated by AR could drive the tumor-infiltrating *CD8 +* T cell exhaustion in male cancer patients, contributing to the sex differences in antitumor immunity[41]. Collectively, our analysis indicated that characterized by the elevated levels of lipid metabolism and *AR* regulation, the *CD3E + KRT8 +* T cells may be involved in the immunological dysfunction in MBC patients.

## The communications between cancer cells and T cells were involved in the immunosuppressive in MBC

We performed the analysis for cell–cell communications among various cell types in MBC and FBC samples to identify the differences in the immunological microenvironment. Results showed that the number of interactions between cancer cells and T cells in MBC samples was approximately twice as many as in FBC samples (Fig. 7a). A majority of T cell subtypes had more interactions with cancer cells in MBC than in FBC samples (Fig. 7b). We further identified the common, male-specific, female-specific ligand–receptor interactions between cancer cells and T cell subpopulations (Fig. 7c, d), indicating both shared and distinct characteristics in MBC and FBC immunology. Notably, interactions of TGF-β and TGF-β receptors were significantly activated in cancer cells and T cells of MBC samples (Fig. 7e, f). Previous studies reported that TGF-β signaling played an important role in T-cell exclusion, immunosuppression, and tumor progression[42–44]. Inhibiting the TGF-β signaling could enhance the immune checkpoint blockade therapy for mammary carcinoma[44]. Besides, the TIGIT-NECTIN2 interaction between T cells and cancer cells was found in MBC samples (Fig. 7e). *TIGIT* (also called T cell immunoreceptor with Ig and ITIM domains) was a key inhibitor of the cancer immunity[45], and TIGIT-NECTIN2 interaction was associated with T cell exhaustion[46]. Also, the immune checkpoint VSIR was expressed on some T cell subpopulations of MBC samples and interacted with the cancer cells via TNF (Fig. 7e). In summary, our results showed that the communications between cancer cells and T cells were involved in the immunosuppressive in MBC samples.

## Discussion

Compared with FBC, MBC is more aggressive and has a worse prognosis[47,48]. Due to the lack of studies on MBC, the current treatment of MBC patients is mainly based on the management of FBC patients. However, sex differences influence a range of biological functions, such as hormonal response, energy metabolism, and immune

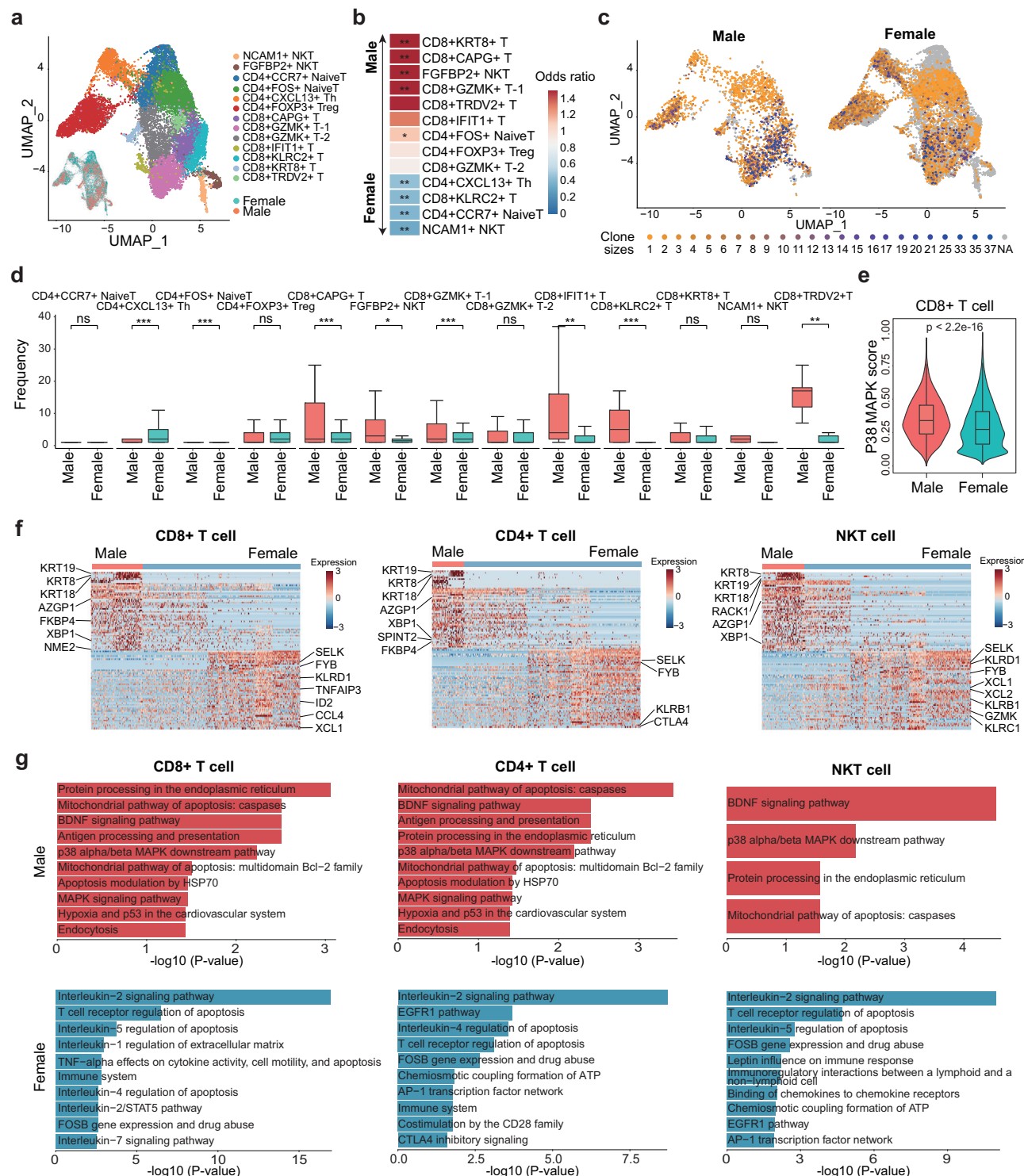

**Fig. 5 | Characterization of subpopulations and clone sizes of T cells in MBC and FBC samples based on scRNA-seq and scTCR-seq. a** UMAP plot showing the subpopulations of T cells. **b** Heatmap showing the odds ratio of each T cell subpopulation calculated by Fisher's exact test. *P < 0.05; **P < 0.01. Red color represents subpopulations enriched in MBC samples, and blue color represents subpopulations enriched in FBC samples. **c** UMAP plot showing the clone sizes of T cells in MBC (left) and FBC (right) samples. **d** Boxplot showing the TCR clone sizes of T cell subpopulations in MBC (n = 3031 cells) and FBC (n = 12,659 cells) samples. Lightcoral represents male and turquoise represents female. *P* value was calculated by two-sided Wilcoxon rank-sum test. *P < 0.05; **P < 0.01; ***P < 0.001; ns: P > 0.05.

**e** Violin plot of p38 MAPK activity in *CD8* + T cells from MBC (n = 1664 cells) and FBC (n = 5248 cells) samples. Lightcoral represents male and turquoise represents female. *P* value was calculated by two-sided Wilcoxon rank-sum test. **f** Heatmap showing the differentially expressed genes between MBC and FBC T cells, including *CD4* +, *CD8* +, and NKT cells. **g** The significant pathways enriched by the differentially expressed genes. Biological pathways in BioPlanet database were used in the enrichment analysis. Red color represents pathways enriched in MBC samples, and blue color represents pathways enriched in FBC samples. In **d**, **e**, box plots show median (center line), the upper and lower quantiles (box), and the range of the data (whiskers). Source data are provided as a Source data file.

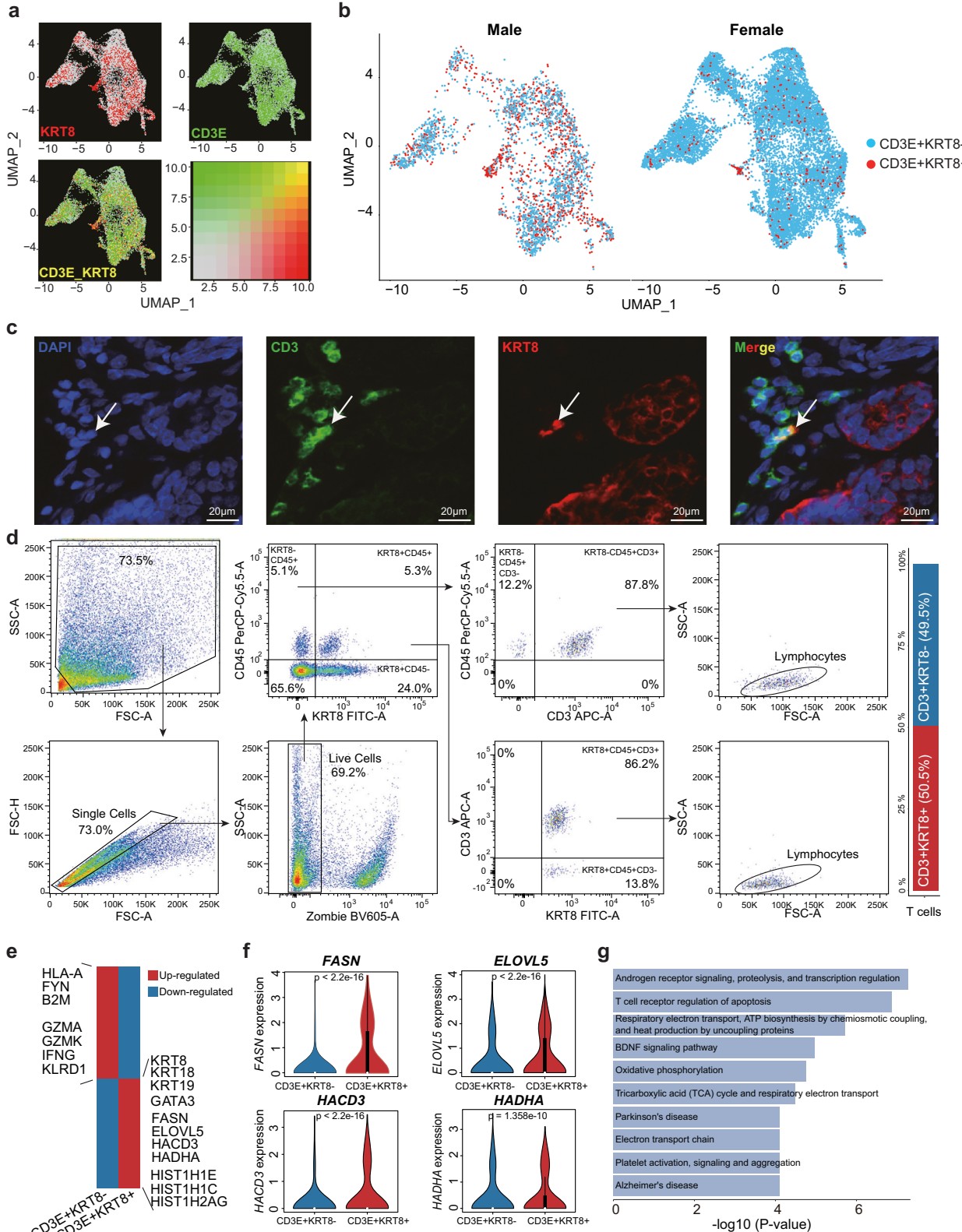

**Fig. 6 | Characteristics of the *CD3E* + *KRT8* + T cells. a** UMAP plot of *KRT8* (gray to red) and *CD3E* (gray to green) expression in T cells. **b** UMAP plot showing the distribution of *CD3E* + *KRT8*- (blue) and *CD3E* + *KRT8*+ (red) T cells in MBC (left) and FBC (right) samples. **c** The immunofluorescence staining of KRT8 and CD3 in an MBC sample. White arrow indicates the CD3 + KRT8 + T cell. Scale bar, 20 µm. Experiments were conducted in triplicate. **d** Full gating strategy of flow cytometry analysis for the identification of KRT8+ and KRT8- T cells in an MBC sample. **e** Heatmap showing the differentially expressed genes between *CD3E* + *KRT8*- and *CD3E* + *KRT8* + T cells. Red represents up-regulated genes and blue represents down-regulated genes. **f** The violin plot of *FASN*, *ELOVL5*, *HACD3* and *HADHA* expression in *CD3E* + *KRT8*− (n = 14085 cells) and *CD3E* + *KRT8* + (n = 1605 cells) T cells. *P* value was calculated by two-sided Wilcoxon rank-sum test. **g** Gene set enrichment analysis of up-regulated genes in *CD3E* + *KRT8* + T cells by BioPlanet database. In **f**, box plots show median (center line), the upper and lower quantiles (box), and the range of the data (whiskers). Source data are provided as a Source data file.

response[8]. Therefore, there is an urgent need for the characterization of cellular and molecular mechanisms of MBC and the identification of novel therapeutic targets to improve the prognosis for MBC patients.

Our study investigated the features of tumor cells and the immune microenvironment in MBC. We performed an integrated single-cell transcriptomic analysis of ER + MBC and FBC patients. Based on the analysis at the single-cell level, we found that MBC patients had a higher tumor purity and a lower degree of immune infiltration than FBC, which were further validated by public RNA-seq data and IHC experiments. Immunosuppression is a primary reason for poor survival and aggressive disease. It has been proven to be an important cause of distal metastasis[49,50], which is associated with cancer growth and progression[51,52]. Furthermore, the transcriptional programs associated with metastasis were markedly activated in MBC patients. Genes associated with cell migration and EMT were significantly activated in MBC, suggesting the high potential of metastasis of MBC.

Differences between males and females can lead to variations in neoplasia characteristics and progression, especially regarding immunological responses and metabolic pathways[7,8]. We found that the fatty acid metabolic pathways were significantly more activated in MBC than in FBC. Tumor cells typically accelerate de novo fatty acid synthesis to provide energy to produce phospholipids and signaling molecules on their cell membranes[53]. Lipid accumulation in tumor-infiltrating myeloid cells can tilt these cells toward immunosuppressive and anti-inflammatory phenotypes through metabolic reprogramming. Li et al. suggested that drugs that target active metabolic pathways in the TME may synergize with immune checkpoint inhibitors by reducing the metabolic stress of tumor-infiltrating lymphocytes (TILs)[54]. Furthermore, the enzyme for fatty acid synthesis FASN also showed a higher level in MBC than in FBC patients. Studies have shown that inhibition of FASN can effectively and extensively inhibit the DNA replication of cancer cells and delay the S phase transition in the cell cycle, suggesting that the pathway of fatty acid synthesis is associated with the growth of cancer cells[55]. A study by Bahlani et al. also reported that FASN was important in regulating the chemosensitivity of different breast cancer types and therefore could serve as a survival factor[56]. FASN inhibitors have been developed to improve the anti-tumor activity against various breast cancers[57]. Moreover, elevated FASN expression was observed in several other cancer types and associated with a poor prognosis of patients[57]. Notably, a previous study demonstrated that lipid metabolism dysregulation driven by FASN upregulation was important in the PRAD progression and castration resistance mediated by AR signaling[24]. Our analysis also indicated the association between FASN expression and poor prognosis in PRAD. These results consistently suggested that FASN-mediated lipid metabolism dysregulation was a potential therapeutic target for hormone-receptor-positive cancers. Notably, the fatty acid metabolism showed a positive correlation with metastasis, and a negative correlation with immune infiltration, implying the activated fatty acid metabolism might involve in the immunological suppression and metastasis of MBC. A previous study demonstrated that FASN could prevent anti-tumor immunity by disrupting tumor-infiltrating dendritic cells[58]. Consistently, our data showed that the increased fatty acid metabolism was associated with the lower infiltration of T cells and B cells in MBC and other cancer types. The up-regulation of pathways associated with lipid oxidation was also observed in MBC T cells. The previous study indicated that lipid peroxidation could promote the dysfunction in CD8 + T cells in tumors[33]. Therefore, we reasonably speculated that targeting the fatty acid metabolism pathway may alleviate the immunosuppressive microenvironment in multiple cancers.

Furthermore, we found that a particular T cells subpopulation that co-expressed KRT8 and CD3 was specifically enriched in MBC samples. Notably, these cells showed significant up-regulation of genes related to fatty acid synthesis and oxidation, such as FASN and HADHA. These T cells had lower cytotoxicity than the CD3E + KRT8- T cells. Compared to CD3E + KRT8- T cells, the up-regulated genes of CD3E + KRT8 + T cells were enriched in AR-regulated programs, consistent with the previous study that demonstrated the sex differences in antitumor immunity driven by AR[41]. With the development of single-cell techniques, we could investigate the cellular characteristics at high resolution and identify the previously unappreciated cells. Intriguingly, a study from Hu et al. identified a non-traditional CD45 + EpCAM+ cell population in the fallopian tube epithelial layer of ovarian cancer patients by scRNA-seq (Smart-Seq2) and validated using immunofluorescence experiments[59]. However, the biological and clinical implications of this population are unclear yet. Our preliminary study validated and characterized CD3E + KRT8 + T cells and implied that these cells may be involved in the immunological dysfunction in MBC patients.

Due to the rarity of MBC occurrence and the stringent sample requirements of single-cell experiments, only limited MBC samples were included in this study, which may have a potential influence on the conclusion of this work. However, this explorative study identified notable differences between MBC and FBC, especially the distinct metabolic and immunological characteristics of MBC patients. These observations need to be further validated with larger sample sizes in the future.

In summary, our study characterizes the immunological and metabolic differences between MBC and FBC at the single-cell level. We indicated that MBC had lower immune infiltration and higher metastasis potential than FBC. The activation of fatty acid metabolism was observed in both tumor cells and T cells in the MBC microenvironment. Moreover, the fatty acid metabolism strongly correlated with metastasis-related programs and the depletion of immune cells in MBC, suggesting that targeting fatty acid metabolism pathways may alleviate the immunosuppressive microenvironment and inhibit cell migration for MBC patients.

## Methods

### Patient samples

Single-cell transcriptomic data from 6 MBC and 13 FBC samples were analyzed, in which eleven FBC samples were collected from a previous study by Wu et al.[60], and other samples were in-house. All of the collected samples were ER+. We defined the ER, PR, HER2, and KI67 status using IHC, and further evaluated the amplification of HER2 based on FISH. The clinicopathological characteristics were shown in Supplementary Data 1. All the collected samples (including MBC and FBC) were negative for HER2 amplification evaluated by FISH. Besides, 18/19 samples were from primary untreated ER+ breast cancers, and FBC8 was from an ER+ female patient treated with neoadjuvant therapy. This study was approved by the Ethics Committee of The First Affiliated Hospital of Nanjing Medical University. Informed consent was obtained from each patient before surgery.

### Cell preparation

Fresh tissue samples were cut into approximately 1 mm³ piece on ice and were transferred into a 1.5 mL tube containing Dulbecco's modified eagle medium (Thermo Fisher Scientific). After mincing with ophthalmic scissors, the tiny tumor pieces were spun down and washed with 1× PBS. The minced tumor tissue from each sample was immediately transferred into a 15 mL tube, and subjected to dissociation using tissue dissociation Kit (Miltenyi Biotec, cat. no. 130-110-203). The suspended cells were subsequently passed through cell strainers with a 70-μm filter and centrifuged at $400 \times g$ for 5 min. After the supernatant was removed, the pelleted cells were suspended in red blood cell lysis buffer (Solarbio) and incubated for 5 min to lyse red blood cells at room temperature (20−22 °C), and then the sample was passed through a 40-μm filter. After washing twice with 1× PBS, the cell

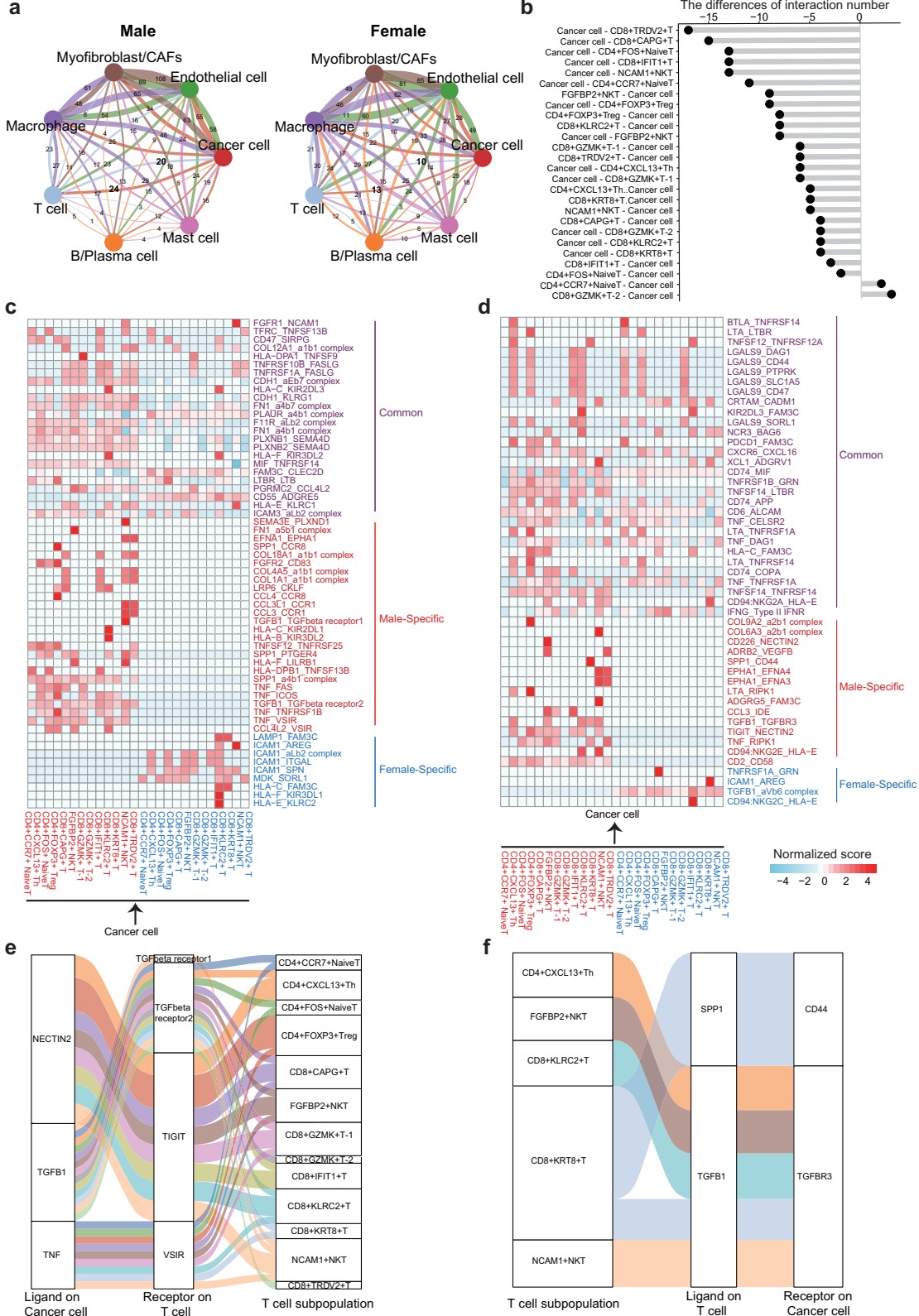

**Fig. 7 | Analysis of cell-cell communications in MBC and FBC samples. a** The number of intercellular communications among different cell types in MBC and FBC samples. The line color represents cell types, and the line thickness represents interaction numbers. **b** The differences of interaction numbers of cancer cells and T cell subtypes between MBC and FBC samples. **c**, **d** Heatmap showing the common (purple), male-specific (red) and female-specific (blue) ligand-receptor pairs in MBC and FBC samples. **e**, **f** Sankey plot showing the representative examples of male-specific ligand-receptor pairs. Source data are provided as a Source data file.

pellets were re-suspended in PBS buffer. Finally, the samples were stained with trypan blue (Sigma) and the cellular viability was evaluated microscopically.

## Library preparation and scRNA-seq

For samples MBC1, MBC2, MBC3, FBC1, and FBC2, 10,000 cells per sample were loaded into a Chromium Single-Cell 3 Chip Kit v2 (10× Genomics, PN-120236) following the established protocols using the Chromium Single Cell 30 Library V2 Kit (10× Genomics, PN-120234). Briefly, reverse transcription, cDNA recovery, cDNA amplification, and library construction were performed using the Single Cell 3' Library and Gel Bead Kit v2 (10× Genomics, PN-120237) and Chromium i7 Multiplex Kit v2 (10× Genomics, PN-120262) according to the manufacturer's instructions. For samples MBC4, MBC5, and MBC6, single-cell suspensions ($1 \times 10^5$ cells/mL) with PBS (HyClone) were loaded into microfluidic devices using the Singleron Matrix Single Cell Processing System (Singleron). Subsequently, the scRNA-seq libraries were constructed according to the protocol of the GEXSCOPE Single Cell RNA Library Kits (Singleron, 5180011)[61]. Individual libraries were diluted to 4 nM and pooled for sequencing. Single-cell library sequencing was performed using the Illumina HiSeq X Ten or NovaSeq 6000, with 150 bp paired-end sequencing.

## ScRNA-seq data pre-processing

The Cell Ranger v3.0.2 pipeline was performed to analyze the raw data and generate gene count data using the default and recommended parameters. The STAR algorithm was used to align the FASTQ output obtained from the sequencing data to the GRCh38 reference genome. Next, gene-barcode matrices were generated for each sample by counting the unique molecular identifiers (UMIs), barcode count, and genes without expression across all cells were removed. Finally, we generated a gene-barcode matrix that contained barcoded cells and gene expression counts. All additional analyses were performed using the Seurat (4.0.4, http://satijalab.org/seurat/) R toolkit[62], including quality control and all subsequent analyses. To eliminate the influence of low-quality cells such as empty droplets and multiplets, cells with expressed genes <200 or >6000 were excluded. The percentage of UMIs mapped to mitochondria was set to less than 25%. Finally, a total of 103,834 cells after quality control were used for further downstream analysis, including 53,028 cells from male samples and 50,806 cells from female samples (Supplementary Data 3).

## Principal component analysis, clustering, and cell-type recognition

We identified the top 2000 variable features using the "vst" method for each dataset. Datasets were then anchored and integrated using the integration procedure from the Seurat package to eliminate the batch effects among the samples. ScaleData function was used to perform a linear scaling transformation on the identified variable features using default parameters. Principal component analysis (PCA) was performed on the scaled data to reduce the dimensionality. The statistical significance of the PCA scores was determined using the JackStraw function. The first 25 principal components were used for identifying the neighbors and clustering the cells with a resolution of 1.5. The cell clusters were visualized using 2D uniform manifold approximation projection (UMAP) or t-distributed stochastic neighbor embedding (tSNE) plots. The FindAllMarkers function was used to identify the genes specifically expressed in each cell cluster. We identified the cell types based on the expression of well-established gene markers. Cells were divided into seven main cell lineages, of which 53,343 were cancer cells.

## Identification of cancer cells

To identify malignant and non-malignant cells, we confidently distinguished malignant from non-malignant cells for each sample using two complementary approaches. First, we identified malignant epithelial cells using the marker genes *EPCAM*, *KRT18*, *KRT14*, and *KRT19*. To verify the identified cancer cells more accurately, we also used the inferCNV R package[63] to evaluate copy number variants (CNVs) levels, using immune cells (T cells, B cells, macrophages, and mast cells) and stromal cells (fibroblasts and endothelial cells) as the control group and epithelial cells as the test group.

## Identification of up-regulated genes of MBC cancer cells

To compare the characteristics of cancer cells from MBC and FBC samples, we integrated cancer cells from 19 samples and identified 36 clusters by unsupervised clustering. Using the FindAllmarkers function of the Seurat package, we identified genes with log2 fold change >0.25 and adjusted *P* value >0.01 for each cluster. Based on the order of log2 fold change, the top 100 genes were further identified as markers of each cluster. By calculating the proportion of cancer cells from MBC samples in each cluster, we defined male, female, and mixed clusters. Specifically, clusters with a proportion of male cancer cells >70% were defined as male clusters, those with a proportion <50% were defined as female clusters, and the others were defined as mixed clusters. To identify the genes specifically expressed in male clusters, gene markers that presented in at least three male clusters were selected, and markers of female or mixed clusters were further removed from this list.

## Transcription factor regulatory activity analysis

We analyzed the regulon activity by using the R package Dorothea (version 1.72)[64], which combined the database of regulons and TF activity inference methods together. Only regulons with confidence levels A, B, and C were selected to better estimate TF activities. Regulon score was calculated for each single cell using VIPER[65], a statistical test based on the average ranks of the targets. We compared the TF activity between male and female clusters and calculate the fold change and *P* values using the two-sided Wilcoxon rank-sum test. MBC-specific TFs were identified with the threshold as follows: fold change >1, expression percentage >30% in MBC cancer cells, and <30% in FBC cancer cells.

## Metabolic pathway analysis

The analysis of the metabolic pathways was performed using the method from Xiao et al.[66]. Single-sample GSEA (ssGSEA) scores were calculated for 85 Kyoto Encyclopedia of Genes and Genomes (KEGG) metabolic pathways based on gene expression levels[67]. The activity difference of KEGG metabolic pathways between male and female cancer cell clusters was measured by two-sided Wilcoxon rank-sum test. *P* values were adjusted for multiple testing using the Benjamini−Hochberg method. Pathways with adjusted *P* value <0.05 were identified as differentially activated pathways between male and female cancer cell clusters.

## Cell−cell communication analysis

To investigate the cell−cell communications mediated by ligand−receptor interaction between different cell types, we used CellPhoneDB[68] to analyze and compare the cell-cell communications between MBC and FBC samples. The ligand-receptor pairs with *P* value < 0.05 were retained.

## Processing of single-cell T cell receptor sequencing data

The TCR was enriched from amplified cDNA from 5' libraries via PCR amplification using the Chromium Single-Cell V(D)J Enrichment kit according to the manufacturer's protocol (10× Genomics). TCR sequences were obtained using the Illumina sequencing platform, and fastq files were generated using the 10× Cell Ranger mkfastq and vdj pipeline.

## Immunohistochemistry and immunofluorescence staining

Immunohistochemistry (IHC) analysis was conducted in tumor tissues from breast cancer patients (Supplementary Data 12). Briefly, paraffin-embedded tissue was sectioned into 4-μm slides and incubated with anti-CD4 (#ab133616, Abcam, 1:500), anti-CD8 (#ab237709, Abcam, 0.25 μg/mL), anti-AR (#5153, CST, 1:500), and anti-FASN (#ab128870, Abcam, 1:450) overnight at 4 °C. Subsequently, slides were incubated with appropriate horseradish peroxidase (HRP)-coupled secondary antibody (1:1000) for one hour at 20–22 °C, followed by incubation with DAB (#SK-4100, Vector Laboratories, Burlingame, CA). The number of positive cells and the total number of cells in the target region of each section were quantified by the ICA Labs-Multiplex IHC V2.2.0 module of Halo V3.0.311.314 analysis software. CD4-positive rates and CD8-positive rates of 30 ER⁺ MBC and 30 ER⁺ FBC samples were then quantified. We also quantified the FASN expression level in the same cohort according to the immunoreactive score (IRS) standard ($P \times I$), where $P$ is the percentage of positive cells, $I$ indicated the staining intensity. $P$ was assessed on scale of 0 to 4 (0: 0–5%, 1: 6–25%, 2: 26–50%, 3: 51–75%, 4: 75–100%); and $I$ was measured on scale of 0 to 3 (0: no staining, 1: light staining intensity, 2: moderate staining intensity, 3: dark staining).

For immunofluorescence staining, slides were washed in PBS containing 1% (v/v) Triton X-100 (PBST) and incubated in blocking buffer consisting of 5% bovine serum albumin for 60 min at room temperature. The samples were incubated with a primary antibody (CD3, #17617-1-AP, Proteintech, 1:500; KRT8, #ab9023, Abcam, 1:200) overnight at 4 °C (Supplementary Data 12). The following day, after washing with PBS, the samples were incubated with a solution containing secondary antibody diluted 1:2000 in PBS for 2 h at room temperature. Z-stack confocal images were obtained using a confocal microscope (CarlZeiss LSM880 with NLO & Airyscan), and an interval of depth between individual pictures was set at 0.71 μm.

## Flow cytometry experiments

Multi-parameter flow cytometry (FCM) was used to determine the expression of KRT8, CD45, and CD3. Fresh tissues (>100 mg) were washed with 1× HBSS (Gibco, 14025092) and cut into small pieces on ice. Digestion was performed for 15–30 min using GEXSCOPE Tissue Dissociation Mix (Singleron, 1200050003) at 37 °C in a shaker. The solution was passed through a 40 μm cell strainer and washed with 1× PBS (Gibco) to obtain single-cell suspensions. At least $2 \times 10^6$ cells were stained with antibodies against human CD3-APC (BD, Clone SK7, 340440), CD45-Percp-Cy5.5 (BD, Clone 2D1, 340953) and KRT8-FITC (Abcam, Clone43, ab176533) as per the manufacturers' instructions for 15 min at 20 °C in the dark (Supplementary Data 12). For intracellular staining, surface-marked cells were fixed for 15 min and then permeabilized using an IntraStain Kit (Dako, DK, K2311) according to the manufacturer's instructions after washing with 1× PBS (Gibco) and centrifuged at $400 \times g$ for 5 min. The samples were kept on ice between sample processing and evaluation using FCM. Flow cytometry was performed using a FACSLyric flow cytometer (BD Biosciences). The intrinsic spectral overlap of the different fluorochromes was corrected using compensation matrices. Due to the scarcity of MBC samples, the experiments of single antibody-labeled compensation controls and FMO controls were performed using ER⁺ FBC samples. The full staining experiments were performed using fresh MBC tumor tissues. Doublets were excluded according to the FSC-A/FSC-H profile. Zombie Yellow Fixable Viability Kit (Biolegend, 423103) was used to exclude the dead cells. All the flow cytometry data were analyzed using FlowJo software (Version 10.8.1, FlowJo LLC). The raw FCS files are deposited in Mendeley Data (https://data.mendeley.com/datasets/wwm9xv56ry/1).

## Bulk transcriptomic data analysis

Bulk transcriptomic data and clinical information from The Cancer Genome Atlas (TCGA) database were downloaded and extracted from the XenaBrowser website https://xenabrowser.net/datapages/. We selected the ER⁺ TCGA-BRCA samples based on the clinical information. Specifically, 835 primary tumor samples with positive breast_carcinoma_estrogen_receptor_status were selected, including both HER2⁺ and HER2⁻ samples. Samples without RNA-seq data were further removed. Finally, we obtained the transcriptomic and clinical data of 722 ER⁺ TCGA-BRCA samples, including 598 ER⁺HER2⁻ FBC, 112 ER⁺HER2⁺ FBC, 9 ER⁺HER2⁻ MBC, and 3 ER⁺HER2⁺ MBC samples. HER2 status is based on the IHC results in the clinical information of the TCGA-BRCA dataset. We also downloaded the bulk transcriptome data of two MBC datasets from the GEO database, GSE104730 (ref. 6) and GSE31259 (ref. 20), for validation. The ESTIMATE R package was utilized to calculate the scores for tumor purity of TCGA samples based on bulk RNA-seq data. We identified the top ten genes with the highest fold-changes of each cell type in our single-cell data and then calculated the ssGSEA scores of these gene signatures for bulk samples. The scores of immune or stromal cells were compared between MBC and FBC samples using two-sided Wilcoxon rank-sum test, which was a non-parametric test that did not assume known distributions[69]. To further validate the reliability of gene signatures derived from the single-cell dataset, we measured the enrichment of TME cells by using immune-deconvolution tools MCP-counter[17], EPIC[18], and xCell[19].

## Evaluating the metastasis-related signature scores for single cells and TCGA samples

Gene markers related to cell migration were obtained from a previous study[70]. EMT and angiogenesis signatures were downloaded from Molecular Signatures Databases (MSigDB). Based on these signatures (Supplementary Data 4), we used ssGSEA to assess the scores of tumor metastasis. The Pearson correlation coefficient between fatty acid metabolism score and metastasis-related signature scores was calculated by the "cor.test" function for TCGA pan-cancer samples.

## Functional enrichment analysis

The gene lists were submitted to Enrichr (https://maayanlab.cloud/Enrichr/) online tool and the top ten terms were retained according to the adjusted p-value.

## Survival analysis

We performed Kaplan–Meier survival analysis of OS, PFI, and DSS for the male and female cancer patients using "Survival" and "Survminer" R packages. The patients were classified as *FASN*_high and *FASN*_low groups for each dataset according to the median of *FASN* expression. The significance was evaluated by the log-rank test.

## Reporting summary

Further information on research design is available in the Nature Portfolio Reporting Summary linked to this article.

# Data availability

The raw single-cell sequencing data generated in this study have been deposited in the Genome Sequence Archive at National Genomics Data Center, China National Center for Bioinformation/Beijing Institute of Genomics, Chinese Academy of Sciences (https://ngdc.cncb.ac.cn/gsa-human, accession no. HRA001341). The processed count matrices data are available at the OMIX (https://ngdc.cncb.ac.cn/omix), accession no. OMIX004533. The flow cytometry data in this study are available in Mendeley Data [https://data.mendeley.com/datasets/wwm9xv56ry/1]. The publicly available scRNA-seq data of 11 FBC samples are available in the Gene Expression Omnibus (GEO) under accession number GSE176078. The publicly available bulk transcriptome data of MBC samples are available in the GEO under accession numbers GSE104730 and GSE31259. All R packages are available online, as described in the

"Methods." All other data are available in the article and its Supplementary Files or from the corresponding author upon request. Source data are provided with this paper.

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

## Acknowledgements

The authors thank the First Affiliated Hospital of Nanjing Medical University for providing histological technology and analysis. We thank Oncocare Life Technology Co., Ltd for the computational assistance. This study was supported by the National Natural Science Foundation of China Grant Nos. 81972486 (Q.D.), 81972358 (Q.W.), 91959113 (Q.W.), 32200590 (K.L.); Key Research & Development Program of Jiangsu Province Grant No. BE2017733 (Q.W.), Basic Research Program of Jiangsu Province Grant No. BK20180036 (Q.W.), Natural Science Foundation of Jiangsu Province Grant No. BK20210530 (K.L.), and China Postdoctoral Science Foundation Grant No. 2021M691645 (K.L.), 2022T150329 (K.L.).

## Author contributions

Q.D., Q.W., and K.L. conceived the concept and supervised the study. K.L., L.Z., M.Z., Y.C., Lu Li, J.T., B.H., B.B., N.L., L.W., W.W., Liangyu Li, Y.L., Lin Luo, and Q.L. integrated and analyzed the data. H. Sun, Z.W., D.G., Y.Z., J.S., L.S., T.X., C.Y., Q.X., Xue Han, W.Z., J.L., D.M., H. Shao, X. Zheng, S.L., H.P., J.K., W.J., X. Zhang, Xuedong Han, J.C., H.A., J.G., C.P., and X.X.W. performed the experiments. K.L. and L.Z. wrote the original draft. Q.D., Q.W., and K.L. reviewed the manuscript. All authors contributed to the research and approved the manuscript.

## Competing interests

The authors declare no competing interests.

## Additional information

[1]Jiangsu Breast Disease Center, The First Affiliated Hospital with Nanjing Medical University, 300 Guangzhou Road, 210029 Nanjing, China. [2]Department of Bioinformatics, Nanjing Medical University, 101 Longmian Avenue, 211166 Nanjing, China. [3]Collaborative Innovation Center for Personalized Cancer Medicine, Jiangsu Key Lab of Cancer Biomarkers, Prevention and Treatment, Nanjing Medical University, 211166 Nanjing, Jiangsu, China. [4]The Affiliated Cancer Hospital of Nanjing Medical University, Jiangsu Cancer Hospital, Jiangsu Institute of Cancer Research, 210002 Nanjing, China. [5]Department of Breast Surgery, The Second People's Hospital of Lianyungang, 41 Hailian East Road, 222006 Lianyungang, China. [6]National Health Commission Key Laboratory of Antibody Techniques, Department of Cell Biology, Jiangsu Provincial Key Laboratory of Human Functional Genomics, School of Basic Medical Sciences, Nanjing Medical University, 211166 Nanjing, Jiangsu, China. [7]Department of Breast Surgery, The First Affiliated Hospital of Soochow University, 188 Shizi Street, 215006 Suzhou, China. [8]Department of Pathology, The First Affiliated Hospital with Nanjing Medical University, 300 Guangzhou Road, 210029 Nanjing, China. [9]Department of Breast Surgery, Affiliated Hospital of Jiangnan University, 1000 Hefeng Road, 214000 Wuxi, China. [10]Department of Breast Surgery, Affiliated Suqian Hospital of Xuzhou Medical University, 138 Huanghe South Road, 223800 Suqian, China. [11]The Affiliated Lianyungang Hospital of Xuzhou Medical University, 6 Zhenhua East Road, 222006 Lianyungang, China. [12]Liyang People's Hospital, 70 Jianshe West Road, 213300 Liyang, China. [13]The Affiliated Hospital of Nantong University, 20 Xisi Road, 226300 Nantong, China. [14]Department of Breast Surgery, The Third Affiliated Hospital of Soochow University, 185 Juqian Street, 213000 Changzhou, China. [15]Department of Breast Surgery, The Affiliated Changzhou Second People's Hospital of Nanjing Medical University, 29 Xinglong Lane, 213000 Changzhou, China. [16]Department of Breast and Thyroid Surgery, Huai'an First People's Hospital, Nanjing Medical University, 1 Huanghe West Road, 223300 Huai'an, China. [17]Department of General Surgery, the First People's Hospital of Yancheng, 66 Renmin South Road, 224001 Yancheng, China. [18]Department of Pathology, The Second People's Hospital of Lianyungang, 41 Hailian East Road, 222006 Lianyungang, China. [19]Department of Breast Surgery, the Second Affiliated Hospital, Zhejiang University, College of Medicine, 88 Jiefang Road, 310009 Hangzhou, China. [20]Institute for Brain Tumors, Jiangsu Key Lab of Cancer Biomarkers, Prevention and Treatment, Collaborative Innovation Center for Personalized Cancer Medicine, Nanjing Medical University, Nanjing, Jiangsu, China. [21]Department of Medicine, Division of Regenerative Medicine, University of California, San Diego, La Jolla, CA, USA. [22]Biomedical Big Data Center, Nanjing Medical University, 211166 Nanjing, Jiangsu, China. [23]These authors contributed equally: Handong Sun, Lishen Zhang, Zhonglin Wang, Danling Gu. ✉e-mail: likening@njmu.edu.cn; wangqh@njmu.edu.cn; dingqiang@njmu.edu.cn

