## [Peer Review File · Nature Communications]

Single-cell transcriptome analysis unveils fatty acid metabolism-mediated metastasis and immunosuppression in male breast cancerEditorial Note:

Parts of this Peer Review File have been redacted as indicated to remove third-party material where no permission to publish could be obtained.

Reviewers' comments:**Reviewer #1 (Remarks to the Author):**

The manuscript "Single-cell transcriptome analysis reveals fatty acid metabolism mediated metastasis and immunosuppression in male breast cancer" is an interesting effort to characterize the differences between breast cancer in men and women, which is an understudied topic. The Authors performed single-cell analysis to identify notable differences in the two groups, such as ESR1 and AR activity and fatty acid metabolism mainly mediated by FASN expression. Moreover, the Authors describe higher levels of tumor purity, cell cycle genes and pathways related to tumor invasiveness, as well as lower immune infiltration in male breast cancer compared to samples from women, findings in line with bulk tumor samples in the TCGA. The Authors also describe the presence of immune-epithelial cells, and the potential relationship between fatty acid metabolism and metastasis-related programs. A prognostic role of these features is also suggested, using bulk data from the TCGA, as well as a potential therapeutic role for FASN inhibition as demonstrated by previous works. The relationship between fatty acid synthesis, immunosuppression and tumor progression is intriguing, but may require some clarification.

The following may need to be addressed:

- As single-cell experiments were performed in 3 male and 2 post-menopausal female breast cancer samples, the explorative nature of the findings should be addressed throughout the manuscript.
- Clinicopathological characteristics such as tumor stage may be relevant when comparing the male and female samples, to ensure that differences observed were not related to differences in the staging (e.g. larger and more advanced tumors may be associated with immune exhaustion). As per supplementary table 1, tumor stage was not available for

female breast cancer samples. The Authors should specify this in the Methods section (lines 406-411). Whether samples are from primary, untreated breast cancers should also be specified.

- Supplementary Table 1 and lines 406-411: Were ER/PR/HER2 positivity defined as per ASCO/CAP guidelines? How was AR positivity defined? What do the Authors mean by “Molecular classification”? Is it based on PAM50 subtyping or defined by IHC? In the latter case, “IHC classification” may be more appropriate. Furthermore, while it is true that all samples were ER-positive, Authors should also mention the presence of HER2-positive samples (1/3 in the male cohort and 2/2 in the female cohort), as biology of HER2-negative and positive tumors is substantially different. Was HER2 status (or ERBB2 expression levels) considered by the Authors in the comparative analysis?

- The Authors may want to specify in the Methods section how correlations were computed (Pearson? Spearman?). Moreover, whether P values are adjusted for multiple testing or not should also be specified.

- Line 176: “up-regulated” may be replaced with “upregulation”.

- Lines 219-226: The Authors describe a positive correlation between FASN expression and tumor purity, and a negative correlation not only with immune cells but also with CAFs and endothelial cells. The Authors then suggest that elevated expression of FASN may promote immune escape. Since higher tumor purity is also necessarily associated with lower stroma content (including immune cells), the cause-effect relationship suggested by the Authors may not be necessarily proved by these findings. This section may need to be adjusted accordingly.

- Similarly, the positive correlation with metastasis-related pathways (e.g. suggested in line 374-376) may not necessarily mean “causation”, as many other factors could play a role.

- Line 270 – Paragraph “MBC-specific T cells that co-expressed epithelial and immune markers were in the apoptosis stage”: here, the Authors describe the presence of T cells

showing both T and epithelial cell markers, suggesting the existence of “epithelial-T cells”. The Authors tried to exclude that this finding were to be related to technical artifacts in the single-cell analysis, and showed the coexistence of CD3E and KRT8 markers with immunofluorescence experiments.

Although intriguing, further validation in other available single cell datasets (e.g. doi: 10.1038/s41588-021-00911-1) is in my opinion warranted, as it would give more robustness to this finding.

Moreover, the section included in lines 278-284 may need to be explained in a clearer way. Indeed, as per Supplementary Figure 6, panel A (Differentially expressed genes and functional analysis of T cells between male and female patients), and as suggested by the Authors (lines 274-278), it seems that KRT genes are expressed only in T cells from sample M3 (and maybe to a lesser extent M1). Confirming these findings in other single-cell datasets would be useful to exclude that the presence of epithelial-T cells are patient-specific, as one may argue that they may not be specific of male breast cancer.

Moreover, the presence of several genes related to dissociation or cellular stress (e.g. mitochondrial gene, FOSB, JUNB, heat-shock protein genes) and ribosomal genes, may raise a concern regarding contamination at the droplet level by dying cells (that would go in the same direction of what is stated in lines 308-313, when the Authors mention the high expression of apoptosis-related genes in this cell type). In this regard, can the Authors exclude that the finding of epithelial-T cells is not related to a potential issue of contamination? Indeed, in case of contamination, filtering by the number of genes may not be enough to identify technical artifacts.

With regards to immunofluorescence (Figure 5 B), can the Authors quantify the number of cells with co-expression of the KRT8 and CD3E marker?

- Lines 504-509: Which statistical method was used to compare KEGG metabolic pathways in male and female clusters?

- Line 558: To ensure reproducibility, which criteria were used for the selection of ER-positive samples in the TCGA? Since some samples in the single-cell cohorts were HER2-positive according to Supplementary Table 1, why was this group excluded from the TCGA analysis?

- Lines 567-571: To further validate the reliability of gene sets derived from the single-cell dataset, these findings may be compared to those derived from available immune-deconvolution tools (e.g. MCP-counter, EPIC, TIMER, xCell...) including the cell types of interest (e.g. in terms of correlations).
- Line 229: From Supplementary Figure 4, it seems that FASN high is significantly ($P = 0.04$) associated with OS in female breast cancer, and not in males ($P = 0.27$), although only 12 male breast cancer samples were present in the TCGA. However, in the text, it is stated that “high expression of FASN could predict poor OS of male patients with BRCA”. This part may need to be rephrased. Related to the comment on lines 588-595, other survival end-points, especially in the breast cancer, may be more informative than OS for evaluating the prognostic value of FASN expression.
- Lines 588-595, Survival analysis: Since overall survival data has to be interpreted carefully in the TCGA, especially for luminal breast cancer (doi:10.1016/j.cell.2018.02.052), did the authors tested also other survival end-points (PFI, DFI)?
- Line 247: For the SingleR tool analysis, which reference dataset was used? From Supplementary Fig. 5 it seems that some cell types are relatively “mixed” together and not well defined in the t-SNE. Did the Authors double checked manually if the automatic annotations were reliable?
- Line 433 and 437: The Cell Ranger versions mentioned are discordant (v2.1.0 and 3.0.2). Is this correct?
- Line 449: Please clarify if cells with more or less than 2000 expressed genes were retained.
- Lines 454-455: Please rephrase specifying in a clearer way if UMI count and MT genes were used as regression terms in the ScaleData function(also, Authors may replace “ScaleDate” with “ScaleData” in the text).
- Line 465: Please specify if the default parameters were used in the IntegrateData function.

Was default integration from Seurat applied from the beginning on all cells, or just for specific cell types?

- Lines 475-480: Please explain in a clearer way this section. What did the “normal cell cluster” used in inferCNV included (e.g. normal epithelial breast cells, stromal cells, immune cells...)? Was it formed by “any other cell” that was not tagged as malignant?

- Line 492: Are P values adjusted for multiple testing or not? This should be stated in the method sections for the other analyses as well.

- Line 509: Which statistical test was used to perform this comparison of metabolic pathways between male and female clusters?

- Figure 3, panel G: Can the Authors add a value for the correlations showed?

- Figure 3, panel J: The difference between the groups of comparisons (cell types and FASN high/low cells) is not clear and may be specified in the Figure legend. Are the 4 main columns representing interactions with opposite directions?

Reviewer #2 (Remarks to the Author):

The is a well written and comprehensive manuscript describing the immune and metabolic landscape of male breast cancer.

The premise of this paper that male and female breast cancers are immunological and metabolically different is very compelling and may potentially provide new insights into therapeutic strategies. The investigators have carefully evaluated a broad range of proliferation, angiogenesis, and metabolic pathways as well as detailed immune characterization. The study includes a limited number (3 and 2) reference cases. The study is expanded by data from the TCGA.

Strength of the study include the clearly distinctive patterns that the evaluated male and female breast cancers. The single cell sequencing is elegantly done, and the figures are beautifully outlined and clearly delineated.

A major concern of the study is that the female breast cancers neither have ER expression (ESR1) nor ER activity. Male breast cancer is mostly ER+, whereas female breast cancer has a broad diversity ranging from triple negative disease to ER+ and HER2 positive disease. The immune landscape, EMT, angiogenesis is vastly different in these subtypes. Particularly, TNBC stand out in their immune profile. The data would be very much strengthened if the authors provided data on ER+ female breast cancer, to show how this is similar or different from an ER+ male breast cancer.

Furthermore, a more in-depth explanation on the significance of the findings. The error bars appear very wide in a large number of examples. How are the p-values adjusted for significance in this multi-parameter assessment?

TCGA data while compelling is not novel and may not provide sufficient annotations to clinical

Reviewer #3 (Remarks to the Author):

Male breast cancer (MBC) is associated with worse prognosis compared to female breast cancer and the cellular and molecular differences between the two remain unclear. The researchers used single-cell RNA (scRNA) sequencing and T cell receptor (scTCR) sequencing characterize the tumor microenvironment of MBC. They sequenced three MBC and two post-menopausal ER+ female breast cancers (FBC) and show evidence that MBC have lower immune infiltration, activated ER and AR regulons, higher fatty acid synthase (FASN) expression, and exhausted CD8 T cells. The authors identify a subset of T-cells that express epithelial cytokeratins. However, the manuscript is lacking good quality evidence for the existence of these epithelial-T cells. The authors should consider removing that entire section or provide additional experiments to validate their findings. Androgens have long been known to drive fatty acid synthase PMID: 9067276, and the authors show good evidence of AR regulon activation in MBC, perhaps more focus on the androgen receptor

would tie this story together. Overall, the study is of interest, but more experiments and analysis are needed for this study.

Specific comments

1. While two of the three MBC samples have low immune infiltrate, one actually has similar levels to the two other FBC samples (Figure 1e). Therefore, one cannot conclude that there are less immune cells in MBC, as this may just be a sampling artefact.

2. Please supply raw p-value and statistical test used in Fig.1g. There are only 12 male samples compared to 1085 female samples in the TCGA, therefore one likely cannot assume the MBC will represent a normal distribution unless proven.

3. Statistical test for Figure 1i needed in figure legend.

4. Representative IHC for foxp3 positive staining appears to be nonspecifically stain tumor cells (Figure 1h). The investigators perhaps should perform dual IF to demonstrate the FOXP3 staining is confined to Treg cells (CD4+). The details of the cohort in Figure 1 needs to be in the figure legend or text.

5. What does IHC look like for FASN and AR in this cohort from Figure 1h?

6. A hallmark of prostate cancer progression is dysregulation of lipid metabolism via overexpression of fatty acid synthase (FASN), a key enzyme in de novo fatty acid synthesis. Why was prostate cancer (PRAD) left out of the survival analysis stratified by FASN levels? Please include citation and discussion of targeting FASN in prostate cancer (PMID: 30578319).

7. Supplementary Fig. 2 legend description inadequate. What fold change and significance and testing performed?

8. Supplementary Fig. 4 legend needs more detail. How were FASN high and low cutoffs determined?

9. The fact that FASN and the ER- and AR-response genesets were significantly enriched by the up-regulated genes of “epithelial-T” co-expression cells, suggests that there may be mixing of epithelial and T cell RNA in these dual positive cells. Therefore, additional experiments are needed for the existence of “epithelial-T cells”. The authors provide dual immunofluorescence (IF), however the staining in Figure 5B is unconvincing. The legend states the scale bar is 50uM, but there is no scale bar and thus hard to interpret. It is not clear whether the staining is from a single mitotic cell or many cells at a distance. The DAPI does not even show uniform nuclear localization. The staining appears to be an artifact. The researchers need to show additional validation of the for IF using positive and negative control tissues. In addition, the investigators need to quantify the CD3 only and epithelial T cells for the IF. The authors should also provide another independent method to support their findings such as flow cytometry (KRT and CD3) of dissociated T cells from fresh tumor tissue if possible.

10. Supplemental Fig. S6a is described as differentially expressed gene across five samples. What are the individual values? Aggregated expression of all the single cells for each tumor? Perhaps showing the expression of KRT8/18/19 and CD3 across all cells annotated by cell type for each tumor would be more convincing for the existence of an epithelial T-cell. This will show the relative KRT levels in true epithelial cells relative to the T cells.

11. Supplemental Fig. 6b shows the percentage of T cells that express KRT (epithelial-T cells) is around 40%, and similar in Fig 5C, however in Fig 5A there it appears that nearly all cells co-expressed CD3 and KRTs. What are the proportions of epithelial T cells in the other MBCs and FBCs or is this just an occurrence in the M3 tumor?

12. The authors should consider evaluating several other scRNA breast cancer datasets for evidence of epithelial T cells.

13. Data availability section is weak, and data are not publicly deposited (this can be blinded

until publication but available for reviewers).

14. The authors should consider evaluating the role of AR in MBC in more detail. Such as performing IHC on specimens, evaluating the RNA-seq for existence to alternative splicing in the androgen receptor.

15. The authors need more detail in most figure legends. It is sometimes hard to interpret the data. For example, Figure 2g and h show expression and activation of transcription factors, but what cell types were evaluated (just epithelial)? There appears to be a bimodal distribution in these blots suggesting there the cells are either in an on or off state. It would be interesting to see what cells are on vs. off.

Reviewers' comments:

**Reviewer #1**

The manuscript “Single-cell transcriptome analysis reveals fatty acid metabolism
mediated metastasis and immunosuppression in male breast cancer” is an interesting
effort to characterize the differences between breast cancer in men and women, which
is an understudied topic. The Authors performed single-cell analysis to identify notable
differences in the two groups, such as ESR1 and AR activity and fatty acid metabolism
mainly mediated by FASN expression. Moreover, the Authors describe higher levels of
tumor purity, cell cycle genes and pathways related to tumor invasiveness, as well as
lower immune infiltration in male breast cancer compared to samples from women,
findings in line with bulk tumor samples in the TCGA. The Authors also describe the
presence of immune-epithelial cells, and the potential relationship between fatty acid
metabolism and metastasis-related programs. A prognostic role of these features is also
suggested, using bulk data from the TCGA, as well as a potential therapeutic role for
FASN inhibition as demonstrated by previous works. The relationship between fatty
acid synthesis, immunosuppression and tumor progression is intriguing, but may
require some clarification.

The following may need to be addressed:

1. As single-cell experiments were performed in 3 male and 2 post-menopausal female
breast cancer samples, the explorative nature of the findings should be addressed
throughout the manuscript.

**Response:** Thank you for your comments. In order to further support and validate the
conclusion in this study, we expand the sample size of both male and female breast
cancer. In this revised version, six MBC and thirteen FBC samples were included, in
which eleven FBC samples were from a previous study by Wu et al. (Nature genetics,
2021, 53(9): 1334-1347. doi: 10.1038/s41588-021-00911-1) and other samples were in-
house. All of the collected samples were ER⁺. The transcriptome of 58,578 and 52,460
single-cells was sequenced in MBC and FBC, respectively (**Response Figure 1**). By
performing the same analysis procedure using this updated dataset, we found that the
main results were consistent with the previous version, and demonstrated the followings:
(1) scRNA-seq, bulk transcriptome, and immunohistochemistry consistently
demonstrated that MBC had a significantly lower degree of T cell infiltration than FBC;
(2) metastasis-related programs such as cell migration, epithelial-mesenchymal
transition (EMT), and angiogenesis were more active in cancer cells from MBC than
FBC; (3) the activated fatty acid metabolism involved by FASN was related to the
cancer cell metastasis and low immune infiltration of MBC; (4) different characteristics

of T cell subpopulations between MBC and FBC were identified. T cells in MBC
showed activation of p38 MAPK and lipid oxidation pathways, indicating the
dysfunctional state. In contrast, T cells in FBC exhibited a higher expression level of
cytotoxic markers such as GZMK and KLRB1, and activated pathways mediated by
immune-modulatory cytokines; (5) the inhibitory interactions between cancer cells and
T cells in the MBC microenvironment were identified, such as cell-cell
communications mediated by TGF- β , TIGIT, and VSIR. (6) KRT⁺ T cells with high
level of fatty acid metabolism were enriched in the MBC microenvironment. These
observations were further validated in bulk-RNAseq data and molecular experiments.

Despite the rarity of MBC occurrence and the stringent sample requirements of
single-cell experiments, we had collected and sequenced six MBC samples as possible
as we can. As far as we know, this study is the first to characterize the differences
between MBC and FBC at the single-cell resolution. Benefiting from the enlarged
sample size (6 MBC vs. 13 FBC), we could statistically evaluate the significance of the
observed differences between MBC and FBC samples. On the other hand, we also
discussed the explorative nature of this preliminary study in the revised manuscript as
follows (**Lines 529-533**): “Due to the rarity of MBC occurrence and the stringent
sample requirements of single-cell experiments, only limited MBC samples were
included in this study. However, this explorative study identified notable differences
between MBC and FBC, especially the distinct metabolic and immunological
characteristics of MBC patients. These observations need to be further validated with
larger sample sizes in the future.”

[FIGURE REDACTED]

**Response Figure 1 (Related to Figure 1a in revised manuscript). Schematic workflow for data**
**collection and single-cell analysis in this study.**

2. Clinicopathological characteristics such as tumor stage may be relevant when
comparing the male and female samples, to ensure that differences observed were not
related to differences in the staging (e.g. larger and more advanced tumors may be
associated with immune exhaustion). As per supplementary table 1, tumor stage was
not available for female breast cancer samples. The Authors should specify this in the
Methods section (lines 406-411). Whether samples are from primary, untreated breast
cancers should also be specified.

**Response:** Thank you for your professional suggestions. We added the tumor size and
TNM staging of each sample in the revised supplementary table 1. Besides, we
compared the clinical characteristics of the collected MBC and FBC samples. Results

showed that there were no significant differences in age, HER2 status, KI67 level, and
 extent of the tumor (T) between the FBC and MBC groups (**Response table 1**),
 avoiding the influence of these factors on the comparison. Due to the absence of tumor
 size and metastasis information of samples from Wu et al.'s study, only the categories
 of the tumor extent (T1 ~ T4) were compared between the two groups. Continuous
 variables, including age and Ki67 level, were compared using 2-sided Mann-Whitney
 U test. Categorical variables, including HER2 status and tumor extent, were compared
 using Fisher's exact test. We added these comparison results in the revised manuscript
 as follows (**Lines 126-132**): "Considering some clinicopathological characteristics such
 as tumor stage may be associated with the immune microenvironment and metabolism
 of patients, we compared the clinical characteristics of the collected MBC and FBC
 samples. Results showed that there were no significant differences in age, HER2 status,
 KI67 level, and extent of the tumor (T1 ~ T4) between the FBC and MBC groups
 (**Supplementary Table 2**), avoiding the influence of these factors on the comparison."

**Response table 1 (Related to Supplementary table 2 in the revised manuscript).**
 **Comparison of the clinical characteristics of the collected MBC and FBC samples**

	male(n=6)	female(n=13)	p-value
Age, Median(IQR)*	63.5(54.8-73)	55(52-67)	0.4293
HER2			
	2+ (n=1)	2+ (n=5)	
	+ (n=0)	+ (n=4)	0.1625
	- (n=5)	- (n=4)	
Ki67 (%) , median(IQR)*	25(12.5-30)	15(10-50)	0.9293
T Stage			
	T1 (n=2)	T1 (n=2)	
	T2 (n=3)	T2 (n=5)	0.7007
	T3 (n=1)	T3 (n=5)	
	T4 (n=0)	T4 (n=1)	

**IQR: interquartile range*

Moreover, 18/19 samples were from primary untreated ER+ breast cancers, and
 FBC8 was from an ER+ female patient treated with neoadjuvant chemotherapy
 (**supplementary table 1**). We added the corresponding description in the revised
 Method section as follows (**Lines 550-552**): "Besides, 18/19 samples were from
 primary untreated ER+ breast cancers, and FBC8 was from an ER+ female patient
 treated with neoadjuvant therapy".

3. Supplementary Table 1 and lines 406-411: Were ER/PR/HER2 positivity defined as
102 per ASCO/CAP guidelines? How was AR positivity defined? What do the Authors
mean by “Molecular classification”? Is it based on PAM50 subtyping or defined by IHC?
In the latter case, “IHC classification” may be more appropriate. Furthermore, while it
is true that all samples were ER-positive, Authors should also mention the presence of
HER2-positive samples (1/3 in the male cohort and 2/2 in the female cohort), as biology
of HER2-negative and positive tumors is substantially different. Was HER2 status (or
ERBB2 expression levels) considered by the Authors in the comparative analysis?

**Response:** The ER/PR/HER2 positivity was defined according to the ASCO guidelines.
We defined the ER, PR, HER2, and KI67 status using IHC, and further evaluated the
amplification of HER2 based on FISH. The updated information was shown in the
revised supplementary table 1. All the collected samples (including MBC and FBC)
were negative for HER2 amplification evaluated by FISH (supplementary table 1). In
order to figure out whether the HER2 status evaluated by IHC was related to the
observation in this study, we further compared the immune infiltration, FASN
expression, and metastasis signature scores among groups of ER⁺HER2⁺ MBC,
ER⁺HER2⁻ MBC, ER⁺HER2⁺ FBC, and ER⁺HER2⁻ FBC samples. The following
results were found: (1) Both the scRNA-seq data and TCGA-BRCA data consistently
showed that the ER⁺HER2⁻ MBC samples had the highest level of cancer cell
enrichment and significantly lower level of T cell and B cell percentage (**Response**
**Figure 2**). Besides, it seemed that the T and B cell percentages were higher in
ER⁺HER2⁺ MBC than in ER⁺HER2⁻ MBC samples, although further evaluation was
needed in a larger cohort. (2) Cancer cells from MBC samples showed higher
expression of FASN than FBC samples, independent of the HER2 status (**Response**
**Figure 3**). (3) Cancer cells of MBC samples, including ER⁺HER2⁺ and ER⁺HER2⁻
samples, showed higher scores of metastasis-related signatures than FBC samples,
especially angiogenesis and cell migration (**Response Figure 4**). These results
indicated that MBC samples had a lower level of immune infiltration, especially
ER⁺HER2⁻ MBC samples. Both ER⁺HER2⁺ and ER⁺HER2⁻ MBC samples had more
active FASN expression and metastasis-related signatures than FBC samples.

Response Figure 2 (Related to Supplementary Figure 4b-c in revised manuscript).

Comparison of cellular components in ER⁺HER2⁻ MBC, ER⁺HER2⁺ MBC, ER⁺HER2⁻ FBC, and ER⁺HER2⁺ FBC samples in the scRNA-seq and TCGA dataset.

(a) Boxplot showing the percentage of cancer cells, T cells, B cells, endothelial cells, macrophages, mast cells and fibroblasts in ER⁺HER2⁻ MBC, ER⁺HER2⁺ MBC, ER⁺HER2⁻ FBC, and ER⁺HER2⁺ FBC samples for ScRNA-seq data. HER2 status is defined by IHC experiments.

(b) Boxplot showing the tumor purity and signature scores of T cells, B cells, endothelial cells, macrophages, mast cells and fibroblasts in ER⁺HER2⁻ MBC, ER⁺HER2⁺ MBC, ER⁺HER2⁻ FBC, and ER⁺HER2⁺ FBC in TCGA ER⁺ BRCA cohort. HER2 status is based on the IHC results in the clinical information of the TCGA-BRCA dataset. P-value was calculated by two-sided Wilcoxon rank-sum test.

Response Figure 3 (Related to Figure 4c-d and Supplementary Figure 8b, d in revised manuscript).

The comparison of expression levels of FASN between MBC and FBC samples.

(a) Violin plot of FASN expression in cancer cells from male and female samples. P-value was calculated by two-sided Wilcoxon rank-sum test.

(b) Violin plot showing FASN expression in cancer cells from ER⁺HER2⁻ MBC, ER⁺HER2⁺ MBC, ER⁺HER2⁻ FBC, and ER⁺HER2⁺ FBC samples in our scRNA-seq dataset. P-value was calculated by two-sided Wilcoxon rank-sum test.

(c) Violin-boxplots showing the FASN expression among male and female samples in TCGA ER⁺ BRCA cohort. P-value was calculated by two-sided Wilcoxon rank-sum test.

(d) Boxplot showing the FASN expression among ER⁺HER2⁻ MBC, ER⁺HER2⁺ MBC, ER⁺HER2⁻ FBC, and ER⁺HER2⁺ FBC samples in TCGA BRCA cohort. P-value was calculated by two-sided Wilcoxon rank-sum test.

Response Figure 4 (Related to Supplementary Figure 6a-b in revised manuscript). Comparison of metastasis signature scores of cancer cells in ER⁺HER2⁻ MBC, ER⁺HER2⁺ MBC, ER⁺HER2⁻ FBC, and ER⁺HER2⁺ FBC samples. (a) Heatmap showing the average ssGSEA scores of cell migration, EMT and angiogenesis in cancer cells from ER⁺HER2⁻ MBC, ER⁺HER2⁺ MBC, ER⁺HER2⁻ FBC, and ER⁺HER2⁺ FBC samples. (b) Violin plots comparing the scores of cell migration, EMT and angiogenesis of cancer cells from ER⁺HER2⁻ MBC, ER⁺HER2⁺ MBC, ER⁺HER2⁻ FBC, and ER⁺HER2⁺ FBC samples. P-value was calculated by two-sided Wilcoxon rank-sum test.

We added the corresponding description in the revised Method section as follows (Lines 544-550): “Single-cell transcriptomic data from six MBC and thirteen FBC samples were analyzed, in which eleven FBC samples were collected from a previous study by Wu et al.⁵⁷, and other samples were in-house. All of the collected samples were ER⁺. We defined the ER, PR, HER2, and KI67 status using IHC, and further evaluated the amplification of HER2 based on FISH. The clinicopathological characteristics were shown in **supplementary table 1**. All the collected samples (including MBC and FBC) were negative for HER2 amplification evaluated by FISH.”

Also, the corresponding results of immune infiltration were updated in the revised Results section as follows (Lines 166-175): “In order to figure out whether the HER2 status has an influence on the comparison of cellular components between MBC and FBC, we further compared the immune infiltration among groups of ER⁺HER2⁺ MBC, ER⁺HER2⁻ MBC, ER⁺HER2⁺ FBC, and ER⁺HER2⁻ FBC samples. Both the scRNA-seq data and TCGA-BRCA data consistently showed that the ER⁺HER2⁻ MBC samples had the highest level of cancer cell enrichment and significantly lower level of T cell and B cell percentages (Supplementary Figure 4b, c). Besides, it seemed that the T and B cell percentages were higher in ER⁺HER2⁺ MBC than in ER⁺HER2⁻ MBC samples, although further evaluation was needed in a larger cohort.”

The comparison results of FASN expression among groups of ER⁺HER2⁺ MBC, ER⁺HER2⁻ MBC, ER⁺HER2⁺ FBC, and ER⁺HER2⁻ FBC samples were added in the revised Results section as follows (Lines 224-226): “Single cancer cells from both ER⁺HER2⁺ and ER⁺HER2⁻ MBC samples showed higher expression of FASN than FBC samples, independent of HER2 status (Supplementary Figure 8b).”

The comparison results of metastasis signature scores of cancer cells from
ER⁺HER2⁺ MBC, ER⁺HER2⁻ MBC, ER⁺HER2⁺ FBC, and ER⁺HER2⁻ FBC samples
were added in the revised Results section as follows (**Lines 195-197**): “Besides, cancer
cells from both ER⁺HER2⁺ and ER⁺HER2⁻ MBC showed higher scores of metastasis-
related signatures than FBC, especially angiogenesis and cell migration
(**Supplementary Figure 6**).”

4. The Authors may want to specify in the Methods section how correlations were
computed (Pearson? Spearman?). Moreover, whether P values are adjusted for multiple
testing or not should also be specified.

**Response:** Thank you for pointing this out. We apologize for not making this clear. The
correlations were calculated by the Pearson correlation coefficient (PCC). The p-values
here were not adjusted because each test was performed separately. We added the
corresponding description in the revised Method section as follows (**Lines 735-738**):
“Based on these signatures (Supplementary Table 4), we used ssGSEA to assess the
scores of tumor metastasis. The Pearson correlation coefficient between fatty acid
metabolism score and metastasis-related signature scores was calculated by the “cor.test”
function for TCGA pan-cancer samples.” The description in the Results section was
also revised as follows (**Lines 253-256**) “As our above results showed that cancer cells
from MBC patients had higher metastasis-related signature scores, we further explored
the correlations between fatty acid metabolism and metastasis in ER⁺ breast cancers of
the TCGA dataset by calculating the Pearson correlation coefficient (PCC)”. Besides,
the method of correlation analysis were added in the revised figure legends of Figure
4h-k, supplementary figure 3a, and 8g.

5. Line 176: “up-regulated” may be replaced with “upregulation”.

**Response:** Thank you. We replaced the “up-regulated” with “upregulation” in the
revised manuscript (**Line 203**).

6. Lines 219-226: The Authors describe a positive correlation between FASN
expression and tumor purity, and a negative correlation not only with immune cells but
also with CAFs and endothelial cells. The Authors then suggest that elevated expression
of FASN may promote immune escape. Since higher tumor purity is also necessarily
associated with lower stroma content (including immune cells), the cause-effect
relationship suggested by the Authors may not be necessarily proved by these findings.
This section may need to be adjusted accordingly.

**Response:** Thank you for your comments. We agree with the reviewer’s concern.
Accordingly, we revised the corresponding part as follows (**Lines 273-279**): “Thus, we
performed a pan-cancer analysis to evaluate the association between FASN expression

and immune infiltration in TCGA datasets. Results showed that FASN expression and
tumor purity were positively correlated in most cancers, while the infiltration scores of
T cells and B cells were negatively associated with FASN expression (**Supplementary**
**Figure 8h**). These results implied that the elevated expression of FASN may be
associated with the immune exclusion.”

7. Similarly, the positive correlation with metastasis-related pathways (e.g. suggested
in line 374-376) may not necessarily mean “causation”, as many other factors could
play a role.

**Response:** Thank you for your comments. We apologize for the inappropriate statement.
We revised the corresponding part as follows (**Lines 502-505**): “Notably, the fatty acid
metabolism showed a positive correlation with metastasis, and a negative correlation
with immune infiltration, implying the activated fatty acid metabolism might involve
in the immunological suppression and metastasis of MBC.”

8. Line 270 – Paragraph “MBC-specific T cells that co-expressed epithelial and immune
markers were in the apoptosis stage”: here, the Authors describe the presence of T cells
showing both T and epithelial cell markers, suggesting the existence of “epithelial-T
cells”. The Authors tried to exclude that this finding were to be related to technical
artifacts in the single-cell analysis, and showed the coexistence of CD3E and KRT8
markers with immunofluorescence experiments.

Although intriguing, further validation in other available single cell datasets (e.g. doi:
10.1038/s41588-021-00911-1) is in my opinion warranted, as it would give more
robustness to this finding.

**Response:** Thank you for your valuable suggestion. With the development of single-
cell techniques, we could investigate the cellular characteristics at high resolution and
identify the previously unappreciated cells. Intriguingly, a study from Hu et al. reported
a non-traditional CD45⁺EpCAM⁺ cell population in the fallopian tube epithelial layer
of ovarian cancer patients (Hu et al., Cancer Cell, 2020, 37(2), 226-242). This
population was also positive for CD3, CD44, CD69, and CD103, suggesting that these
cells are possibly tissue-resident memory T lymphocytes (TRMs). They identified these
cells by scRNA-seq (Smart-Seq2) and validated them using immunofluorescence
experiments. However, the biological and clinical implications of this population are
unclear yet. To further validate the existence of “epithelial-T cells” in breast cancer, we
downloaded and performed an integrated analysis for the scRNA-seq data of ER⁺
BRCA from the previous study (Wu et al., Nature genetics, 2021, 53(9): 1334-1347)
suggested by the reviewer, in which all the samples were from female patients. By
integrating the transcriptomic data of T cells from in-house and Wu et al. (**Response**

**Figure 5a**), we calculated the percentage of CD3E⁺KRT8⁺ T cells of in-house MBC,
 in-house FBC, and Wu et al.'s FBC samples, respectively. Results showed that MBC
 samples had a significantly higher percentage of CD3E⁺KRT8⁺ T cells than the FBC
 samples from the two datasets (**Response Figure 5b, 5c**). Besides, the percentages of
 CD3E⁺KRT8⁺ T cells were similar in in-house and Wu et al.'s FBC samples (**Response**
 **Figure 5c**), suggesting the existence of CD3E⁺KRT8⁺ T cells and the enrichment of
 these cells in male samples. We also excluded the influence of doublets or multipl
 by evaluating the CD3E⁺KRT8⁺ T cell percentage under different cell-filtering criteria
 for in-house MBC, in-house FBC, and Wu et al.'s FBC datasets. Considering there may
 be more expressed genes that could be detected in doublets or multipl
 the number of expressed genes within each single cell using different cutoffs, rang
 from 1500 to 5000. Results showed that, in all of the three datasets, the percentag
 CD3E⁺KRT8⁺ T cells did not decline with the screening criteria became strict and
 remained at a robust level in all tests (**Response Figure 5d**), partially avoiding the
 technical artifacts caused by doublets or multipl
 that CD3E⁺KRT8⁺ T cells existed in both in-house and Wu et al.'s data, especially in
 MBC samples. We added these results in the revised manuscript (**Lines 355-361**) as
 follows: "To further validate the existence of these cells, we calculated the percentag
 of CD3E⁺KRT8⁺ T cells of in-house MBC, in-house FBC, and Wu et al.'s FBC sampl
 respectively (**Supplementary Figure 13a, b**). Results showed that the percentages of
 CD3E⁺KRT8⁺ T cells were similar in in-house and Wu et al.'s FBC samples
 (**Supplementary Figure 13c**). MBC samples had a significantly higher percentage of
 CD3E⁺KRT8⁺ T cells than the FBC samples from the two datasets (**Supplementary**
 **Figure 13c**)."

**Response Figure 5 (Related to Supplementary Figure 13a-b and Supplementary Figure 14a in**

**revised manuscript). Evaluation of the existence of CD3E⁺KRT8⁺ T cells in the scRNA-seq**
**dataset. (a)** T-SNE plot of T cells colored by data sources. **(b)** T-SNE plots showing the distribution
of CD3E⁺KRT8⁻ and CD3E⁺KRT8⁺ T cells in in-house MBC samples (left), in-house FBC samples
(middle), and FBC samples from Wu et al. (right). **(c)** Barplot showing the percentage of
CD3⁺KRT8⁺ T cells in different datasets. **(d)** The line chart showing the percentage of
CD3E⁺KRT8⁺ T cells under different feature filter thresholds in in-house MBC samples (left), in-
house FBC samples (middle), and FBC samples from Wu et al. (right).

9. Moreover, the section included in lines 278-284 may need to be explained in a clearer
way.

**Response:** Thanks for mention it. We revised the corresponding explanation as follows
**(Lines 378-390):** “By evaluating the CD3E⁺KRT8⁺ T cell percentage under different
cell-filtering criteria, we excluded the influence of low-quality cells that would be
possibly included during the tissue dissociation, including the doublets or multiplets
and broken/dying cells. Considering there may be more expressed genes that could be
detected in doublets or multiplets, we limited the number of expressed genes within
each single cell using different cutoffs, ranging from 1500 to 5000. Also, dying or
broken cells often exhibit extensive mitochondrial contamination. Thus, we calculated
the percentage of reads that mapped to the mitochondrial genome in each single cell.
Gradient cell-filtering criteria were performed to limit the number of expressed genes
and mitochondrial reads percentage. Results showed that the percentage of
CD3E⁺KRT8⁺ T cells did not decline with the screening criteria becoming strict and
remained at a robust level in all tests **(Supplementary Figure 14a, b)**, partially
avoiding the technical artifacts caused by low-quality cells.”

10. Indeed, as per Supplementary Figure 6, panel A (Differentially expressed genes and
functional analysis of T cells between male and female patients), and as suggested by
the Authors (lines 274-278), it seems that KRT genes are expressed only in T cells from
sample M3 (and maybe to a lesser extent M1). Confirming these findings in other
single-cell datasets would be useful to exclude that the presence of epithelial-T cells are
patient-specific, as one may argue that they may not be specific of male breast cancer.

**Response:** In order to figure out whether the observed CD3E⁺KRT8⁺ T cells were
patient-specific or generally existed, we evaluated the percentage of CD3E⁺KRT8⁺ T
cells across 19 samples, including 6 in-house MBC samples, 2 in-house FBC samples,
and 11 FBC samples from Wu et al.. It turned out that 17/19 breast cancer samples had
CD3E⁺KRT8⁺ T cells with different degrees, ranging from 0.2% to 83.1% **(Response**
**Figure 6a)**. Especially, MBC samples showed higher percentage of CD3E⁺KRT8⁺ T
cell component (6.7% ~ 83.1%), and FBC samples had relatively lower percentage (0.2%
~ 17.9%). The Wilcoxon rank-sum test showed a significant difference of
CD3E⁺KRT8⁺ T cell enrichment between MBC and FBC groups **(Response Figure 6b;**

p-value: 0.0014). We added these results in the revised manuscript (Lines 361-377) as
 follows: “In order to figure out whether the observed CD3E⁺KRT8⁺ T cells were
 patient-specific or generally existed, we evaluated the percentage of CD3E⁺KRT8⁺ T
 cells across 19 samples, including 6 in-house MBC samples, 2 in-house FBC samples,
 and 11 FBC samples from Wu et al.. It turned out that 17/19 breast cancer samples had
 CD3E⁺KRT8⁺ T cells with different degrees, ranging from 0.2% to 83.1%
 (Supplementary Figure 13d). Especially, MBC samples showed higher percentage of
 CD3E⁺KRT8⁺ T cell component (6.7% ~ 83.1%), and FBC samples had relatively lower
 percentage (0.2% ~ 17.9%). We re-clustered the cells from each sample and then
 visualized all cell types and marker expressions at the single-cell level. MBC and FBC
 samples with the highest percentage of CD3⁺KRT⁺ cells were shown in Supplementary
 Figure 13e, f. To further evaluate the expression of KRT8/18/19 in T cells, we also
 showed the aggregated expression of these markers of epithelial and T cells in each
 sample using the dot-plot (Supplementary Figure 13g). The T cells from MBC2, MBC3,
 MBC4, MBC5, MBC6, and FBC13 had KRT8/18/19 expression, but were lower than
 these levels in epithelial cells. The Wilcoxon rank-sum test showed a significant
 difference of CD3E⁺KRT8⁺ T cell enrichment between MBC and FBC groups
 (Supplementary Figure 13h; p-value: 0.0014).”

**Response Figure 6 (Related to Supplementary Figure 13d, h in revised manuscript).**
 **Evaluation of the existence of CD3E⁺KRT8⁺ T cells in the scRNA-seq dataset.** (a) Barplot
 showing the percentage of CD3⁺KRT8⁺ T cells in each MBC and FBC sample. (b) Boxplot
 comparing the percentage of CD3E⁺KRT8⁺ T cells between MBC and FBC samples. P-value was
 calculated by two-sided Wilcoxon rank-sum test.

 11. Moreover, the presence of several genes related to dissociation or cellular stress (e.g.
 mitochondrial gene, FOSB, JUNB, heat-shock protein genes) and ribosomal genes, may
 raise a concern regarding contamination at the droplet level by dying cells (that would
 go in the same direction of what is stated in lines 308-313, when the Authors mention
 the high expression of apoptosis-related genes in this cell type). In this regard, can the
 Authors exclude that the finding of epithelial-T cells is not related to a potential issue
 of contamination? Indeed, in case of contamination, filtering by the number of genes
 may not be enough to identify technical artifacts.

Response: Thank you for your insightful comments. We agree with the reviewer that
some low-quality cells would be possibly included during the tissue dissociation,
including the stressed, broken, or dying cells, and doublets or multiplets. Firstly, by
performing the standard cell-filtering procedures that are commonly used in many
scRNA-seq studies, we had tried to limit the dissociation-related artifacts of multiplets
and broken/dying cells. Specifically, cells with expressed genes less than 200 or greater
than 6000 were excluded to remove the empty droplets and multiplets. Considering that
dying cells often exhibit extensive mitochondrial contamination, we calculated the
percentage of reads that mapped to the mitochondrial genome and filtered cells that
had >25% mitochondrial reads. Secondly, gradient cell-filtering criteria were
performed to limit the number of expressed genes and mitochondrial reads percentage.
Results showed that the percentage of CD3E⁺KRT8⁺ T cells did not decline with the
mitochondria filtering threshold (**Response Figure 7a**), indicating that the observation
of these cells may be not caused by technical artifacts. Thirdly, to further address the
concern of cellular stress and dying cell contamination, we performed GSEA analyses
using the signature of mitochondria, ribosome, and heat-shock protein for the gene
expression profile of T cells. Results showed that the up-regulated genes of
CD3E⁺KRT8⁺ T cells were not enriched by these signatures (**Response Figure 7b**).
Furthermore, we found that the CD3E⁺KRT8⁺ T cells had significantly higher
expression levels of genes related to ‘Granzyme A mediated apoptosis pathway’ and ‘T
cell receptor regulation of apoptosis’, but not enriched in ‘Apoptosis modulation by
HSP70’ (**Response Figure 7c**), indicating that the apoptosis of these cells was induced
by immune response rather than cellular stress. We added the description of the above
results in the revised manuscript as follows (**Lines 378-394**): “By evaluating the
CD3E⁺KRT8⁺ T cell percentage under different cell-filtering criteria, we excluded the
influence of low-quality cells that would be possibly included during the tissue
dissociation, including the doublets or multiplets and broken/dying cells. Considering
there may be more expressed genes that could be detected in doublets or multiplets, we
limited the number of expressed genes within each single cell using different cutoffs,
ranging from 1500 to 5000. Also, dying or broken cells often exhibit extensive
mitochondrial contamination. Thus, we calculated the percentage of reads that mapped
to the mitochondrial genome in each single cell. Gradient cell-filtering criteria were
performed to limit the number of expressed genes and mitochondrial reads percentage.
Results showed that the percentage of CD3E⁺KRT8⁺ T cells did not decline with the
screening criteria becoming strict and remained at a robust level in all tests
(Supplementary Figure 14a, b), partially avoiding the technical artifacts caused by low-
quality cells. To further address the concern of cellular stress and dying cell
contamination, we performed GSEA analyses using the signature of mitochondria,

ribosome, and heat-shock protein for the gene expression profile of T cells. Results
 showed that the up-regulated genes of CD3E⁺KRT8⁺ T cells were not enriched in these
 signatures (Supplementary Figure 14c).” and (Lines 422-425): “We found that the
 CD3E⁺KRT8⁺ T cells had significantly higher expression levels of genes related to
 apoptosis induced by the immune response, such as granzyme-A and T cell receptor
 mediated apoptosis pathway, but not enriched in the apoptosis related to cellular stress
 (Supplementary Figure 14d).”

 **Response Figure 7 (Related to Supplementary Figure 14b-d in revised manuscript). Validation**
 **and functional analysis of CD3E⁺KRT8⁺ T cells.** (a) The line chart showing the percentage of
 CD3E⁺KRT8⁺ T cells under different mitochondria filter thresholds in in-house MBC samples (left),
 in-house FBC samples (middle), and FBC samples from Wu et al. (right). (b) GSEA analysis of
 mitochondria (left), ribosome (middle) and regulation of HSF-1 mediated heat shock response (right)
 pathway between CD3E⁺KRT8⁺ and CD3E⁺KRT8⁻ T cells. (c) Violin plots showing the scores of
 apoptosis-related pathways in CD3E⁺KRT8⁺ and CD3E⁺KRT8⁻ T cells.

 12. With regards to immunofluorescence (Figure 5 B), can the Authors quantify the
 number of cells with co-expression of the KRT8 and CD3E marker?

**Response:** Thank you for your professional suggestion. We are sorry for the unclear
 immunofluorescence results in the previous version. According to the advice from
 reviewer #3, we performed the immunofluorescence experiments again and showed the
 cells with different phenotypes, including CD3E⁺KRT8⁻, CD3E⁻KRT8⁺, and

CD3E⁺KRT8⁺ cells. According to the immunofluorescence, CD3E⁺KRT8⁺ cells were
located at the interface between KRT8⁺ epithelial cells and CD3⁺ T cells (**Response**
**Figure 8a**). Furthermore, flow cytometry of KRT8 and CD3 was performed using fresh
tumor tissues from two MBC patients to validate and quantify the number of
CD3E⁺KRT8⁺ cells (**Response Figure 8b**). We gated the CD45⁺ immune cells and
evaluated the expression of KRT8 of these cells. Results showed that there were 35.55%
and 2.11% CD45⁺KRT8⁺ cells in two samples, respectively. Notably, 57.07% and 20.82%
of these KRT8⁺ immune cells were CD3⁺ T cells in two samples. Thus, the
immunofluorescence and flow cytometry experiments indicated that the CD3⁺KRT8⁺
cells existed with various percentage in MBC samples. We added these corresponding
evidence in the revised manuscript as follows (**Lines 395-405**): “Further validation
using immunofluorescence experiments for the MBC sample confirmed the above
observation and showed that the CD3⁺KRT8⁺ cells were located at the interface
between KRT8⁺ epithelial cells and CD3⁺ T cells (Figure 6c). Furthermore, flow
cytometry of KRT8 and CD3 was performed using fresh tumor tissue from two MBC
patients to validate and quantify the number of CD3⁺KRT8⁺ cells (Figure 6d). We gated
the CD45⁺ immune cells and evaluated the expression of KRT8 in these cells. Results
showed that there were 35.55% and 2.11% CD45⁺KRT8⁺ cells in two samples,
respectively. Notably, 57.07% and 20.82% of these KRT8⁺ immune cells were CD3⁺ T
cells in two samples. Therefore, these results indicated the biological existence of
CD3⁺KRT8⁺ T cells and the enrichment of these cells with various percentages in MBC
samples.”

a

**Response Figure 8 (Related to Figure 6c-d in revised manuscript). Validation of the existence**
 **of CD3⁺KRT8⁺ T cells by the immunofluorescence and flow cytometry experiments. (a) The**
 **immunofluorescence staining of KRT8 and CD3 in an MBC sample. White arrows indicate the**
 **CD3⁺KRT8⁺ T cells. Scale bar, 50 μ m. (b) Flow cytometry showing the percentage of CD3⁺KRT8⁺**
 **cells in two MBC samples.**

12. Lines 504-509: Which statistical method was used to compare KEGG metabolic
 pathways in male and female clusters?

**Response:** Sorry for our unclear description. The differentially activated metabolic
 pathways between male and female cancer cell clusters were identified by the Wilcoxon
 rank-sum test. We revised the corresponding description as follows (**Lines 644-651**):
 **“The analysis of the metabolic pathways was performed as described previously by**
 **Xiao et al.⁶². Single-sample GSEA (ssGSEA) scores were calculated for 85 Kyoto**
 **Encyclopedia of Genes and Genomes (KEGG) metabolic pathways based on gene**
 **expression levels⁶³. The activity difference of KEGG metabolic pathways between male**
 **and female cancer cell clusters was measured by two-sided Wilcoxon rank-sum test. P-**
 **values were adjusted for multiple testing using the Benjamini-Hochberg method.**
 **Pathways with adjusted p-value less than 0.05 were identified as differentially activated**
 **pathways between male and female cancer cell clusters.”**

13. Line 558: To ensure reproducibility, which criteria were used for the selection of
 ER-positive samples in the TCGA? Since some samples in the single-cell cohorts were
 HER2-positive according to Supplementary Table 1, why was this group excluded from
 the TCGA analysis?

**Response:** We selected the ER⁺ TCGA-BRCA samples based on the clinical

information in the XenaBrowser website (<https://xenabrowser.net/datapages/>).
Specifically, 835 primary tumor samples with positive
breast_carcinoma_estrogen_receptor_status were selected, including both HER2⁺ and
HER2⁻ samples. Samples without RNA-seq data were further removed. Finally, we
obtained the transcriptomic and clinical data of 722 ER⁺ TCGA-BRCA samples,
including 598 ER⁺HER2⁻ FBC, 112 ER⁺HER2⁺ FBC, 9 ER⁺HER2⁻ MBC, and 3
ER⁺HER2⁺ MBC samples. To figure out the influence of HER2 status on the
observation in this study, we further compared the immune infiltration, FASN
expression, and metastasis signatures scores among these four groups (**Response**
**Figure 2-4**). We added the description for sample selection of TCGA data in the revised
Methods section as follows (**Lines 711-720**): “Bulk transcriptomic data and clinical
information from The Cancer Genome Atlas (TCGA) database were downloaded and
extracted from the XenaBrowser website <https://xenabrowser.net/datapages/>. We
selected the ER⁺ TCGA-BRCA samples based on the clinical information. Specifically,
835 primary tumor samples with positive breast_carcinoma_estrogen_receptor_status
were selected, including both HER2⁺ and HER2⁻ samples. Samples without RNA-seq
data were further removed. Finally, we obtained the transcriptomic and clinical data of
722 ER⁺ TCGA-BRCA samples, including 598 ER⁺HER2⁻ FBC, 112 ER⁺HER2⁺ FBC,
9 ER⁺HER2⁻ MBC, and 3 ER⁺HER2⁺ MBC samples. HER2 status is based on the IHC
results in the clinical information of the TCGA-BRCA dataset.”

14. Lines 567-571: To further validate the reliability of gene sets derived from the
single-cell dataset, these findings may be compared to those derived from available
immune-deconvolution tools (e.g. MCP-counter, EPIC, TIMER, xCell...) including the
cell types of interest (e.g. in terms of correlations).

**Response:** Thank you for your professional suggestion. We performed the immune-
deconvolution analysis for the 722 ER⁺ TCGA-BRCA samples using MCP-counter
(Becht E, et al. Genome biology, 2016), EPIC (Racle J, Gfeller D. Bioinformatics for
Cancer Immunotherapy, 2020), and xCell (Aran D, et al. Genome biology, 2017). By
comparing the scores of immune or stromal cell types calculated by these tools between
MBC and FBC samples, we found that the results of immune-deconvolution tools were
largely consistent with our previous observation based on signatures derived from the
single-cell dataset (**Response Figure 9**). Notably, results from both single-cell
signature and immune-deconvolution tools showed that the levels of T cells and B cells
were significantly higher in MBC samples than in FBC samples. Besides, we evaluated
the correlation of putative cell type levels derived from single-cell signatures and
immune-deconvolution tools and found a significantly positive correlation between
these methods (**Response Figure 10**), indicating the reliability of gene signatures

derived from our single-cell dataset, as well as the immunological difference between
 MBC and FBC. We added the above validation in the revised Results section as follows
 (Lines 146-159): “To further validate this result, we calculated the scores of various
 cell types for 722 ER⁺ TCGA-BRCA samples based on the gene signatures derived
 from our single-cell data (see Methods; Figure 2f). These scores between
 premenopausal and postmenopausal FBC patients were also compared (Figure 2g).
 Results verified that MBC had a relatively higher tumor purity and lower proportions
 of T cells and B cells, consistent with the observation at the single-cell level. The
 immunological components of TCGA samples were also verified using three immune-
 deconvolution tools, including MCP-counter¹⁷, EPIC¹⁸, and xCell¹⁹. We evaluated the
 correlation of putative cell type levels derived from single-cell signatures and immune-
 deconvolution tools and found a significantly positive correlation between these
 methods (Supplementary Figure 3a). Consistently, results from immune-
 deconvolution tools indicated that the levels of T cells and B cells were significantly
 lower in MBC samples than in FBC samples of the TCGA dataset (Supplementary
 Figure 3b)”.

We also added the description of the corresponding validation procedures in the
 revised Methods section as follows (Lines 723-730): “We identified the top ten genes
 with the highest fold-changes of each cell type in our single-cell data and then
 calculated the ssGSEA scores of these gene signatures for bulk samples. The scores of
 immune or stromal cells were compared between MBC and FBC samples using two-
 sided Wilcoxon rank-sum test, which was a non-parametric test that did not assume
 known distributions⁶⁵. To further validate the reliability of gene signatures derived
 from the single-cell dataset, we measured the enrichment of TME cells by using
 immune-deconvolution tools MCP-counter¹⁷, EPIC¹⁸, and xCell¹⁹.”

**Response Figure 9 (Related to Supplementary Figure 3b in revised manuscript). Cellular**
 **components in TCGA MBC and FBC ER+ samples inferred by immune-deconvolution tools.**
 **Boxplot showing the scores of immune and stromal cells in TCGA MBC and FBC ER+ samples**
 **inferred by xCell, MCP, and EPIC. P-value was calculated by two-sided Wilcoxon rank-sum test.**

 **Response Figure 10 (Related to Supplementary Figure 3a in revised manuscript). The Pearson**
 **correlation analysis of putative cell type levels derived from single-cell signatures and**
 **immune-deconvolution tools.**

15. Line 229: From Supplementary Figure 4, it seems that FASN high is significantly
 ($P = 0.04$) associated with OS in female breast cancer, and not in males ($P = 0.27$),
 although only 12 male breast cancer samples were present in the TCGA. However, in
 the text, it is stated that “high expression of FASN could predict poor OS of male
 patients with BRCA”. This part may need to be rephrased.

Related to the comment on lines 588-595, other survival end-points, especially in
 the breast cancer, may be more informative than OS for evaluating the prognostic value
 of FASN expression. Lines 588-595, Survival analysis: Since overall survival data has
 to be interpreted carefully in the TCGA, especially for luminal breast cancer
 (doi:10.1016/j.cell.2018.02.052), did the authors tested also other survival end-points
 (PFI, DFI)?

**Response:** Thanks for your valuable suggestion. We apologize for the inaccurate
 statement. The significant results of overall survival analyses were shown in the revised
 supplementary figure (**Response Figure 11**). We agree with the reviewer that
 progression-free interval (PFI), disease-free interval (DFI), or disease-specific survival
 (DSS) are important for evaluating the prognostic value of FASN expression, especially
 for luminal breast cancer. Accordingly, we also performed survival analyses for PFI and
 DSS of TCGA pan-cancer datasets by categorizing the patients into FASN-high and
 FASN-low groups for each dataset according to the median of FASN expression. The
 analysis of DFI was not included due to the missing data of MBC samples. Results

showed that FASN expression was prognostic for the OS, PFI, and DSS of many types
of cancers, especially for male cancer patients (**Response Figure 11-13**). Male BRCA
patients with higher expression of FASN had a relatively poor prognosis but were not
statistically significant possibly due to that only 12 MBC samples were present in the
TCGA. Besides, high expression of FASN could predict poor OS and PFI of male
patients with bladder urothelial carcinoma (BLCA) and kidney renal clear cell
carcinoma (KIRC). The PFI of FASN-high male patients with kidney renal papillary
cell carcinoma (KIRP) and uveal melanoma (UVM) was also significantly poor. The
DSS of lung squamous cell carcinoma (LUSC) male patients with high FASN
expression was significantly poorer than those with low FASN expression. However,
the prognosis of female patients with these cancers was not associated with the FASN
expression.

We revised the corresponding description and added the undated results as follows
(**Lines 280-298**): “We performed analyses for overall survival (OS), progression-free
interval (PFI), and disease-specific survival (DSS) of TCGA pan-cancer datasets ²⁸ by
categorizing the patients into FASN-high and FASN-low groups for each dataset
according to the median of FASN expression. Results showed that FASN expression
was prognostic for the OS, DSS, and PFI of many types of cancers, especially for male
cancer patients (**Supplementary Figure 9-11**). Male BRCA patients with higher
expression of FASN had a relatively poor prognosis but were not statistically significant
possibly due to that only 12 MBC samples were present in the TCGA. Besides, high
expression of FASN could predict poor OS and PFI in male patients with bladder
urothelial carcinoma (BLCA) and kidney renal clear cell carcinoma (KIRC). The PFI
of FASN-high male patients with kidney renal papillary cell carcinoma (KIRP) and
uveal melanoma (UVM) was also significantly poor. The DSS of lung squamous cell
carcinoma (LUSC) male patients with high FASN expression was significantly poorer
than those with low FASN expression. However, the prognosis of female patients with
these cancers was not associated with the FASN expression. Notably, higher FASN
expression was prognostic for the poor DSS of PRAD patients, consistent with a
previous study that demonstrated that targeting FASN could inhibit the aggressive and
resistant PRAD²⁴. This result suggested that FASN may be a potential therapeutic target
for male patients with these cancers.”

overall survival

Response Figure 11 (Related to Supplementary Figure 9 in revised manuscript). Overall survival analysis of male and female patients in various cancer types based on the FASN expression. Patients are categorized into FASN-high and FASN-low groups for each dataset according to the median of FASN expression. BRCA: Breast invasive carcinoma; BLCA: Bladder Urothelial Carcinoma; KIRC: Kidney renal clear cell carcinoma; LAML: Acute Myeloid Leukemia; MESO: Mesothelioma; THCA: Thyroid carcinoma.

disease-specific survival

Response Figure 12 (Related to Supplementary Figure 10 in revised manuscript). Disease-

605 **specific survival analysis of male and female patients in various cancer types based on the**
 606 **FASN expression.** Patients are categorized into FASN-high and FASN-low groups for each dataset
 according to the median of FASN expression. BRCA: Breast invasive carcinoma; KIRC: Kidney
 renal clear cell carcinoma; KIRP: Kidney renal papillary cell carcinoma; THCA: Thyroid carcinoma;
 LUSC: Lung squamous cell carcinoma; PRAD: Prostate adenocarcinoma.

**Response Figure 13 (Related to Supplementary Figure 11 in revised manuscript). Progression-**
 **free interval analysis of male and female patients in various cancer types based on the FASN**
 **expression.** Patients are categorized into FASN-high and FASN-low groups for each dataset
 according to the median of FASN expression. BRCA: Breast invasive carcinoma; BLCA: Bladder
 Urothelial Carcinoma; KIRC: Kidney renal clear cell carcinoma; KIRP: Kidney renal papillary cell
 carcinoma; UVM: Uveal Melanoma.

 16. Line 247: For the SingleR tool analysis, which reference dataset was used? From
 Supplementary Fig. 5 it seems that some cell types are relatively “mixed” together and
 not well defined in the t-SNE. Did the Authors double checked manually if the
 automatic annotations were reliable?

**Response:** Thank you for your comment. We agree with the reviewer that the
 annotations of T cell subtypes using reference “MonacoImmuneData” in the SingleR
 package were confusing. When analyzing the updated transcriptomic dataset of 15,690
 single T cells from 19 BRCA samples, we had tried to annotate the subpopulations using
 multiple references from SingleR package but got some “mixed” results possibly due
 to the complicated phenotypes of T cells in different tissues and conditions. Therefore,
 we manually defined the T cell subpopulations based on the specifically-expressed

genes of each cell cluster in the revised manuscript, as shown in **Response Figure 14**.
 Besides, the top 30 genes that were specifically expressed in each subpopulation were
 listed in the revised supplementary table 6.

**Response Figure 14 (Related to Figure 5a and Supplementary Figure 12a in revised**
 **manuscript). Identification of T cell subpopulations. (a)** T-SNE plot showing the subpopulations
 **of T cells. (b)** Expression levels of representative genes in each subpopulation.

17. Line 433 and 437: The Cell Ranger versions mentioned are discordant (v2.1.0 and
 3.0.2). Is this correct?

**Response:** Thank you for your kind comment. Sorry for our mistake. The Cell Ranger
 version used for our data analysis pipeline is 3.0.2. We have further clarified this in the
 revised manuscript as follows (**Lines 585-586**): “The Cell Ranger v3.0.2 pipeline was
 performed to analyze the raw data and generate gene count data using the default and
 recommended parameters”.

18. Line 449: Please clarify if cells with more or less than 2000 expressed genes were
 retained.

**Response:** We apologize for the unclear description. In order to remove the empty
 droplets and multiplets, cells with expressed genes less than 200 or greater than 6000
 were excluded. We modified the description for quality control in the revised Methods
 section as follows (**Lines 593-595**): “To eliminate the influence of low-quality cells
 such as empty droplets and multiplets, cells with expressed genes less than 200 or
 greater than 6000 were excluded.”

19. Lines 454-455: Please rephrase specifying in a clearer way if UMI count and MT
 genes were used as regression terms in the ScaleData function(also, Authors may
 replace “ScaleDate” with “ScaleData” in the text).

**Response:** We apologize for not making this point clear. By using the default parameters,
 UMI count and MT genes were not regressed out in the ScaleData function. We revised
 the corresponding description as follows (**Lines 600-604**): “We identified the top 2000

variable features using the “vst” method for each dataset. Datasets were then anchored
and integrated using the integration procedure from the Seurat package to eliminate the
batch effects among the samples. ScaleData function was used to perform a linear
scaling transformation on the identified variable features using default parameters.”

20. Line 465: Please specify if the default parameters were used in the IntegrateData
function. Was default integration from Seurat applied from the beginning on all cells,
or just for specific cell types?

Response: Sorry for our ambiguous description. The integration procedure from the
Seurat package was performed at the beginning on all cells, not just for specific cell
types. In the revised manuscript, we rephrased this description according to the order
of data processing as follows (Lines 600-612): “We identified the top 2000 variable
features using the “vst” method for each dataset. Datasets were then anchored and
integrated using the integration procedure from the Seurat package to eliminate the
batch effects among the samples. ScaleData function was used to perform a linear
scaling transformation on the identified variable features using default parameters.
Principal component analysis (PCA) was performed on the scaled data to reduce the
dimensionality. The statistical significance of the PCA scores was determined using the
JackStraw function. The first 25 principal components were used for identifying the
neighbors and clustering the cells with a resolution of 1.5. The cell clusters were
visualized using 2D uniform manifold approximation projection (UMAP) or t-
distributed stochastic neighbor embedding (tSNE) plots. The FindAllMarkers function
was used to identify the genes specifically expressed in each cell cluster. We identified
the cell types based on the expression of well-established gene markers.”

21. Lines 475-480: Please explain in a clearer way this section. What did the “normal
cell cluster” used in inferCNV included (e.g. normal epithelial breast cells, stromal cells,
immune cells...)? Was it formed by “any other cell” that was not tagged as malignant?

Response: We apologize for the unclear description. The “normal cell clusters” included
immune cells (T cells, B cells, macrophages, mast cells) and stromal cells (fibroblasts
and endothelial cells). We revised the corresponding description in the manuscript
(Lines 616-622) as follows: “First, we identified malignant epithelial cells using the
marker genes EPCAM, KRT18, KRT14, and KRT19. To verify the identified cancer
cells more accurately, we also used the inferCNV R package⁶⁰ to evaluate copy number
variants (CNVs) levels, using immune cells (T cells, B cells, macrophages, and mast
cells) and stromal cells (fibroblasts and endothelial cells) as the control group and
epithelial cells as the test group.”

22. Line 492: Are P values adjusted for multiple testing or not? This should be stated in
the method sections for the other analyses as well.

Response: Thank you for pointing this out. P values were adjusted for multiple testing
when identifying differentially expressed genes or pathways throughout the whole
study. We updated the description of p-value adjustment in the revised manuscript as
follows: (1) **Lines 626-629**: “we identified genes with log₂ fold change greater than
0.25 and adjusted p-value less than 0.01 for each cluster. Based on the order of log₂
706 fold change, the top 100 genes were further identified as markers of each cluster.”. (2)
**Lines 647-651**: “The activity difference of KEGG metabolic pathways between male
and female cancer cell clusters were measured by two-sided Wilcoxon rank-sum test.
P-values were adjusted for multiple testing using the Benjamini-Hochberg method.
Pathways with adjusted p-value less than 0.05 were identified as differentially activated
pathways between male and female cancer cell clusters”. (3) **Lines 740-741**: “The gene
lists were submitted to Enrichr (<https://maayanlab.cloud/Enrichr/>) online tool, and the
top ten terms were retained according to the adjusted p-value”.

23. Line 509: Which statistical test was used to perform this comparison of metabolic
pathways between male and female clusters?

Response: Sorry for our unclear description. The differentially activated metabolic
pathways between male and female cancer cell clusters were identified by the two-side
Wilcoxon rank-sum test. We revised the corresponding description as follows (**Lines**
**647-651**): “The activity difference of KEGG metabolic pathways between male and
female cancer cell clusters were measured by two-sided Wilcoxon rank-sum test. P-
values were adjusted for multiple testing using the Benjamini-Hochberg method.
Pathways with adjusted p-value less than 0.05 were identified as differentially activated
pathways between male and female cancer cell clusters.”

24. Figure 3, panel G: Can the Authors add a value for the correlations showed?

Response: We apologize for forgetting to show the p-values and correlation coefficients.
Both p-values and correlation coefficients were added in the corresponding figure
(**Response Figure 15, related to Figure 4j in the revised version**).

**Response Figure 15 (Related to Figure 4j in revised manuscript). The Pearson correlation**
 **analysis between the scores of metastasis-related signatures and fatty acid metabolic pathway**
 **in TCGA ER+ BRCA cohort.**

 25. Figure 3, panel J: The difference between the groups of comparisons (cell types and
 FASN high/low cells) is not clear and may be specified in the Figure legend. Are the 4
 main columns representing interactions with opposite directions?

**Response:** We apologize for the confusing visualization and unclear description in the
 previous version. To comprehensively illustrate the cell-cell communications in MBC
 and FBC samples, we re-analyzed the inter-cellular interactions using the updated
 single-cell datasets. The ligand-receptor interactions were visualized using heatmaps
 and Sankey plots, as shown in **Response Figures 16 and 17**. To visualize more clearly,
 we split the interactions with opposite directions into two plots, and marked the
 ‘common’, ‘male-specific’, and ‘female-specific’ interactions using different font
 colors (**Response Figure 16**). We also clarified the corresponding descriptions in the
 revised figure legends.

 **Response Figure 16 (Related to Figure 7c-d in revised manuscript). Heatmap showing the**
 **common, male-specific and female-specific ligand-receptor pairs in MBC and FBC samples.**

Response Figure 17 (Related to Figure 7e-f in revised manuscript). Sankey plot showing the representative examples of male-specific ligand-receptor pairs.

Response References:

1. Wu S Z, Al-Eryani G, Roden D L, et al. A single-cell and spatially resolved atlas of human breast cancers[J]. Nature genetics, 2021, 53(9): 1334-1347.

2. Hu Z, Artibani M, Alsaadi A, et al. The repertoire of serous ovarian cancer non-genetic heterogeneity revealed by single-cell sequencing of normal fallopian tube epithelial cells[J]. Cancer Cell, 2020, 37(2): 226-242.

3. Becht E, Giraldo N A, Lacroix L, et al. Estimating the population abundance of tissue-infiltrating immune and stromal cell populations using gene expression[J]. Genome biology, 2016, 17(1): 1-20.

4. Racle J, Gfeller D. EPIC: a tool to estimate the proportions of different cell types from bulk gene expression data[M]. Bioinformatics for Cancer Immunotherapy. Humana, New York, NY, 2020: 233-248.

5. Aran D, Hu Z, Butte A J. xCell: digitally portraying the tissue cellular heterogeneity landscape[J]. Genome biology, 2017, 18(1): 1-14.

**Reviewer #2**

This is a well written and comprehensive manuscript describing the immune and
metabolic landscape of male breast cancer.

1. The premise of this paper that male and female breast cancers are immunological and
metabolically different is very compelling and may potentially provide new insights
into therapeutic strategies. The investigators have carefully evaluated a broad range of
proliferation, angiogenesis, and metabolic pathways as well as detailed immune
characterization. The study includes a limited number (3 and 2) reference cases. The
study is expanded by data from the TCGA.

**Response:** We are grateful for your comments. In order to further support and validate
the conclusion in this study, we expand the sample size of both male and female breast
cancer. In this revised version, six MBC and thirteen FBC samples were included, in
which eleven FBC samples were from a previous study by Wu et al. (Nature genetics,
2021, 53(9): 1334-1347. doi: 10.1038/s41588-021-00911-1) and other samples were in-
house. All of the collected samples were ER⁺. The transcriptome of 58,578 and 52,460
single-cells was sequenced in MBC and FBC, respectively. By performing the same
analysis procedure using this updated dataset, we found that the main results were
consistent with the previous version, and demonstrated the followings: (1) scRNA-seq,
bulk transcriptome, and immunohistochemistry consistently demonstrated that MBC
had a significantly lower degree of T cell infiltration than FBC; (2) metastasis-related
programs such as cell migration, epithelial-mesenchymal transition (EMT), and
angiogenesis were more active in cancer cells from MBC than FBC; (3) the activated
fatty acid metabolism involved by FASN was related to the cancer cell metastasis and
low immune infiltration of MBC; (4) different characteristics of T cell subpopulations
between MBC and FBC were identified. T cells in MBC showed activation of p38
MAPK and lipid oxidation pathways, indicating the dysfunctional state. In contrast, T
cells in FBC exhibited a higher expression level of cytotoxic markers such as GZMK
and KLRB1, and activated pathways mediated by immune-modulatory cytokines; (5)
the inhibitory interactions between cancer cells and T cells in the MBC
microenvironment were identified, such as cell-cell communications mediated by TGF-
β , TIGIT, and VSIR. (6) KRT8⁺ T cells with high level of fatty acid metabolism were
enriched in the MBC microenvironment. These observations were further validated in
bulk-RNAseq data and molecular experiments.

Despite the rarity of MBC occurrence and the stringent sample requirements of
single-cell experiments, we had collected and sequenced six MBC samples as possible
as we can. As far as we know, this study is the first to characterize the differences
between MBC and FBC at the single-cell resolution. Benefiting from the enlarged

sample size (6 MBC vs. 13 FBC), we could statistically evaluate the significance of the
observed differences between MBC and FBC samples. On the other hand, we also
discussed the explorative nature of this preliminary study in the revised manuscript as
follows (**Lines 529-533**): “Due to the rarity of MBC occurrence and the stringent
sample requirements of single-cell experiments, only limited MBC samples were
included in this study. However, this explorative study identified notable differences
between MBC and FBC, especially the distinct metabolic and immunological
characteristics of MBC patients. These observations need to be further validated with
larger sample sizes in the future.”

[FIGURE REDACTED]

**Response Figure 18 (Related to Figure 1a in revised manuscript). Schematic workflow for**
**data collection and single-cell analysis in this study.**

2. Strength of the study include the clearly distinctive patterns that the evaluated male
and female breast cancers. The single cell sequencing is elegantly done, and the figures
are beautifully outlined and clearly delineated.

**Response: Thank you for the positive evaluation of our work.**

3. A major concern of the study is that the female breast cancers neither have ER
expression (ESR1) nor ER activity. Male breast cancer is mostly ER+, whereas female
breast cancer has a broad diversity ranging from triple negative disease to ER+ and
HER2 positive disease. The immune landscape, EMT, angiogenesis is vastly different
in these subtypes. Particularly, TNBC stand out in their immune profile. The data would
be very much strengthened if the authors provided data on ER+ female breast cancer,
to show how this is similar or different from an ER+ male breast cancer.

**Response: All of the collected male and female samples were from ER+ breast cancer**
**patients without HER2 amplification. The clinicopathological characteristics of the**
**collected samples were listed in the revised supplementary table 1, including age, ER**
**status, PR status, IHC results for HER2, FISH results for HER2, KI67 level, tissue size,**
**and TNM stage. Accordingly, we clarified the description of clinicopathological**
**characteristics of the collected samples in the revised Methods section as follows (Lines**
**544-550): “Single-cell transcriptomic data from six MBC and thirteen FBC samples**
**were analyzed, in which eleven FBC samples were collected from a previous study by**
**Wu et al.⁵⁷, and other samples were in-house. All of the collected samples were ER+.**
**We defined the ER, PR, HER2, and KI67 status using IHC, and further evaluated the**
**amplification of HER2 based on FISH. The clinicopathological characteristics were**
**shown in supplementary table 1. All the collected samples (including MBC and FBC)**

were negative for HER2 amplification evaluated by FISH.”

4. Furthermore, a more in-depth explanation on the significance of the findings. The
error bars appear very wide in a large number of examples. How are the p-values
adjusted for significance in this multi-parameter assessment?

Response: Thank you for pointing this out. In order to address this concern, we used
the violin-boxplots to better visualize the distribution of data in the revised Figure 4
and 5 (Response Figure 19-21). Specifically, the shape of violins represents the data’s
density: the thicker part means the values in that section of the violin have higher
frequency, and the thinner part implies lower frequency. Boxplots were also added
inside the violins to show the medians, ranges and variabilities of the data. P values
were adjusted for multiple testing when identifying differentially expressed genes or
pathways throughout the whole study. We updated the description of p-value adjustment
in the revised manuscript as follows: (1) Lines 626-629: “we identified genes with log2
862 fold change greater than 0.25 and adjusted p-value less than 0.01 for each cluster. Based
on the order of log2 fold change, the top 100 genes were further identified as markers
of each cluster.” (2) Lines 647-651: “The activity difference of KEGG metabolic
pathways between male and female cancer cell clusters was measured by two-sided
Wilcoxon rank-sum test. P-values were adjusted for multiple testing using the
Benjamini-Hochberg method. Pathways with adjusted p-value less than 0.05 were
identified as differentially activated pathways between male and female cancer cell
clusters”. (3) Lines 740-741: “The gene lists were submitted to Enrichr
(<https://maayanlab.cloud/Enrichr/>) online tool, and the top ten terms were retained
according to the adjusted p-value”.

Response Figure 19 (Related to Figure 4a in revised manuscript). Violin-boxplots showing the
signature scores of fatty acid metabolic pathways in cancer cells of male and mixed/female
clusters. P-value was calculated by two-sided Wilcoxon rank-sum test and adjusted for multiple
testing using the Benjamini-Hochberg method.

Response Figure 20 (Related to Figure 4c-d in revised manuscript). The expression levels of FASN between MBC and FBC samples in ScRNA-seq data and TCGA ER+ BRCA cohort.

Response Figure 21 (Related to Figure 5e in revised manuscript). Violin plot of p38 MAPK activity in CD8+ T cells from MBC and FBC samples. P-value was calculated by two-sided Wilcoxon rank-sum test.

5. TCGA data while compelling is not novel and may not provide sufficient annotations
to clinical

**Response:** Thank you for your professional advice. We agree with the reviewer that it
would be more convincing to validate the findings of this study using multiple
independent datasets of male breast cancer (MBC). Besides, the limited number of
MBC samples in the TCGA dataset may not sufficient for the comparison between FBC
and MBC samples. Thus, we collected two gene expression profiles of MBC samples
from previous studies, GSE104730 (RNA-seq, 46 samples, Severson T M, et al. Nature
communications, 2018) and GSE31259 (microarray data, 74 samples, Johansson I, et
al. Breast Cancer Research, 2012). Using the analysis procedure based on the ssGSEA
algorithm, we calculated the scores of immune or stromal cells for MBC samples from
TCGA, GSE104730, GSE31259, as well as for FBC samples from the TCGA dataset.
These scores were compared between MBC and FBC samples using two-sided
Wilcoxon rank-sum test. Results showed that the scores of T cells and B cells were
significantly lower in MBC samples from three independent datasets than in FBC
samples, confirming the results of low immune infiltration in MBC samples observed
in the single-cell dataset (**Response Figure 22**). We added this validation result in the
revised manuscript as follows (**Lines 159-166**): “**To further verify this result with larger**
**MBC sample size, we also collected two gene expression profiles of MBC samples**

from previous studies, including RNA-seq data of 46 MBC samples (GSE104730)⁶ and
 microarray data of 74 MBC samples (GSE31259)²⁰. We calculated and compared the
 scores of immune or stromal cells for MBC samples from three datasets, and for FBC
 samples from the TCGA dataset. Results showed that the scores of T cells and B cells
 were significantly lower in MBC samples from three independent datasets than in FBC
 samples (Supplementary Figure 4a), further confirming the results of low immune
 infiltration in MBC samples.” However, we failed to collect the survival data of MBC
 patients except for the TCGA dataset, possibly due to the rarity of MBC occurrence.

 **Response Figure 22 (Related to Supplementary Figure 4a in revised manuscript). Comparison**
 **of cellular components between MBC and FBC in independent datasets.** Boxplots showing the
 signature scores of T cells, B cells, endothelial cells, macrophages, mast cells and fibroblasts in ER⁺
 MBC samples from GSE104730, GSE31259, and TCGA datasets, as well as ER⁺ FBC samples
 from TCGA dataset. P-value was calculated by two-sided Wilcoxon rank-sum test.

 **Response References**

- 1. Wu S Z, Al-Eryani G, Roden D L, et al. A single-cell and spatially resolved atlas of human
 breast cancers[J]. Nature genetics, 2021, 53(9): 1334-1347.
- 2. Severson T M, Kim Y, Joosten S E P, et al. Characterizing steroid hormone receptor
 chromatin binding landscapes in male and female breast cancer[J]. Nature communications,
 2018, 9(1): 1-12.
- 3. Johansson I, Nilsson C, Berglund P, et al. Gene expression profiling of primary male breast
 cancers reveals two unique subgroups and identifies N-acetyltransferase-1 (NAT1) as a
 novel prognostic biomarker[J]. Breast Cancer Research, 2012, 14(1): 1-15.

**Reviewer #3**

Male breast cancer (MBC) is associated with worse prognosis compared to female
breast cancer and the cellular and molecular differences between the two remain unclear.
The researchers used single-cell RNA (scRNA) sequencing and T cell receptor (scTCR)
sequencing characterize the tumor microenvironment of MBC. They sequenced three
MBC and two post-menopausal ER⁺ female breast cancers (FBC) and show evidence
that MBC have lower immune infiltration, activated ER and AR regulons, higher fatty
acid synthase (FASN) expression, and exhausted CD8 T cells. The authors identify a
subset of T-cells that express epithelial cytokeratins. However, the manuscript is lacking
good quality evidence for the existence of these epithelial-T cells. The authors should
consider removing that entire section or provide additional experiments to validate their
findings. Androgens have long been known to drive fatty acid synthase PMID: 9067276,
and the authors show good evidence of AR regulon activation in MBC, perhaps more
focus on the androgen receptor would tie this story together. Overall, the study is of
interest, but more experiments and analysis are needed for this study.

Specific comments:

1. While two of the three MBC samples have low immune infiltrate, one actually has
similar levels to the two other FBC samples (Figure 1e). Therefore, on cannot conclude
that there are less immune cells in MBC, as this may just be a sampling artefact.

**Response:** We are grateful for your comments. We agree with the reviewer that the
conclusions are not convincing due to the limited sample size for the scRNA analysis.
In order to further support and validate the conclusion in this study, we expand the
sample size of both male and female breast cancer. In this revised version, 6 MBC and
13 FBC samples were included, in which eleven FBC samples were from a previous
study by Wu et al. (Nature genetics, 2021, 53(9): 1334-1347. doi: 10.1038/s41588-021-
00911-1) and other samples were in-house. All of the collected samples were ER⁺. The
transcriptome of 58,578 and 52,460 single-cells was sequenced in MBC and FBC,
respectively (**Response Figure 23**). By performing the same analysis procedure using
this updated dataset, we found that the main results were consistent with the previous
version. Benefiting from the enlarged sample size of scRNA-seq data, we could
statistically evaluate the significance of the cellular component difference between
MBC and FBC samples. Results showed that compared with FBC, MBC showed a
significantly higher proportion of cancer cells and a lower proportion of immune cells,
such as T cells and B cells, indicating a lower level of immune infiltration (**Response**
**Figure 24a-d**). These immune cell proportions had no obvious differences between

premenopausal and postmenopausal FBC patients (**Response Figure 24e**). To further
validate this result, we calculated the scores of various cell types for 722 ER⁺ TCGA-
BRCA samples based on the gene signatures derived from single-cell data (**Response**
**Figure 24f**). These scores between premenopausal and postmenopausal FBC patients
were also compared (**Response Figure 24g**). Results verified that MBC had a relatively
higher tumor purity and lower proportions of T cells and B cells, consistent with the
observation at the single-cell level. These observations of immunological components
of TCGA samples were also verified using three immune-deconvolution tools,
including MCP-counter, EPIC, and xCell. Consistently, results from these immune-
deconvolution tools indicated that the levels of T cells and B cells were significantly
lower in MBC samples than in FBC samples of the TCGA dataset (**Response Figure**
**25**). To further verify this result with larger MBC sample size, we also collected two
gene expression profiles of MBC samples from previous studies, including RNA-seq
data of 46 MBC samples (GSE104730) and microarray data of 74 MBC samples
(GSE31259). Using the analysis procedure based on the ssGSEA algorithm, we
calculated and compared the scores of immune or stromal cells for MBC samples from
three datasets, and for FBC samples from TCGA dataset. Results showed that the scores
of T cells and B cells were significantly lower in MBC samples from three independent
datasets than in FBC samples (**Response Figure 26**). Furthermore, we performed
immunohistochemistry (IHC) analysis for T cell markers CD4 and CD8 in 30 ER⁺ MBC
and 30 ER⁺ FBC samples. Results suggested that T cell markers had a lower expression
proportion in MBC than in FBC (**Response Figure 27**). Therefore, the analysis of
scRNA-seq, bulk transcriptome and IHC consistently demonstrated that MBC had a
significantly lower degree of immune cell infiltration than FBC.

We added the above results in the revised manuscript as follows (**Lines 142-166**):
“Results showed that compared with FBC, MBC showed a significantly higher
proportion of cancer cells and a lower proportion of immune cells, such as T cells and
B cells, indicating a lower level of immune infiltration (**Figure 2a-d**). These immune
cell proportions had no obvious differences between premenopausal and
postmenopausal FBC patients (**Figure 2e**). To further validate this result, we calculated
the scores of various cell types for 722 ER⁺ TCGA-BRCA samples based on the gene
signatures derived from our single-cell data (**see Methods; Figure 2f**). These scores
between premenopausal and postmenopausal FBC patients were also compared
(**Figure 2g**). Results verified that MBC had a relatively higher tumor purity and lower
proportions of T cells and B cells, consistent with the observation at the single-cell level.
The immunological components of TCGA samples were also verified using three
immune-deconvolution tools, including MCP-counter¹⁷, EPIC¹⁸, and xCell¹⁹. We
evaluated the correlation of putative cell type levels derived from single-cell signatures

and immune-deconvolution tools and found a significantly positive correlation between
 these methods (**Supplementary Figure 3a**). Consistently, results from immune-
 deconvolution tools indicated that the levels of T cells and B cells were significantly
 lower in MBC samples than in FBC samples of the TCGA dataset (**Supplementary**
 **Figure 3b**). To further verify this result with larger MBC sample size, we also collected
 two gene expression profiles of MBC samples from previous studies, including RNA-
 seq data of 46 MBC samples (GSE104730)⁶ and microarray data of 74 MBC samples
 (GSE31259)²⁰. We calculated and compared the scores of immune or stromal cells for
 MBC samples from three datasets, and for FBC samples from the TCGA dataset.
 Results showed that the scores of T cells and B cells were significantly lower in MBC
 samples from three independent datasets than in FBC samples (**Supplementary Figure**
 **4a**), further confirming the results of low immune infiltration in MBC samples.”

[FIGURE REDACTED]

**Response Figure 23 (Related to Figure 1a in revised manuscript). Schematic workflow for**
 **data collection and single-cell analysis in this study.**

Response Figure 24 (Related to Figure 2a-g in revised manuscript). Comparison of cellular components between MBC and FBC samples. (a) The t-SNE plot of MBC, postmenopausal and premenopausal FBC samples. Colors represent cell types. **(b)** Sankey diagram showing the fraction of each cell type between male and female samples. **(c)** Sankey diagram showing the fraction of each cell type between MBC, postmenopausal and premenopausal FBC samples. **(d)** Boxplot showing the percentage of cancer cells, T cells, B cells, endothelial cells, macrophages, mast cells and fibroblasts in MBC and FBC samples. P-value was calculated by two-sided Wilcoxon rank-sum test. **(e)** Boxplot showing the percentage of cancer cells, T cells, B cells, endothelial cells, macrophages, mast cells and fibroblasts in MBC, postmenopausal and premenopausal FBC samples. P-value was calculated by two-sided Wilcoxon rank-sum test. **(f)** Boxplot showing the tumor purity and signature scores of various cell types between MBC and FBC in TCGA ER⁺ BRCA cohort. P-value was calculated by two-sided Wilcoxon rank-sum test. **(g)** Boxplot showing the tumor purity and signature scores of various cell types between MBC, postmenopausal and premenopausal FBC samples in TCGA ER⁺ BRCA cohort. P-value was calculated by two-sided Wilcoxon rank-sum test.

Response Figure 25 (Related to Supplementary Figure 3b in revised manuscript). Cellular components in TCGA MBC and FBC ER⁺ samples inferred by immune-deconvolution tools. Boxplot showing the scores of immune and stromal cells in TCGA MBC and FBC ER⁺ samples inferred by xCell, MCP, and EPIC. P-value was calculated by two-sided Wilcoxon rank-sum test.

Response Figure 26 (Related to Supplementary Figure 4a in revised manuscript). Comparison of cellular components between MBC and FBC in independent datasets. Boxplots showing the signature scores of T cells, B cells, endothelial cells, macrophages, mast cells and fibroblasts in ER⁺ MBC samples from GSE104730, GSE31259, and TCGA datasets, as well as ER⁺ FBC samples from TCGA dataset. P-value was calculated by two-sided Wilcoxon rank-sum test.

Response Figure 27 (Related to Figure 2h-i in revised manuscript). Statistical quantification of CD4 and CD8 staining in MBC and FBC. (a) IHC images representing MBC and FBC samples stained for T cell markers CD4 and CD8. Scale bar, 20 μm. (b) Boxplot indicating the IHC scores of CD4 and CD8 in 30 ER⁺ male and 30 ER⁺ female patients (identified by the percentage of positive cells). P-value was calculated by two-sided Wilcoxon rank-sum test.

Moreover, we also discussed the explorative nature of this preliminary study in the revised manuscript as follows (Lines 529-533): “Due to the rarity of MBC occurrence and the stringent sample requirements of single-cell experiments, only limited MBC samples were included in this study. However, this explorative study identified notable differences between MBC and FBC, especially the distinct metabolic and immunological characteristics of MBC patients. These observations need to be further validated in more samples in the future.”

2. Please supply raw p-value and statistical test used in Fig.1g. There are only 12 male samples compared to 1085 female samples in the TCGA, therefore one likely cannot assume the MBC will represent a normal distribution unless proven.

Response: Thanks for your professional suggestions. We showed raw p-values in all figures of the revised version, including Figure 2, 3, 4, 5, 6, and Supplementary Figure 3-11. We selected the ER⁺ TCGA-BRCA samples based on the clinical information in the XenaBrowser website (<https://xenabrowser.net/datapages/>). Specifically, 835 primary tumor samples with positive breast_carcinoma_estrogen_receptor_status were selected. Samples without RNA-seq data were further removed. Finally, we obtained the transcriptomic and clinical data of 722 ER⁺ TCGA-BRCA samples. The tumor

purity and signature scores of immune cells between 12 MBC and 710 FBC were
compared using two-sided Wilcoxon rank-sum test, which was a non-parametric test
that did not assume known distributions (Hogg, R.V. and Tanis, E.A., Probability and
Statistical Inference, 7th Ed, Prentice Hall, 2006). We added description of the
statistical test in the revised Methods as follows (**Lines 725-728**): “The scores of
immune or stromal cells were compared between MBC and FBC samples using two-
sided Wilcoxon rank-sum test, which was a non-parametric test that did not assume
known distributions ⁶⁵.” Besides, the legend of this figure was revised to specify the
statistical test: “Boxplot showing the tumor purity and signature scores of various cell
types between MBC and FBC in TCGA ER⁺ BRCA cohort. P-value was calculated by
two-sided Wilcoxon rank-sum test.”

3. Statistical test for Figure 1i needed in figure legend.

Response: We apologize for the unclear legend. The two-sided Wilcoxon rank-sum test
was used to measure the differences between two groups. The legend was revised to
“Boxplot indicating the IHC scores of CD4 and CD8 in 30 ER⁺ male and 30 ER⁺ female
patients (identified by the percentage of positive cells). P-value was calculated by two-
sided Wilcoxon rank-sum test.”

4. Representative IHC for foxp3 positive staining appears to be nonspecifically stain
tumor cells (Figure 1h). The investigators perhaps should perform dual IF to
demonstrate the FOXP3 staining is confined to Treg cells (CD4+). The details of the
cohort in Figure 1 needs to be in the figure legend or text.

Response: Thank you for pointing this out. We agree with the reviewer that the IHC
staining of FOXP3 was not specific. It would be better to perform dual staining for both
CD4 and FOXP3 to identify the Tregs. However, these IHC staining results were used
to validate the significantly higher enrichment of T cells in FBC samples than in MBC
samples, which was observed in the single-cell data and bulk RNA-seq data.
Considering the Treg infiltration is not the concern of this context, we only retained the
IHC staining of CD4 and CD8 in the revised manuscript, as shown in **Response Figure**
**27** (Figure 2h in the revised manuscript). This validation cohort includes 30 ER⁺ MBC
and 30 ER⁺ FBC samples. The details of this cohort were described in the revised
manuscript as follows (**Lines 175-178**): “Furthermore, we performed
immunohistochemistry (IHC) analysis for T cell markers CD4 and CD8 in 30 ER⁺ MBC
and 30 ER⁺ FBC samples. Results suggested that these T cell markers had a lower
expression proportion in MBC than in FBC samples (**Figure 2h, i**)”. We also added the
corresponding description in the figure legend as “Boxplot indicating the IHC scores
of CD4 and CD8 in 30 ER⁺ male and 30 ER⁺ female patients (identified by the

percentage of positive cells). P-value was calculated by two-sided Wilcoxon rank-sum
test”.

5. What does IHC look like for FASN and AR in this cohort from Figure 1h?

Response: Thank you for your valuable suggestion. Accordingly, we performed the IHC
staining for FASN in the same cohort, including 30 ER⁺ MBC and 30 ER⁺ FBC samples.
Results showed that the protein levels of FASN were remarkably higher in MBC than
in FBC samples (Wilcoxon rank-sum test, p-value: 0.0052; **Response Figure 28**). We
added this result in the revised manuscript as follows (**Lines 229-234**): “Moreover, the
IHC staining for FASN in 30 ER⁺ MBC samples and 30 ER⁺ FBC samples were
compared. Results showed that the protein levels of FASN were remarkably higher in
MBC than in FBC samples (Wilcoxon rank-sum test, p-value: 0.0052; Figure 4e-f).
This observation indicated that fatty acids played an important role in tumor cell energy
metabolism in MBC patients.”

**Response Figure 28 (Related to Figure 4e-f in revised manuscript). Statistical quantification**
**of FASN staining in MBC and FBC.** (a) IHC images of MBC and FBC samples stained for FASN;
Scale bar, 20 μm. (b) Boxplot indicating the IHC score of FASN. P-value was calculated by two-
sided Wilcoxon rank-sum test.

Due to the absence of available qualified tissue samples, we are sorry that it is
unable to perform the IHC experiments for AR in this cohort. But alternatively, based
on the clinical diagnosis information, we retrospectively investigated the AR levels
evaluated by IHC in a large sample cohort, including 113 ER⁺ MBC and 86 ER⁺ FBC
samples (**Response Figure 29**). Results showed that the percentage of AR⁻ patients was
significantly lower in MBC than in FBC samples (5.3% vs. 17.4% in MBC and FBC
samples, respectively), whereas the percentage of AR⁺⁺⁺ patients was higher in MBC
than in FBC samples (69.9% vs. 50.0% in MBC and FBC samples, respectively). This
result further validated the activated AR regulon in MBC patients observed at the
single-cell level. We added this result in the revised manuscript as follows (**Lines 206-**
**213**): “To further evaluate the observation of AR, we retrospectively investigated the
AR levels evaluated by IHC in a large sample cohort, including 113 ER⁺ MBC and 86

ER⁺ FBC samples (**Figure 3i-j**). Results showed that the percentage of AR-negative
 patients was significantly lower in MBC than in FBC samples (5.3% vs. 17.4% in MBC
 and FBC samples, respectively), whereas the percentage of AR+++ patients was higher
 in MBC than in FBC samples (69.9% vs. 50.0% in MBC and FBC samples,
 respectively). This result further validated the activated AR regulon in MBC patients
 observed at the single-cell level.”

**Response Figure 29 (Related to Figure 3i-j in revised manuscript). The expression levels of AR**
 **in MBC and FBC samples. (a)** IHC images representing MBC and FBC samples stained for AR.
 Scale bar, 20 μm. **(b)** Barplot showing the percentage of AR-negative, AR+, AR++, and AR+++
 samples from MBC and FBC ER⁺ patients. P-value was calculated by fisher's exact test.

6. A hallmark of prostate cancer progression is dysregulation of lipid metabolism via
 overexpression of fatty acid synthase (FASN), a key enzyme in de novo fatty acid
 synthesis. Why was prostate cancer (PRAD) left out of the survival analysis stratified
 by FASN levels? Please include citation and discussion of targeting FASN in prostate
 cancer (PMID: 30578319).

**Response:** Thank you for your professional advice. In the previous version, we
 performed the analysis for overall survival (OS) of male and female patients with
 different cancer types and showed that FASN expression was prognostic for male
 patients but not for female patients in some cancers, such as bladder urothelial
 carcinoma (BLCA) and kidney renal clear cell carcinoma (KIRC). Prostate cancer
 (PRAD) was previously not included in the analysis due to the absence of female
 patients. In order to address the reviewer’s concern, we performed survival analyses for
 OS, disease-specific survival (DSS), and progression-free interval (PFI) of TCGA pan-
 cancer datasets by categorizing the patients into FASN-high and FASN-low groups for
 each dataset according to the median of FASN expression (**Response Figure 30-32**).
 Results showed that higher FASN expression was prognostic for the poor DSS of PRAD
 patients, suggesting the involvement of FASN in PRAD progression (**Response Figure**
 **31**). We have carefully read Zadra et al.’s study you recommended and are very inspired
 to know the association between FASN and AR signaling, as well as with the
 aggressiveness and resistance of PRAD (Zadra G, et al. *Proceedings of the National*
 *Academy of Sciences*, 2019). We added the results of survival analysis for PRAD in the
 revised manuscript as follows (**Lines 294-296**): “**Notably, higher FASN expression was**

prognostic for the poor DSS of PRAD patients, consistent with a previous study that
 demonstrated that targeting FASN could inhibit the aggressive and resistant PRAD²⁴.”
 In addition, we added discussion about the association between FASN and disease
 progression of patients with hormone-receptor-positive cancers as follows (Lines 497-
 502): “Notably, a previous study demonstrated that lipid metabolism dysregulation
 driven by FASN upregulation was important in the PRAD progression and castration
 resistance mediated by AR signaling²⁴. Our analysis also indicated the association
 between FASN expression and poor prognosis in PRAD. These results consistently
 suggested that FASN-mediated lipid metabolism dysregulation was a potential
 therapeutic target for hormone-receptor-positive cancers.”

 **Response Figure 30 (Related to Supplementary Figure 9 in revised manuscript). Overall**
 **survival analysis of male and female patients in various cancer types based on the FASN**
 **expression.** Patients are categorized into FASN-high and FASN-low groups for each dataset
 according to the median of FASN expression. BRCA: Breast invasive carcinoma; BLCA: Bladder
 Urothelial Carcinoma; KIRC: Kidney renal clear cell carcinoma; LAML: Acute Myeloid Leukemia;
 MESO: Mesothelioma; THCA: Thyroid carcinoma.

**Response Figure 31 (Related to Supplementary Figure 10 in revised manuscript). Disease-specific survival analysis of male and female patients in various cancer types based on the**
 **FASN expression. Patients are categorized into FASN-high and FASN-low groups for each dataset**
 **according to the median of FASN expression. BRCA: Breast invasive carcinoma; KIRC: Kidney**
 **renal clear cell carcinoma; KIRP: Kidney renal papillary cell carcinoma; THCA: Thyroid carcinoma;**
 **LUSC: Lung squamous cell carcinoma; PRAD: Prostate adenocarcinoma.**

**Response Figure 32 (Related to Supplementary Figure 11 in revised manuscript). Progression-**
**free interval analysis of male and female patients in various cancer types based on the FASN**
**expression.** Patients are categorized into FASN-high and FASN-low groups for each dataset
according to the median of FASN expression. BRCA: Breast invasive carcinoma; BLCA: Bladder
Urothelial Carcinoma; KIRC: Kidney renal clear cell carcinoma; KIRP: Kidney renal papillary cell
carcinoma; UVM: Uveal Melanoma

7. Supplementary Fig. 2 legend description inadequate. What fold change and
significance and testing performed?

Response: We apologize for the unclear description. To compare the characteristics of
cancer cells from MBC and FBC samples, we integrated cancer cells from 19 samples
and identified 36 clusters by unsupervised clustering. Using the MAST method with
default parameters in the Seurat package, we identified genes with log₂ fold change
greater than 0.25 and adjusted p-value less than 0.01 for each cluster. Based on the order
of log₂ fold change, the top 100 genes were further identified as markers of each cluster.
By calculating the proportion of cancer cells from MBC samples in each cluster, we
defined male, female, and mixed clusters. Specifically, clusters with a proportion of
male cancer cells higher than 70% were defined as male clusters, those with a
proportion lower than 50% were defined as female clusters, and the others were defined
as mixed clusters. To identify the genes specifically expressed in male clusters, gene
markers that presented in at least three male clusters were selected, and markers of
female or mixed clusters were further removed from this list. We re-phrased the
description of this procedure in the revised Method section, and revised the figure
legend as follows: “The expression levels of specifically expressed genes of male
cancer cell clusters. Genes with log₂ fold change greater than 0.25 and adjusted p-value
less than 0.01 for each cluster were identified using the MAST method with default
parameters. Gene markers that presented in at least three male clusters were selected,
and markers of female or mixed clusters were further removed from this list.”

8. Supplementary Fig. 4 legend needs more detail. How were FASN high and low
cutoffs determined?

Response: Thank your pointing this out. We categorized the patients into FASN-high
and FASN-low groups for each dataset according to the median of FASN expression.
The corresponding legend was revised as: “Overall survival analysis of male and female
patients in various cancer types based on the FASN expression. Patients are categorized
into FASN-high and FASN-low groups for each dataset according to the median of
FASN expression. BRCA: Breast invasive carcinoma; BLCA: Bladder Urothelial
Carcinoma; KIRC: Kidney renal clear cell carcinoma; LAML: Acute Myeloid
Leukemia; MESO: Mesothelioma; THCA: Thyroid carcinoma.”

9. The fact that FASN and the ER- and AR-response genesets were significantly
enriched by the up-regulated genes of “epithelial-T” co-expression cells, suggests that
there may be mixing of epithelial and T cell RNA in these dual positive cells. Therefore,
additional experiments are needed for the existence of “epithelial-T cells”. The authors
provide dual immunofluorescence (IF), however the staining in Figure 5B is
unconvincing. The legend states the scale bar is 50uM, but there is no scale bar and thus
hard to interpret. It is not clear whether the staining is from a single mitotic cell or many
cells at a distance. The DAPI does not even show uniform nuclear localization. The
staining appears to be an artifact. The researchers need to show additional validation of
the for IF using positive and negative control tissues. In addition, the investigators need
to quantify the CD3 only and epithelial T cells for the IF. The authors should also
provide another independent method to support their findings such as flow cytometry
(KRT and CD3) of dissociated T cells from fresh tumor tissue if possible.

**Response:** We appreciate the reviewer for highlighting this concerns. With the
development of single-cell techniques, we could investigate the cellular characteristics
at high resolution and identify the previously unappreciated cells. Intriguingly, a study
from Hu et al. also reported a non-traditional CD45⁺EpCAM⁺ cell population in the
fallopian tube epithelial layer of ovarian cancer patients (Hu et al., Cancer Cell,
2020, 37(2), 226-242). This population was also positive for CD3, CD44, CD69, and
CD103, suggesting that these cells are possibly tissue-resident memory T lymphocytes
(TRMs). They identified these cells by scRNA-seq (Smart-Seq2) and validated using
immunofluorescence experiments. However, the biological and clinical implications of
this populations are unclear yet. We are sorry for the unclear immunofluorescence
results in the previous version. We performed the immunofluorescence experiments
again and showed the cells with different phenotypes, including CD3⁺KRT8⁻, CD3⁻
KRT8⁺, and CD3⁺KRT8⁺ cells. According to the immunofluorescence, CD3⁺KRT8⁺
cells were located at the interface between KRT8⁺ epithelial cells and CD3⁺ T cells
(**Response Figure 33a**). Furthermore, flow cytometry for antibody of KRT8 and CD3
was performed using fresh tumor tissue from two MBC patients to validate and quantify
the number of CD3⁺KRT8⁺ cells (**Response Figure 33b**). We gated the CD45⁺ immune
cells and evaluated the expression of KRT8 of these cells. Results showed that there
were 35.55% and 2.11% CD45⁺KRT8⁺ cells in two samples, respectively. Notably,
57.07% and 20.82% of these KRT8⁺ immune cells were CD3⁺ T cells in two samples.
Thus, the immunofluorescence and flow cytometry experiments indicated that the
CD3⁺KRT8⁺ cells existed with various percentage in MBC samples. We added these
corresponding evidence in the revised manuscript as follows (**Lines 395-405**): “Further
validation using immunofluorescence experiments for the MBC sample confirmed the

above observation and showed that the CD3⁺KRT8⁺ cells were located at the interface
 between KRT8⁺ epithelial cells and CD3⁺ T cells (**Figure 6c**). Furthermore, flow
 cytometry for antibodies of KRT8 and CD3 was performed using fresh tumor tissue
 from two MBC patients to validate and quantify the number of CD3⁺KRT8⁺ cells
 (**Figure 6d**). We gated the CD45⁺ immune cells and evaluated the expression of KRT8
 in these cells. Results showed that there were 35.55% and 2.11% CD45⁺KRT8⁺ cells in
 two samples, respectively. Notably, 57.07% and 20.82% of these KRT8⁺ immune cells
 were CD3⁺ T cells in two samples. Therefore, these results indicated the biological
 existence of CD3⁺KRT8⁺ T cells and the enrichment of these cells with various
 percentages in MBC samples.”

a

b

**Response Figure 33 (Related to Figure 6c-d in revised manuscript). Validation of the existence**
**of CD3⁺KRT8⁺ T cells by the immunofluorescence and flow cytometry experiments. (a)** The
immunofluorescence staining of KRT8 and CD3 in an MBC sample. White arrows indicate the
CD3⁺KRT8⁺ T cells. Scale bar, 50 μ m. **(b)** Flow cytometry showing the percentage of CD3⁺KRT8⁺
cells in two MBC samples.

10. Supplemental Fig. S6a is described as differentially expressed gene across five
samples. What are the individual values? Aggregated expression of all the single cells
for each tumor? Perhaps showing the expression of KRT8/18/19 and CD3 across all
cells annotated by cell type for each tumor would be more convincing for the existence
of an epithelial T-cell. This will show the relative KRT levels in true epithelial cells
relative to the T cells.

**Response:** We agree with the reviewer about this concern. It is important to show the
expression of epithelial markers and T cell markers across all cell types. Accordingly,
we re-clustered the cells from each sample and then visualized all cell types and marker
expressions at the single-cell level. MBC and FBC samples with high percentage of
CD3⁺KRT⁺ cells were shown in **Response Figure 34 and 35**. To further evaluate the
expression of KRT8/18/19 in T cells, we also showed the aggregated expression of these
markers of epithelial and T cells in each sample using the dot-plot (**Response Figure**
**36**). The T cells from MBC2, MBC3, MBC4, MBC5, MBC6, and FBC13 had
KRT8/18/19 expression, but were lower than these levels in epithelial cells. We added
these corresponding results in the revised manuscript as follows (**Lines 368-375**): “**We**
**re-clustered the cells from each sample and then visualized all cell types and marker**
**expressions at the single-cell level. MBC and FBC samples with the highest percentage**
**of CD3⁺KRT⁺ cells were shown in Supplementary Figure 13e, f. To further evaluate**
**the expression of KRT8/18/19 in T cells, we also showed the aggregated expression of**
**these markers of epithelial and T cells in each sample using the dot-plot**
**(Supplementary Figure 13g). The T cells from MBC2, MBC3, MBC4, MBC5, MBC6,**
**and FBC13 had KRT8/18/19 expression, but were lower than these levels in epithelial**
**cells.”**

Response Figure 34 (Related to Supplementary Figure 13e in revised manuscript). The expression of CD3E and KRT8 in various cell types from representative MBC samples. CD3E⁺KRT⁺ T cells were circled with dashed lines.

Response Figure 35 (Related to Supplementary Figure 13f in revised manuscript). The expression of CD3E and KRT8 in various cell types from representative FBC samples. CD3E⁺KRT⁺ T cells were circled with dashed lines.

Response Figure 36 (Related to Supplementary Figure 13g in revised manuscript). Dotplot depicting aggregated expression of KRT8/KRT18/KRT19 and CD3D/CD3E/CD3G in epithelial and T cells from MBC and FBC samples.

11. Supplemental Fig. 6b shows the percentage of T cells that express KRT (epithelial-
T cells) is around 40%, and similar in Fig 5C, however in Fig 5A there it appears that
nearly all cells co-expressed CD3 and KRTs. What are the proportions of epithelial T
cells in the other MBCs and FBCs or is this just an occurrence in the M3 tumor?

**Response:** We thank the reviewer for highlighting this issue. The previous Figure 5A
showed the expression of CD3 and KRT8 by taking MBC3 as an example, which had
the highest percentage (83.1%) of CD3-KRT8 co-expressed cells comparing with other
samples. Thus, due to the high enrichment of CD3-KRT8 co-expressed cells in MBC3,
it visually seems that nearly all cells co-expressed CD3 and KRTs in that figure. The
Supplementary Figure 6b in the previous version showed the percentage of co-
expressed cells in all MBC samples, around 40%. We apologize for the unclear
description and visualization in the previous version. In order to figure out whether the
observed CD3E⁺KRT8⁺ T cells were patient-specific or generally existed, we evaluated
the percentage of CD3E⁺KRT8⁺ T cells using the updated datasets, including 6 in-house
MBC samples, 2 in-house FBC samples, and 11 FBC samples from Wu et al.. It turned
out that 17/19 breast cancer samples had CD3E⁺KRT8⁺ T cells with different degree,
ranging from 0.2% to 83.1% (**Response Figure 37a**). Especially, MBC samples showed
higher percentage of CD3E⁺KRT8⁺ T cell component (6.7% ~ 83.1%), and FBC
samples had relatively lower percentage (0.2% ~ 17.9%). The wilcoxon rank sum test
showed a significant difference of CD3E⁺KRT8⁺ T cell enrichment between MBC and
FBC groups (**Response Figure 37b**; p-value: 0.0014). We added this results in the
revised manuscript (**Lines 361-377**) as follows: “In order to figure out whether the
observed CD3E⁺KRT8⁺ T cells were patient-specific or generally existed, we evaluated
the percentage of CD3E⁺KRT8⁺ T cells across 19 samples, including 6 in-house MBC
samples, 2 in-house FBC samples, and 11 FBC samples from Wu et al.. It turned out
that 17/19 breast cancer samples had CD3E⁺KRT8⁺ T cells with different degrees,
ranging from 0.2% to 83.1% (Supplementary Figure 13d). Especially, MBC samples
showed higher percentage of CD3E⁺KRT8⁺ T cell component (6.7% ~ 83.1%), and
FBC samples had relatively lower percentage (0.2% ~ 17.9%). We re-clustered the cells
from each sample and then visualized all cell types and marker expressions at the
single-cell level. MBC and FBC samples with the highest percentage of CD3⁺KRT⁺
cells were shown in Supplementary Figure 13e, f. To further evaluate the expression of
KRT8/18/19 in T cells, we also showed the aggregated expression of these markers of
epithelial and T cells in each sample using the dot-plot (Supplementary Figure 13g).
The T cells from MBC2, MBC3, MBC4, MBC5, MBC6, and FBC13 had KRT8/18/19
expression, but were lower than these levels in epithelial cells. The Wilcoxon rank-sum
test showed a significant difference of CD3E⁺KRT8⁺ T cell enrichment between MBC

and FBC groups (Supplementary Figure 13h; p-value: 0.0014).”

**Response Figure 37 (Related to Supplementary Figure 13d, h in revised manuscript).**
**Evaluation of the existence of CD3E+KRT8+ T cells in the scRNA-seq dataset. (a)** Barplot
showing the percentage of CD3E+KRT8+ T cells in each MBC and FBC sample. **(b)** Boxplot
comparing the percentage of CD3E+KRT8+ T cells between MBC and FBC samples. P-value was
calculated by two-sided Wilcoxon rank-sum test.

12. The authors should consider evaluating several other scRNA breast cancer datasets
for evidence of epithelial T cells.

**Response:** We agree with the reviewer’s concern regarding the evaluation of
CD3E+KRT8+ cells in other scRNA dataset. To further validate the existence of these
cells, we downloaded and performed an integrated analysis for the scRNA-seq data of
ER+ BRCA from the previous study (Wu et al., *Nature Genetics*, 2021), in which all the
samples were from female patients. By integrating the transcriptomic data of T cells
from in-house and Wu et al. (**Response Figure 38a-b**), we calculated the percentage of
CD3E+KRT8+ T cells of in-house MBC, in-house FBC, and Wu’s FBC samples,
respectively. Results showed that MBC samples had a significantly higher percentage
of CD3E+KRT8+ T cells than the FBC samples from two datasets. Besides, the
percentages of CD3E+KRT8+ T cells were similar in in-house and Wu et al.’s FBC
samples(**Response Figure 39**), suggesting the existence of CD3E+KRT8+ T cells and
the enrichment of these cells in male samples. We added this results in the revised
manuscript (**Lines 355-361**) as follows: “To further validate the existence of these cells,
we calculated the percentage of CD3E+KRT8+ T cells of in-house MBC, in-house FBC,
and Wu et al.’s FBC samples, respectively (**Supplementary Figure 13a, b**). Results
showed that the percentages of CD3E+KRT8+ T cells were similar in in-house and Wu
et al.’s FBC samples (**Supplementary Figure 13c**). MBC samples had a significantly
higher percentage of CD3E+KRT8+ T cells than the FBC samples from the two datasets
(**Supplementary Figure 13c**).”

Response Figure 38 (Related to Supplementary Figure 13a-b in revised manuscript). Evaluation of the existence of CD3E⁺KRT8⁺ T cells in the scRNA-seq dataset. (a) T-SNE plot of T cells colored by data sources. (b) T-SNE plots showing the distribution of CD3E⁺KRT8⁻ and CD3E⁺KRT8⁺ T cells in in-house MBC samples (left), in-house FBC samples (middle), and FBC samples from Wu et al. (right).

Response Figure 39 (Related to Supplementary Figure 13c in revised manuscript). Barplot showing the percentage of CD3⁺KRT8⁺ T cells in different datasets.

13. Data availability section is weak, and data are not publicly deposited (this can be
blinded until publication but available for reviewers).

**Response:** We appreciate your comment and apologize for forgetting to include the
access link. The single-cell RNA-seq data of this study have been deposited in Genome
sequence Archive database with accession number HRA001341. Reviewers can access
these sequence files via the link: <https://ngdc.cncb.ac.cn/gsa-human/s/Mv4xF4IP>. The
data will be publicly accessed after the publication of this study.

14. The authors should consider evaluating the role of AR in MBC in more detail. Such
as performing IHC on specimens, evaluating the RNA-seq for existence to alternative
splicing in the androgen receptor.

**Response:** We thank the reviewer for this helpful suggestion. Accordingly, we
retrospectively investigated the AR levels evaluated by IHC in a large sample cohort,
including 113 ER⁺ MBC and 86 ER⁺ FBC samples (**Response Figure 29**). Results
showed a significantly higher percentage of ER⁺ samples in MBC than in FBC (Fisher's
exact test, p-value: 0.00025). We added this result in the revised manuscript as follows
(**Lines 206-213**): "To further evaluate the observation of AR, we retrospectively

investigated the AR levels evaluated by IHC in a large sample cohort, including 113
ER⁺ MBC and 86 ER⁺ FBC samples (**Figure 3i-j**). Results showed that the percentage
of AR-negative patients was significantly lower in MBC than in FBC samples (5.3%
vs. 17.4% in MBC and FBC samples, respectively), whereas the percentage of AR+++
patients was higher in MBC than in FBC samples (69.9% vs. 50.0% in MBC and FBC
samples, respectively). This result further validated the activated AR regulon in MBC
patients observed at the single-cell level.”

In addition, inspired by the previous studies that demonstrated the fatty acid
metabolism driven by AR in PRAD (Zadra G, et al. *Proceedings of the National*
*Academy of Sciences*, 2019; Swinnen J V, et al. *Cancer research*, 1997), we investigated
the association between AR and FASN expression in breast cancer samples (**Response**
**Figure 40**). Results showed that the expression of AR and FASN had significantly
positive correlations in MBC samples from GSE31259 and GSE104730, but had no
obvious correlations in the TCGA dataset possibly due to the limited number of MBC
samples in TCGA. We added this result in the revised manuscript as follows (**Lines**
**247-252**): “In addition, inspired by the previous studies that demonstrated the fatty acid
metabolism driven by AR in PRAD^{24,25}, we investigated the association between AR
and FASN expression in MBC and FBC samples (**Figure 4i**). Results showed that the
expressions of AR and FASN were positively correlated in MBC samples from
GSE104730⁶ and GSE31259²⁰, but had no obvious correlations in the FBC samples of
the TCGA dataset.”

**Response Figure 40 (Related to Figure 4i in revised manuscript). The Pearson correlation**
**analysis between the expression of FASN and AR in independent breast cancer datasets.**

Although alternative splicing in AR has been reported to play an important role in
the progression and resistance of PRAD (Antonarakis E S, et al. *New England Journal*
*of Medicine*, 2014; Zadra G, et al. *Proceedings of the National Academy of Sciences*,
2019), the existence of AR alternative splicing in MBC remains unexplored. However,
using the 10X Genomics Chromium (10X) approach, our scRNA-seq data capture
transcripts through poly(A) tails and have 3'-bias in coverage, limiting the capability of
performing the alternative splicing analysis on the single-cell level (Wang X, et al.

*Genomics, proteomics & bioinformatics*, 2021). We analyzed the alternative splicing
events of ER⁺ TCGA-BRCA samples based on the data from TCGASpliceSeq database
(Ryan M, et al. *Nucleic acids research*, 2016). Result showed that there was no
significant differences in the expression of AR isoforms between ER⁺ MBC and FBC
samples in the TCGA dataset (**Response Figure 41**).

Response Figure 41. The expression of AR splicing isoforms between MBC and FBC samples in TCGA ER+ dataset. AP: Alternate Promoter; ES: Exon Skip.

15. The authors need more detail in most figure legends. It is sometimes hard to
1470 interpret the data. For example, Figure 2g and h show expression and activation of
1471 transcription factors, but what cell types were evaluated (just epithelial)? There appears
to be a bimodal distribution in these blots suggesting there the cells are either in an on
or off state. It would be interesting to see what cells are on vs. off.

Response: We thank reviewer for highlighting these concerns. We apologize for the
1475 unclear figure legends. The previous Figure 2 (Figure 3 in the revised manuscript) had
1476 shown the comparison analysis of transcriptome of cancer cells from MBC and FBC
samples. Thus, all the plots in this Figure showed the results of cancer (epithelial) cells.

We revised the legend as follows: “Heatmap showing the activity scores of transcription
factors (TFs) in cancer cells from male, female, and mixed clusters” and “Ridgeline
plot showing the activity levels of MBC-specific TFs in cancer cells from male, female,
and mixed clusters”. Using the updated dataset, we performed the transcriptional
regulon analysis for cancer cells from six MBC samples and thirteen FBC samples.
Results showed that the distribution of AR and SREBF1 activity was not obviously
bimodal (**Response Figure 42**).

Response Figure 42 (Related to Figure 3g in revised manuscript). Ridgeline plot showing the activity levels of MBC-specific TFs in cancer cells from male, female, and mixed clusters.

Response References

1. Wu S Z, Al-Eryani G, Roden D L, et al. A single-cell and spatially resolved atlas of human breast cancers[J]. *Nature genetics*, 2021, 53(9): 1334-1347.

2. Hogg, R.V. and Tanis, E.A., *Probability and Statistical Inference*, 7th Ed, Prentice Hall, 2006

3. Zadra G, Ribeiro C F, Chetta P, et al. Inhibition of de novo lipogenesis targets androgen receptor signaling in castration-resistant prostate cancer[J]. *Proceedings of the National Academy of Sciences*, 2019, 116(2): 631-640.

4. Hu Z, Artibani M, Alsaadi A, et al. The repertoire of serous ovarian cancer non-genetic heterogeneity revealed by single-cell sequencing of normal fallopian tube epithelial cells[J]. *Cancer Cell*, 2020, 37(2): 226-242.

5. Swinnen J V, Esquenet M, Goossens K, et al. Androgens stimulate fatty acid synthase in the human prostate cancer cell line LNCaP[J]. *Cancer research*, 1997, 57(6): 1086-1090.

6. Wang X, He Y, Zhang Q, et al. Direct comparative analyses of 10X genomics chromium and smart-seq2[J]. *Genomics, proteomics & bioinformatics*, 2021, 19(2): 253-266.

7. Ryan M, Wong W C, Brown R, et al. TCGASpliceSeq a compendium of alternative mRNA splicing in cancer[J]. *Nucleic acids research*, 2016, 44(D1): D1018-D1022.

8. Antonarakis E S, Lu C, Wang H, et al. AR-V7 and resistance to enzalutamide and abiraterone in prostate cancer[J]. *New England Journal of Medicine*, 2014, 371(11): 1028-1038.

REVIEWER COMMENTS

Reviewer #1 (Remarks to the Author):

The authors addressed adequately my comments and improved considerably the manuscript.

I still have a few minor questions and remarks:

1. Regarding the existence of CD3D+/KRT8+ cells, did the authors try to use a software such as CellBender to decontaminate the raw UMI matrices and further investigate if this specific population of cells still remain in the resulting clustering ?
2. It seems that the authors are using CD79A to identify B-cells. However, CD79A is expressed by both B and plasma cells, and two different clusters are visible within the umap. Did the authors try to use MS4A1 and JCHAIN to identify B-cells and plasma cells respectively ?
3. A similar remark can be done for endothelial cells. Did the authors tried to discriminate between vascular and lymphatic endothelial cells using specific markers ?
4. The authors used Dorothea to analyze regulon activity (<https://bioconductor.org/packages/release/data/experiment/vignettes/dorothea/inst/doc/dorothea.html>). However, it seems from the documentation that Dorothea is a database of regulons used as input for other statistical methods such as decoupleR or SCENIC. Did the authors used such packages to analyze regulons activity ?

Reviewer #2 (Remarks to the Author):

The responses are extensive and extremely helpful. No further comments.

Reviewer #3 (Remarks to the Author):

The authors have made substantial improvements to the manuscript and the additional experimentation and analysis have cleared up all my concerns but one. The evidence for KRT8+ CD3 T cells is still insufficient for conclusively identifying this novel cell type. The authors have added single cell analysis (Supplemental figs 13e and f), however, but “KRT8/18/19 expression, but were lower than these levels in epithelial cells.” Since there are no scale bars, and min/max should be different for each gene analyzed, it is hard to determine if these CD3 cells are truly KRT8 positive or that the scales are set so low that all cells have some KRT8 positivity.

The new IF images (Figure 6C) show that the only KRT18 positive T cells are at the interface with the tumor cells. In fact, the cell in the top arrow is clearly KRT8 positive, but does not appear to be CD3 positive and is likely an exposure artefact. The additional cells identified appear to be sectioning artefacts in which multiple layers of cells are stained. I would anticipate that some CD3 cells away from the tumor would have to be KRT8 positive to truly identify this novel cell type.

The flow cytometry in Response Figure 8 is also unconvincing because it lacks a positive control for KRT8 (epithelial cells). Additionally, the gate in upper panel of response 8b appears to be loosely gated for CD45 and potentially including CD45 negative cells. Clearly all the T cells (CD3D) in figure 1B are negative for KRT8 and I suspect KRT18 as well. In Figure 5 F, all the male T cells are either KRT18 or KRT8 positive, which is different than figure 1B.

These discrepancies need to be addressed to conclusively state that KRT8 positive T cell exist.

Reviewer #1 (Remarks to the Author):

The authors addressed adequately my comments and improved considerably the manuscript. I still have a few minor questions and remarks:

1. Regarding the existence of CD3D+/KRT8+ cells, did the authors try to use a software such as CellBender to decontaminate the raw UMI matrices and further investigate if this specific population of cells still remain in the resulting clustering?

Response: Thank you for your valuable suggestion. Accordingly, we used CellBender[1] to decontaminate the in-house scRNA-seq data, of which the raw UMI matrices were available. After removing the empty droplets and retrieving background-free gene expression profiles by CellBender, we found that CD3E⁺KRT8⁺ cells still existed in all samples (**Response Figure 1**), keeping consistent with the previous results based on Cell Ranger. This result double-confirmed the existence of CD3E⁺KRT8⁺ cells and avoided the potential influence of technical contamination. We added the corresponding result in the revised manuscript as follows (**Lines 402-408**): “Moreover, we used CellBender³⁶ to decontaminate the in-house scRNA-seq data, of which the raw UMI matrices were available. After removing the empty droplets and retrieving background-free gene expression profiles, we found that CD3E⁺KRT8⁺ cells still existed in all samples (**Supplementary Figure 15c**), keeping consistent with the results based on Cell Ranger (**Supplementary Figure 14a**). This result double-confirmed the existence of CD3E⁺KRT8⁺ cells and avoided the potential influence of technical contamination.”

Response Figure 1 (Related to Supplementary Figure 14a and 15c in the revised manuscript). Evaluation of the existence of CD3E⁺KRT8⁺ T cells in the scRNA-seq dataset. (a) Barplot showing the percentage of CD3E⁺KRT8⁺ T cells in each MBC and FBC sample. (b) Barplot showing the percentage of CD3E⁺KRT8⁺ T cells in in-house samples after decontamination analysis by CellBender.

2. It seems that the authors are using CD79A to identify B-cells. However, CD79A is expressed by both B and plasma cells, and two different clusters are visible within the umap. Did the authors try to use MS4A1 and JCHAIN to identify B-cells and plasma cells respectively?

Response: We didn't notice this problem before. Many thanks for the reviewer's suggestion. Accordingly, we evaluated the expression levels of *MS4A1* and *JCHAIN* in each single-cell cluster. It turned out that the smaller B/plasma cluster (1356 cells) had a specific *JCHAIN* expression, and was annotated as plasma cluster; the larger B/plasma cluster (2292 cells) had specific *MS4A1* expression and was annotated as B cluster (**Response Figure 2**). The corresponding figures and text were modified in the revised manuscript (*please also see the response to the relevant Comment #3*).

Response Figure 2 (Related to Figure 1b in the revised manuscript). Identification of B cells and Plasma cells.

3. A similar remark can be done for endothelial cells. Did the authors tried to discriminate between vascular and lymphatic endothelial cells using specific markers?

Response: According to the reviewer's suggestion, we annotated the endothelial cell clusters in detail by using the following marker genes: arterial endothelial cells (*GJA5* and *BMX*), venous endothelial cells (*SELE*, *ACKR1*, and *SELP*), capillary endothelial cells (*PLVAP* and *RAMP3*), and lymphatic endothelial cells (*PDPN* and *PROX1*). Results showed that the endothelial cell clusters could be further categorized into arterial endothelial cells (**Response Figure 3**), venous endothelial cells (**Response Figure 4**), and capillary endothelial cells (**Response Figure 5**). But we did not identify the lymphatic endothelial cells in our data (**Response Figure 6**). We updated the feature-plots and cell type annotations in Figure 1 of the revised manuscript (**Response Figure 7**). Accordingly, the text was updated in the revised manuscript as follows (**Lines 133-137**): “By analyzing the expression of marker genes, we annotated the

various cell types in the BRCA ecosystem, including epithelial cells, T cells, B cells, plasma cells, macrophages, mast cells, myofibroblasts, cancer-associated fibroblasts (CAFs), arterial endothelial cells, venous endothelial cells, and capillary endothelial cells (**Figure 1b, c and Supplementary Table 3**).”

Response Figure 3 (Related to Figure 1b in the revised manuscript). Identification of arterial endothelial cells.

Response Figure 4 (Related to Figure 1b in the revised manuscript). Identification of venous endothelial cells.

Response Figure 5 (Related to Figure 1b in the revised manuscript). Identification of capillary endothelial cells.

Response Figure 6. Identification of lymphatic endothelial cells.

Response Figure 7 (Related to Figure 1 in the revised manuscript). Cell type annotations of scRNA-seq data of breast cancer patients. (a) The expression of marker genes of each cell type. (b) The t-distributed stochastic neighbor embedding (t-SNE) plots of cells types and resources profiled in this study. (c) Heatmap showing genes (columns) that were differentially expressed across various cell types (rows).

4. The authors used Dorothea to analyze regulon activity (<https://bioconductor.org/packages/release/data/experiment/vignettes/dorothea/inst/doc/dorothea.html>). However, it seems from the documentation that Dorothea is a database of regulons used as input for other statistical methods such as decoupleR or SCENIC. Did the authors use such packages to analyze regulons activity?

Response: Thank you for pointing this out. We apologize for not providing a detailed description of the version of the R package we used. We analyzed the regulon activity by using the R package Dorothea (version 1.72), which combined the database of regulons and TF activity inference methods together in one single package. A detailed description of this package can be found in the corresponding publication “Luz Garcia-Alonso, et al., Benchmark and integration of resources for the estimation of human transcription factor activities, *Genome Research*, 2019, 29(8): 1363-1375. doi: 10.1101/gr.240663.118” [2]. In April 2022, the developer of Dorothea uncoupled this R package into two parts, one is for the regulon database (version 1.8 of Dorothea), and the other is for TF activity inference (decoupleR[3]). Based on the TF regulons curated in the Dorothea database, we calculated the regulon activity using VIPER[4], a statistical method included in Dorothea (version 1.72). Besides, only regulons with confidence levels A, B, and C were selected to better estimate TF activities. We updated the description of this analysis process in the revised Method section as follows (**Lines 659-663**): “We analyzed the regulon activity by using the R package Dorothea (version 1.72)⁶², which combined the database of regulons and TF activity inference methods together. Only regulons with confidence levels A, B, and C were selected to better estimate TF activities. Regulon score was calculated for each single cell using VIPER⁶³, a statistical test based on the average ranks of the targets.”

Reviewer #2 (Remarks to the Author):

The responses are extensive and extremely helpful. No further comments.

Reviewer #3 (Remarks to the Author):

1. The authors have made substantial improvements to the manuscript and the additional experimentation and analysis have cleared up all my concerns but one. The evidence for KRT8⁺ CD3 T cells is still insufficient for conclusively identifying this novel cell type. The authors have added single cell analysis (Supplemental figs 13e and f), however, but “KRT8/18/19 expression, but were lower than these levels in epithelial cells.” Since there are no scale bars, and min/max should be different for each gene analyzed, it is hard to determine if these CD3 cells are truly KRT8 positive or that the scales are set so low that all cells have some KRT8 positivity.

Response: Thank you for the professional comment. We apologize for missing the detailed scale bars in the previous Supplementary Figure 13e, f. The CD3 and KRT8 expression feature-plots in MBC3 and FBC13 were updated with identical scale bars in the revised figures (**Response Figure 8**). The expression level of KRT8 in T cells intuitively seemed to be lower than in epithelial cells, due to hundreds of cells overlapping in the feature-plot. The visualization of feature-plots could be affected by the proportion of cells with KRT8 expression. Actually, almost all epithelial cells are KRT8⁺ (red in feature-plot; 99.4% in MBC3 and 86.2% in FBC13), but only a part of T cells are KRT8⁺ (82.8% in MBC3 and 16.7% in FBC13; **Response Figure 9**), resulting in the overlapping of KRT8⁺ (red) and KRT8⁻ (grey) cells in the area of T cell cluster. In order to illustrate the expression of KRT8 more clearly, we split the feature-plot into two separate parts based on whether KRT8 was positive in T cells (**Response Figure 10**). Violin plots were used to further statistically compare the KRT8 expression among epithelial cells, KRT8⁺ T cells, and KRT8⁻ T cells. Results showed that KRT8⁺ T cells had a similar or lower level of KRT8 expression as epithelial cells (**Response Figure 11**).

More important, to verify the existence of KRT8⁺ T cells, we evaluated the expression of CD3 and KRT8 in MBC samples by performing immunofluorescence

staining and flow cytometry according to the reviewer's comment #2 and #3 (*Please refer to the response to comment #2 and #3 as well*).

We added these results in the revised Supplementary Figure 14, and the corresponding description in the revised manuscript as follows (**Lines 375-383**): “MBC and FBC samples with the highest percentage of CD3E⁺KRT8⁺ cells were shown in Supplementary Figure 14b, c. Because only a part of T cells were KRT8⁺ (Supplementary Figure 14d, e), we split the feature-plot into two separate parts based on whether KRT8 was positive in T cells to clearly illustrate the expression of KRT8. We found that some T cells did express KRT8 but others had no expression (Supplementary Figure 14f, g). Violin plots were used to further statistically compare the KRT8 expression levels among epithelial cells, KRT8⁺ T cells, and KRT8⁻ T cells, suggesting that KRT8⁺ T cells had a similar or lower level of KRT8 expression compared with epithelial cells (Supplementary Figure 14h, i)”.

Response Figure 8 (Related to Supplementary Figure 14b, c in the revised manuscript). T-SNE plots showing the cell types, and expression of KRT8 and CD3E in MBC3 (a) and FBC13 (b). T cells were circled with dashed lines.

Response Figure 9 (Related to Supplementary Figure 14d, e in the revised manuscript). Barplots showing the percentage of KRT8⁺ epithelial cells and T cells in MBC3 (left) and FBC13 (right).

Response Figure 10 (Related to Supplementary Figure 14f, g in the revised manuscript). T-SNE plots showing the KRT8 expression of epithelial cells, KRT8⁺ T cells, and KRT8⁻ T cells in MBC3 (upper panel) and FBC13 (bottom panel). T cells were circled with dashed lines. The feature-plots were split into two separate parts based on whether KRT8 was positive in T cells.

Response Figure 11 (Related to Supplementary Figure 14h, i in the revised manuscript). Violin plots showing the expression of KRT8 among epithelial cells, KRT8⁺ T cells and KRT8⁻ T cells in MBC3 (left) and FBC13 (right).

2. The new IF images (Figure 6C) show that the only KRT18 positive T cells are at the interface with the tumor cells. In fact, the cell in the top arrow is clearly KRT8 positive, but does not appear to be CD3 positive and is likely an exposure artefact. The additional cells identified appear to be sectioning artefacts in which multiple layers of cells are stained. I would anticipate that some CD3 cells away from the tumor would have to be KRT8 positive to truly identify this novel cell type.

Response: We agree with the reviewer that providing solid evidence for the existence of CD3⁺KRT8⁺ cells is essential. According to your suggestion, we found some instances of CD3⁺KRT8⁺ cells that were away from tumor or T cells to avoid the exposure artifact, although finding these cells is challenging due to the relatively low proportion (*please also see the response to the relevant Comment #3*). As shown in **Response Figure 12 and 13**, the CD3⁺KRT8⁺ cells were not necessarily to be located at the interface between tumor cells and T cells. In order to further avoid the artifacts from multiple layers of cells, we obtained a series of Z-stack confocal images of one single CD3⁺KRT8⁺ cell with a confocal microscope (CarlZeiss LSM880 with NLO & Airyscan; **Response Figure 14**). We hope that these images could prove the existence of CD3⁺KRT8⁺ cells in MBC samples. These results were added in the Figure 6c and Supplementary Figure 16 in the revised manuscript. The corresponding description was revised as follows (**Lines 413-417**): “Further validation using immunofluorescence experiments for the MBC sample confirmed the above observation and showed the

existence of CD3⁺KRT8⁺ cells (Figure 6c and Supplementary Figure 16a). In order to avoid the artifacts from multiple layers of cells, we further obtained a series of Z-stack confocal images of one single CD3⁺KRT8⁺ cell with a confocal microscope (Supplementary Figure 16b)’’.

Notably, recent literature has reported cumulative evidence for the existence of cells co-expressing T-cell and epithelial-cells markers in various tissues [5-9]. For example, Hu et al.[8] reported a non-traditional CD45⁺EpCAM⁺ cell population in the fallopian tube epithelial layer of ovarian cancer patients (Hu et al., Cancer cell, 2020, 37(2), 226-242). This population was also positive for CD3, CD44, CD69, and CD103, suggesting that these cells are possibly tissue-resident memory T lymphocytes (TRMs). They identified these cells by scRNA-seq (Smart-Seq2) and validated them using immunofluorescence experiments. Besides, using scRNA-seq, flow cytometry, cell co-culture experiments, RNA-FISH, and immunofluorescent staining, another study from Chen et al.[9] reported that infiltrated CD8⁺ effector T cells expressed tumor markers in prostate cancer (Chen et al., Nature cell biology, 2021, 23(1): 87-98). They also demonstrated that these cells were induced by the extracellular vesicle (EV) derived from prostate tumor cells, and associated with the micrometastases. Moreover, other studies also found cells expressing both epithelial and immune cell markers in ovarian carcinoma and colon adenocarcinoma, etc.[5-7]. Our observation of CD3⁺KRT8⁺ cells in MBC will be an important complement to these previous findings that characterized this non-traditional cell type.

Response Figure 12 (Related to Figure 6c in the revised manuscript). The immunofluorescence staining of KRT8 and CD3 in an MBC sample. White arrow indicates the CD3⁺KRT8⁺ T cell. Scale bar, 20 μm.

Response Figure 13 (Related to Supplementary Figure 16a in the revised manuscript). The immunofluorescence staining of KRT8 and CD3 in an MBC sample. White arrow indicates the CD3⁺KRT8⁺ T cell. Scale bar, 20 μm.

Response Figure 14 (Related to Supplementary Figure 16b in the revised manuscript). Z-stack confocal images of one CD3⁺KRT8⁺ T cell from an MBC sample. Scale bar, 5 μm. The interval for Z-stack was 0.71 μm.

3. The flow cytometry in Response Figure 8 is also unconvincing because it lacks a positive control for KRT8 (epithelial cells). Additionally, the gate in upper panel of response 8b appears to be loosely gated for CD45 and potentially including CD45 negative cells.

Response: Thank you for pointing this out. Accordingly, we re-analyzed the flow cytometry data to show the epithelial cells, T cells and KRT8⁺CD45⁺CD3⁺ cells in two MBC samples (**Response Figure 15**). Firstly, KRT8 and CD45 were used to distinguish the epithelial cells (KRT8⁺CD45⁻), immune cells (KRT8⁻CD45⁺), and KRT8⁺CD45⁺ cells. There were 8.7% and 3.76% KRT8⁺CD45⁺ cells, in which 33.4% and 35% were

CD3⁺ T cells in MBC-7 and MBC-8, respectively. Among all T cells, KRT8⁺ T cells accounted for 41.4% and 6.8% in MBC-7 and MBC-8, respectively. Therefore, these results indicated the biological existence of KRT8⁺CD45⁺CD3⁺ T cells and the enrichment of these cells with various percentages in different MBC samples. This result was updated in the revised Figure 6d. The corresponding description was revised as follows (Lines 418-426): “Besides, we performed flow cytometry experiments for fresh tumor tissues from two MBC patients to validate and quantify the number of CD3⁺KRT8⁺ cells (Figure 6d). Firstly, KRT8 and CD45 were used to distinguish the epithelial cells (KRT8⁺CD45⁻), immune cells (KRT8⁻CD45⁺), and KRT8⁺CD45⁺ cells. There were 8.7% and 3.76% KRT8⁺CD45⁺ cells, in which 33.4% and 35% were CD3⁺ T cells in MBC-7 and MBC-8, respectively. Among all T cells, KRT8⁺ T cells accounted for 41.4% and 6.8% in MBC-7 and MBC-8, respectively. Therefore, these results indicated the biological existence of KRT8⁺CD45⁺CD3⁺ T cells and the enrichment of these cells with various percentages in different MBC samples”.

Response Figure 15 (Related to Figure 6d in the revised manuscript). Flow cytometry analysis showing the identification of epithelial cells (KRT8⁺CD45⁻), T cells (KRT8⁻CD45⁺CD3⁺) and KRT8⁺CD45⁺CD3⁺ cells in two MBC samples.

4. Clearly all the T cells (CD3D) in figure 1B are negative for KRT8 and I suspect KRT18 as well. In Figure 5 F, all the male T cells are either KRT18 or KRT8 positive, which is different than figure 1B. These discrepancies need to be addressed to conclusively state that KRT8 positive T cell exist.

Response: We apologize for the confusing visualization of Figure 5F in the previous submission and appreciate the reviewer's comment. Because as many as 100 differentially expressed genes (including 50 up-regulated genes and 50 down-regulated genes) were included in the heatmap, it is impossible to show the names of all genes in Figure 5F due to the limited space. Thus, only some representative gene names were shown beside the heatmap. Maybe the inexact pointing of KRT8/KRT18/KRT19 in the previous submission caused the misunderstanding of the proportion of MBC T cells with positive expression. We showed the original heatmaps for CD8⁺, CD4⁺, and NKT cells, with all gene names being displayed in **Response Figures 16, 17, and 18**. These figures showed that nearly half of MBC T cells were positive for KRT8/KRT18/KRT19 expression. Accordingly, we corrected the pointing of gene names and tried our best to make sure that the representative genes were exactly pointed beside the corresponding rows in the revised Figure 5F (**Response Figure 19**). Besides, the whole list of differentially expressed genes between T cells from MBC and FBC samples were shown in **Supplementary Tables 7, 8, and 9**.

Response Figure 16 (Related to Supplementary table 7 in the revised manuscript). Heatmap showing the differentially expressed genes between MBC and FBC CD8⁺ T cells.

Response Figure 17 (Related to Supplementary table 8 in the revised manuscript). Heatmap showing the differentially expressed genes between MBC and FBC CD4⁺ T cells.

Response Figure 18 (Related to Supplementary table 9 in the revised manuscript). Heatmap showing the differentially expressed genes between MBC and FBC NKT cells.

Response Figure 19 (Related to Figure 5f in the revised manuscript). Heatmap showing the differentially expressed genes between MBC and FBC T cells, including CD4⁺, CD8⁺, and NKT cells.

The reason for why T cell clusters seem to be KRT8-negative in Figure 1B is similar to the issue of feature-plot visualization mentioned in *comment #1*. Specifically, the integrated t-SNE plot in Figure 1B included all cells from both MBC and FBC samples. Almost all epithelial cells (90.9%) were KRT8⁺, while only 10.3% of T cells were KRT8⁺ (**Response Figure 20a**). The low proportion of KRT8⁺ T cells resulted in the overlapping of red and grey, showing a dominant grey color in the area of T cell cluster in figure 1B. The percentage of KRT8⁺ T cells was significantly different between MBC and FBC (**Response Figure 20b, c**). KRT8⁺ T cells accounted for 45.3% in T cells from MBC (**Response Figure 20b**), consistent with the observation in Figure 5F (**Response Figure 16-19**). In contrast, only 2.1% of T cells were KRT8⁺ in FBC (**Response Figure 20c**). Thus, we showed the KRT8 expression intensity on the t-SNE plot based on sex and whether T cells were KRT8⁺ (**Response Figure 21**). The results clearly showed that KRT8 was expressed on some T cells, especially the T cells from MBC samples (**Response Figure 21**). These results were added in the revised Supplementary Figure 13. The corresponding description was revised as follows (**Lines 362-366**): “About 50% of T cells from MBC were KRT8⁺, while only 2.1% of T cells from FBC were KRT8⁺ (Supplementary Figure 13d, e). We showed the KRT8

expression intensity on the t-SNE plot based on sex and whether T cells were KRT8⁺, and found that KRT8 was expressed on some T cells, especially the T cells from MBC samples (Supplementary Figure 13f, g)

Response Figure 20 (Related to Figure 13d, e in the revised manuscript). Barplot showing the percentage of KRT8⁺ epithelial cells and T cells in all samples (a), MBC samples (b) and FBC samples (c).

Response Figure 21 (Related to Supplementary Figure 13f, g in the revised manuscript). T-SNE plots showing the cell types and KRT8 expression in MBC (upper panel) and FBC (bottom panel) samples. T cells were circled with dashed lines. The feature-plots were split based on sex and whether T cells were KRT8⁺.

Response References

1. Fleming, S.J., J.C. Marioni, and M. Babadi, *CellBender remove-background: a deep generative model for unsupervised removal of background noise from scRNA-seq datasets*. BioRxiv, 2019: p. 791699.
2. Garcia-Alonso, L., et al., *Benchmark and integration of resources for the estimation of human transcription factor activities*. Genome Res, 2019. **29**(8): p. 1363-1375.
3. Badia-i-Mompel, P., et al., *decoupleR: ensemble of computational methods to infer biological activities from omics data*. Bioinformatics Advances, 2022. **2**(1).
4. Alvarez, M.J., et al., *Functional characterization of somatic mutations in cancer using network-based inference of protein activity*. Nat Genet, 2016. **48**(8): p. 838-47.
5. Sutton, T.L., B.S. Walker, and M.H. Wong, *Digesting the Importance of Cell Fusion in the Intestine*. Cell Mol Gastroenterol Hepatol, 2021. **11**(1): p. 299-302.
6. Ramakrishnan, M., S.R. Mathur, and A. Mukhopadhyay, *Fusion-derived epithelial cancer cells express hematopoietic markers and contribute to stem cell and migratory phenotype in ovarian carcinoma*. Cancer Res, 2013. **73**(17): p. 5360-70.
7. Gast, C.E., et al., *Cell fusion potentiates tumor heterogeneity and reveals circulating hybrid cells that correlate with stage and survival*. Sci Adv, 2018. **4**(9): p. eaat7828.
8. Hu, Z., et al., *The Repertoire of Serous Ovarian Cancer Non-genetic Heterogeneity Revealed by Single-Cell Sequencing of Normal Fallopian Tube Epithelial Cells*. Cancer Cell, 2020. **37**(2): p. 226-242 e7.
9. Chen, S., et al., *Single-cell analysis reveals transcriptomic remodellings in distinct cell types that contribute to human prostate cancer progression*. Nat Cell Biol, 2021. **23**(1): p. 87-98.

REVIEWER COMMENTS

Reviewer #1 (Remarks to the Author):

No additional comments.

Reviewer #3 (Remarks to the Author):

While the additional flow cytometry data is unconvincing, the additional scRNA analyses and immunofluorescence (especially the confocal z stack) provide sufficient evidence for the existence of KRT8 positive CD3+ cells. The authors have addressed all my concerns in this revised manuscript.

Response to reviewer #3:

While the additional flow cytometry data is unconvincing, the additional scRNA analyses and immunofluorescence (especially the confocal z stack) provide sufficient evidence for the existence of KRT8 positive CD3+ cells. The authors have addressed all my concerns in this revised manuscript.

Response: Thank you for your comments. We are pleased to receive your positive evaluation regarding the confocal z-stack images for proving the existence of KRT8⁺CD3⁺ cells, and terribly sorry for the unsatisfying flow cytometry experiments in the previous submission. In order to address the concern of flow cytometry evidence, we made a thorough revision and re-performed the experiments to further exclude the potentially confounding factors and provide convincing results. We used single antibody-labeled compensation samples and fluorescence minus one (FMO) controls to determine where the gates should be appropriately set (**Response Figure 1, 2**). Doublets were excluded according to the FSC-A/FSC-H profile (**Response Figure 1, 2, 3**). Zombie Yellow (Cat: 423104, Biolegend) was used to stain the live/dead cells to exclude the effect of dead cells (**Response Figure 1, 2, 3**).

Response Figure 1 (Related to Supplementary Figure 17 in the revised manuscript). Single antibody-labeled compensation controls of flow cytometry analysis for KRT8, CD45, and CD3.

Response Figure 2 (Related to Supplementary Figure 18 in the revised manuscript). Fluorescence minus one (FMO) controls flow cytometry analysis for KRT8, CD45, and CD3.

Due to the scarcity of male breast cancer (MBC, only accounting for 1% of all breast cancers) samples, the experiments of single staining controls and FMO controls were performed using female breast cancer (FBC) samples with the same subtypes (ER⁺). The full staining experiments were performed using fresh tumor tissues from an MBC patient. **Response Figure 3** showed the full gating strategy of the MBC sample. Firstly, debris was excluded by forward and side scatters gating, and single cells were gated using the FSC-A/FSC-H profile. Dead cells were further excluded using live/dead staining by Zombie. Secondly, KRT8 and CD45 were used to distinguish the epithelial cells (KRT8⁺CD45⁻, 24.0%), immune cells (KRT8⁻CD45⁺, 5.1%), and KRT8⁺CD45⁺ double-positive cells (5.3%). Among the KRT8⁺CD45⁺ double-positive cells, 86.2% were KRT8⁺CD45⁺CD3⁺ T cells. Similarly, 87.8% of KRT8⁻CD45⁺ immune cells were CD3⁺ T cells. To better determine the T cell subpopulations, the KRT8⁺CD45⁺CD3⁺ and KRT8⁻CD45⁺CD3⁺ T cells were backgated and overlaid onto the FSC-A/SSC-A plots. Results showed that both KRT8⁺CD45⁺CD3⁺ and KRT8⁻CD45⁺CD3⁺ T cells were located in the lymphocyte gate. Among all T cells (CD45⁺CD3⁺), KRT8⁺ and KRT8⁻ cells accounted for 50.5% and 49.5% in this MBC sample, respectively.

Response Figure 3 (Related to Figure 6d in the revised manuscript). Full gating strategy of flow cytometry analysis for the identification of KRT8⁺ and KRT8⁻ T cells in an MBC sample.

The raw FCS files (including the single staining controls, FMO controls, and full-staining experiments) of the above flow cytometry data have been submitted to Mendeley Data (<https://data.mendeley.com/datasets/wwm9xv56ry/1>).

Correspondingly, the results of flow cytometry experiments were revised in this submission (*Lines 427-443*): “Besides, we performed flow cytometry experiments for fresh tumor tissues from an MBC patient to validate and quantify CD3⁺KRT8⁺ double-positive T cells (Figure 6d). Single antibody-labeled compensation samples and fluorescence minus one (FMO) controls were used to determine where the gates should be set (Supplementary Figure 17, 18). Firstly, debris was excluded by forward and side scatters gating, and single cells were gated using the FSC-A/FSC-H profile. Dead cells were further excluded using live/dead staining by Zombie. Secondly, KRT8 and CD45 were used to distinguish the epithelial cells (KRT8⁺CD45⁻, 24.0%), immune cells (KRT8⁻CD45⁺, 5.1%), and KRT8⁺CD45⁺ double-positive cells (5.3%). Among the KRT8⁺CD45⁺ double-positive cells, 86.2% were KRT8⁺CD45⁺CD3⁺ T cells. Similarly, 87.8% of KRT8⁻CD45⁺ immune cells were CD3⁺ T cells. To better determine the T cell subpopulations, the KRT8⁺CD45⁺CD3⁺ and KRT8⁻CD45⁺CD3⁺ T cells were backgated and overlaid onto the FSC-A/SSC-A plots. Results showed that both KRT8⁺CD45⁺CD3⁺ and KRT8⁻CD45⁺CD3⁺ T cells were located in the lymphocyte gate. Among all T cells (CD45⁺CD3⁺), KRT8⁺ and KRT8⁻ cells accounted for 50.5% and

49.5% in this MBC sample, respectively. Therefore, these results indicated the biological existence of KRT8⁺CD45⁺CD3⁺ T cells”. In addition, we added the corresponding methodological description in the revised manuscript as follows (*Lines 744-754*): “Flow cytometry was performed using a FACSLyric flow cytometer (BD Biosciences). The intrinsic spectral overlap of the different fluorochromes was corrected using compensation matrices. Due to the scarcity of MBC samples, the experiments of single antibody-labeled compensation controls and FMO controls were performed using ER⁺ FBC samples. The full staining experiments were performed using fresh MBC tumor tissues. Doublets were excluded according to the FSC-A/FSC-H profile. Zombie Yellow (Cat: 423104, Biolegend) was used to exclude the dead cells. All the flow cytometry data were analyzed using FlowJo software (Version 10.8.1, FlowJo LLC). The raw FCS files are deposited in Mendeley Data (<https://data.mendeley.com/datasets/wwm9xv56ry/1>)”.

Therefore, evidence of scRNA-seq analysis, immunofluorescence (including confocal z-stack images), and flow cytometry collectively demonstrated the existence of KRT8⁺CD3⁺ T cells in MBC samples (see **main Figures 5, 6 and Supplementary Figures 13, 14, 15, 16, 17, 18, 19 in this submission**).

REVIEWERS' COMMENTS

Reviewer #3 (Remarks to the Author):

The added FACS analysis and gating strategy have provided additional evidence for the existence of KRT8+/CD3+ cells and the authors have addressed all my concerns

The fourth revision

Reviewer #3 (Remarks to the Author):

The added FACS analysis and gating strategy have provided additional evidence for the existence of KRT8+/CD3+ cells and the authors have addressed all my concerns

Response: We thank the reviewer for providing many professional suggestions and advice. We are delighted that all concerns have been successfully addressed.